# Adult skull bone marrow is an expanding and resilient haematopoietic reservoir

Bong Ihn Koh[1✉], Vishal Mohanakrishnan[1], Hyun-Woo Jeong[2], Hongryeol Park[1], Kai Kruse[3], Young Jun Choi[4], Melina Nieminen-Kelhä[5], Rahul Kumar[6], Raquel S. Pereira[7], Susanne Adams[1], Hyuek Jong Lee[8], M. Gabriele Bixel[1], Peter Vajkoczy[5], Daniela S. Krause[6] & Ralf H. Adams[1✉]

The bone marrow microenvironment is a critical regulator of haematopoietic stem cell self-renewal and fate[1]. Although it is appreciated that ageing, chronic inflammation and other insults compromise bone marrow function and thereby negatively affect haematopoiesis[2], it is not known whether different bone compartments exhibit distinct microenvironmental properties and functional resilience. Here we use imaging, pharmacological approaches and mouse genetics to uncover specialized properties of bone marrow in adult and ageing skull. Specifically, we show that the skull bone marrow undergoes lifelong expansion involving vascular growth, which results in an increasing contribution to total haematopoietic output. Furthermore, skull is largely protected against major hallmarks of ageing, including upregulation of pro-inflammatory cytokines, adipogenesis and loss of vascular integrity. Conspicuous rapid and dynamic changes to the skull vasculature and bone marrow are induced by physiological alterations, namely pregnancy, but also pathological challenges, such as stroke and experimental chronic myeloid leukaemia. These responses are highly distinct from femur, the most extensively studied bone marrow compartment. We propose that skull harbours a protected and dynamically expanding bone marrow microenvironment, which is relevant for experimental studies and, potentially, for clinical treatments in humans.

Niche microenvironments composed of multiple cell types and molecular signals regulate the self-renewal and the ultimate fate of haematopoietic stem cells (HSCs) in bone marrow[1,3] (BM). The skull, a flat bone encasing the brain, develops via intramembranous ossification and fundamentally differs in both composition and remodelling from long bones, which undergo endochondral ossification during development and sustain load-bearing mechanical stress[4–6]. The thin and flat mouse calvarium, the main component of the skull roof, has been successfully used as a platform for the imaging of HSC dynamics[7,8]. Potential functional differences between BM compartments, such as those in long and flat bone, however, remain little understood. It has been shown that HSCs are homogenously distributed throughout calvarium and long bone (namely diaphysis and epiphysis) under homeostatic conditions, whereas chimerism in mice transplanted with congenic haematopoietic stem and progenitor cells (HSPCs) is initially higher in calvarium and epiphysis relative to diaphysis[9]. The discovery of designated channels between the calvarial BM and the dura mater of the meninges indicates that skull BM acts as a haematopoietic reservoir for meningeal and central nervous system immunity[10,11]. Ageing causes an overall decline in haematopoietic function, with compromised lymphopoiesis and a bias toward dysfunctional myelopoiesis[2,12]. Again, it remains unknown whether all BM compartments are equally affected by ageing processes and therefore collectively contribute to haematopoietic decline. Here we show that calvarial BM is fundamentally different to the BM in long bone. Calvarial BM continuously expands during adulthood, is resistant to most major hallmarks of BM niche ageing, and gradually increases its systemic haematopoietic contribution throughout adult life and ageing. Our work also establishes that BM compartments exhibit unexpected functional differences, with implications for a range of physiological conditions such as pregnancy and ageing, as well as pharmacological treatments and disease settings.

## Skull BM expands during adult life

To characterize changes in calvarial BM throughout adult life, we collected skulls from young adult (10–14 weeks; hereafter referred to as 'young'), middle-aged (31–37 weeks), old (52–75 weeks) and geriatric (95+ weeks) mice. Initial stereoscopic observation showed that blood-filled structures, representing blood vessels and surrounding BM, occupy only limited areas of the frontal and parietal bone in young adults, whereas interparietal bone is mostly filled by blood cells (Fig. 1a). Although angiogenic growth of blood vessels is normally confined to

[1]Department of Tissue Morphogenesis, Max Planck Institute for Molecular Biomedicine, Münster, Germany. [2]Sequencing Core Facility, Max Planck Institute for Molecular Biomedicine, Münster, Germany. [3]Bioinformatics Service Unit, Max Planck Institute for Molecular Biomedicine, Münster, Germany. [4]Department of Radiology and Research Institute of Radiology, Asan Medical Center, University of Ulsan College of Medicine, Seoul, Republic of Korea. [5]Department of Neurosurgery, Charité-Universitätsmedizin Berlin, corporate member of Freie Universität Berlin and Humboldt-Universität zu Berlin, Berlin, Germany. [6]Institute of Transfusion Medicine, Transfusion Center, University Medicine Mainz, Mainz, Germany. [7]Georg-Speyer-Haus Institute for Tumor Biology and Experimental Medicine and Goethe University Frankfurt, Frankfurt, Germany. [8]Center for Vascular Research, Institute for Basic Science, Daejeon, Republic of Korea. ✉e-mail: bong-ihn.koh@mpi-muenster.mpg.de; ralf.adams@mpi-muenster.mpg.de

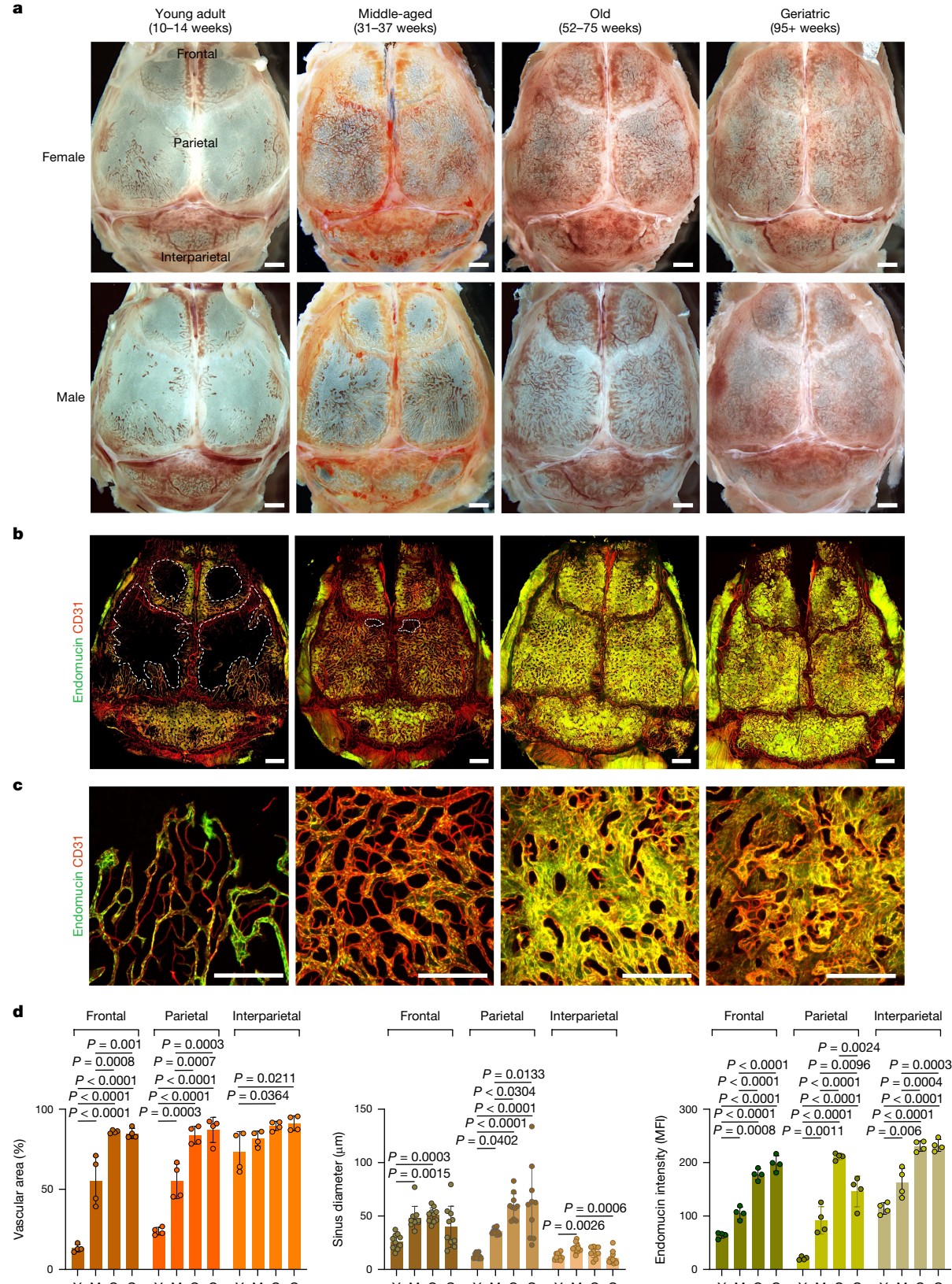

**Fig. 1 | Age-related expansion of blood vessels in adult skull. a.** Transverse view of mouse skull at the indicated stages of adulthood and ageing. Scale bars, 1 mm. **b,c,** In vivo immunofluorescence staining of blood vessels in skull showing substantial vascular expansion (**b**) and changes in vessel branching and morphology (**c**). Representative images from three independent experiments. Scale bars, 1 mm (**b**) and 500 μm (**c**). **d,** Quantification of vascular area, diameter and endomucin expression in different skull parts from young (Y), middle-aged (M), old (O) and geriatric (G) mice. For vascular area and endomucin expression, each dot indicates a value from 1 mouse and $n = 4$ mice per group from 3 independent experiments. For sinus diameter, each dot indicates a randomly selected vessel from all four mice. Data are mean ± s.d. $P$ values from Tukey multiple comparison test (one-way analysis of variance (ANOVA)). MFI, mean fluorescence intensity.

development, regeneration or certain pathological conditions[13,14], substantial expansion of the vasculature can be seen in calvaria from middle-aged mice (Fig. 1a–d). Further increases in vascular area and vessel diameter can be observed in old and geriatric skulls (Fig. 1a–d). Flow cytometric analyses show that the expansion of calvarial vasculature is accompanied by a profound increase in total haematopoietic cells, HSCs, HSPCs, committed haematopoietic progenitors and stromal cells (Supplementary Data 1; gating strategies shown in Supplementary Data 2). Skull cross-sections from young and geriatric mice show that the enlargement of calvarial BM occurs more rapidly in female mice than in males (Fig. 1a and Extended Data Fig. 1a,b). Expansion of calvarial BM results in an increase in total skull thickness (of approximately 70% between young and geriatric mice of both sexes) without decreasing the thickness of the cortical bone tables (Extended Data Fig. 1a,b). Immunofluorescence analysis of skull sections shows a profound increase in V-type proton ATPase-positive activated osteoclasts lining the surface of the inner and outer cortical bone tables in skulls from old relative to young mice (Extended Data Fig. 1c,d), indicating that bone resorption by osteoclasts contributes to BM expansion.

To address whether skull BM expansion also occurs in ageing humans, we analysed computed tomography (CT) head scan images obtained from 36 human patients evaluated for small cerebral aneurysms. This revealed significantly increased area enclosed by the inner and outer cortical bone tables in older (61–69 years of age) compared with younger (21–40 years of age) individuals (Extended Data Fig. 2a–c). A larger difference was seen in female skulls (83.2% increase versus 24.6% in males), showing that this sexual dichotomy occurs in both mice and humans. These results demonstrate a substantial expansion of the BM volume in both mouse and human skull during adulthood and ageing.

## Vessel growth during skull BM expansion

Previous studies have shown that arterial and sinusoidal endothelial cells are important components of the HSC microenvironment[15,16]. To gain more insight into the organization of the calvarial vasculature, we first utilized an in vivo immunofluorescence staining method with CD31 antibody in *Flk1*-GFP knock-in mice[17] (*Flk1* is also known as *Kdr*). This approach revealed profound differences in vascular architecture in young adult frontal, parietal and interparietal skull bones (Extended Data Fig. 3a,c). Specifically, larger FLK1+CD31+ sinusoidal vessels with distal sprouts dominate in frontal bone, thinner FLK1+CD31+ sinusoidal vessel loops connect to FLK1low/−CD31+ arterioles in parietal bone, and interparietal bone is completely filled by a dense network of sinusoidal vessels and arterioles (Extended Data Fig. 3a,c). Vascular leakage assays with Evans Blue corroborated local differences in vascular properties in different skull parts and revealed significantly higher leakage in frontal bone compared with the other two skull bones (Extended Data Fig. 3b,d).

To gain more insight into the organization of the calvarial vasculature and the changes during adulthood and ageing, we performed in vivo labelling with fluorescent CD31 and endomucin antibodies. In young adult mice, only interparietal bone was largely filled by a dense vascular network, whereas substantial and sustained vascular expansion was observed in frontal and parietal bones of middle-aged and old mice (Fig. 1b–d). Vascular area increased by 6.4- and 3.6-fold in frontal and parietal bones, respectively, between young and geriatric mice. Endomucin, which decorates calvarial sinusoidal vessels but not arteries and smaller arterioles, has been implicated in bone maintenance and HSC function[18,19]. Vascular endomucin expression significantly increased until the geriatric stages (3.1-fold increase in frontal bone and 7.0-fold increase in parietal bone). We next focused on parietal bone, which showed the largest BM expansion and a transition from thin looping sinusoidal vessels in young adult mice to a dense network at later stages (Fig. 1b,c). Sinus diameter increased 4.8-fold and dorsal–ventral vertical extension of the vascular network can be

observed at the old stage (Fig. 1c,d and Extended Data Fig. 3e). Immunofluorescence staining of caveolin-1[19,20] showed a 2.0-fold increase of interconnecting arterioles during ageing (Extended Data Fig. 3f,g). In vivo immunofluorescence staining of haematopoietic cells using fluorescent CD45 antibody and flow cytometric analyses show that vascular growth is associated with a substantial increase in haematopoietic cells (13.1-fold increase between young and geriatric mice) within the expanding BM (Extended Data Fig. 4a,b and Supplementary Data 1). CD45+ cells were predominantly associated with large-calibre sinusoidal vessels, which were highly abundant in old and geriatric skull (Extended Data Fig. 4a). The link between vessel calibre and BM was further confirmed by a genetic approach; labelling of haematopoietic cells in the *Vav-cre;Rosa26-mTmG* background[21,22] shows that haematopoietic cells expressing GFP surround large-calibre sinusoidal vessels (Extended Data Fig. 4c). Similarly, CD3e+ T lymphocytes, B220+ B lymphocytes and CD11b+ cells (including monocytes/macrophages, granulocytes and natural killer cells) are associated with large-calibre sinusoids in skull BM (Extended Data Fig. 4d). With the exception of interparietal bone, BM-associated sinusoidal vessels with strong endomucin immunostaining are comparably sparse in young adult mice and increase substantially throughout adult life and ageing (Fig. 1b–d).

Interparietal bone, which is largely filled by vessels and BM at the young adult stage, shows minimal age-related changes in vascular area (24.3% increase between young and geriatric samples) and sinus diameter (9.3% decrease between young and geriatric samples) but a significant increase in endomucin expression (2.0-fold increase between young and geriatric samples) (Fig. 1b,d). By contrast, vascular density in the metaphyseal region of the femoral BM decreased by 45.2% between the middle-aged and old stages, but rebounded at geriatric stages with the emergence of very thin vessels displaying high endomucin expression (Extended Data Fig. 5a,b), which have been recently characterized as dysfunctional sinusoidal vessels[23]. Electron micrographs of the luminal surface of endothelial cells in young and old skull BM show comparable patterns of endothelial fenestrations, which are a hallmark of vessels in primary lymphoid organs including BM. By contrast, endothelial cells in old femur BM show a highly irregular pattern of fenestrae and the emergence of larger pores in the endothelial surface (Extended Data Fig. 5c). In contrast to the increasing vasculature of skull marrow, vessel density in the dura mater, which is adjacent to the skull and connected directly to the skull BM via specialized channels[10], decreases significantly during ageing (Extended Data Fig. 5d,e). Collectively, these results show continuous expansion of the BM and vasculature in the adult and ageing calvarium but not in the adjacent dura mater. Moreover, ageing femoral BM displays features of compromised vascular integrity, which are not seen in ageing skull BM.

## Skull BM responses in pathophysiology

High demand for certain blood cells leads to strongly increased haematopoiesis in a range of physiological but also pathological conditions[1,24]. Pregnancy, for example, induces extramedullary haematopoiesis and the rapid expansion of maternal blood volume and red blood cells[25,26]. Skull BM and associated vasculature increased substantially during pregnancy, namely 1.8-fold at 17 days postcoitum (dpc) and 2.3-fold at 2 days postpartum (dpp), which involves substantial vessel enlargement in frontal and parietal bone together with a strong increase in total haematopoietic cells, HSCs and HSPCs, committed haematopoietic progenitors and stromal cells (Fig. 2a,b and Supplementary Data 1). As reported previously[27], CD31hiendomucinhi type H vessels in femur increased strongly at late-stage pregnancy and decreased postpartum, but overall vascular density in the metaphyseal region showed more limited changes (26.2% increase at 17 dpc and 8.5% decrease at 2 dpp) relative to skull (Extended Data Fig. 6a,b). Although the total number of haematopoietic cells in femur was not significantly changed by pregnancy, flow cytometry showed increases

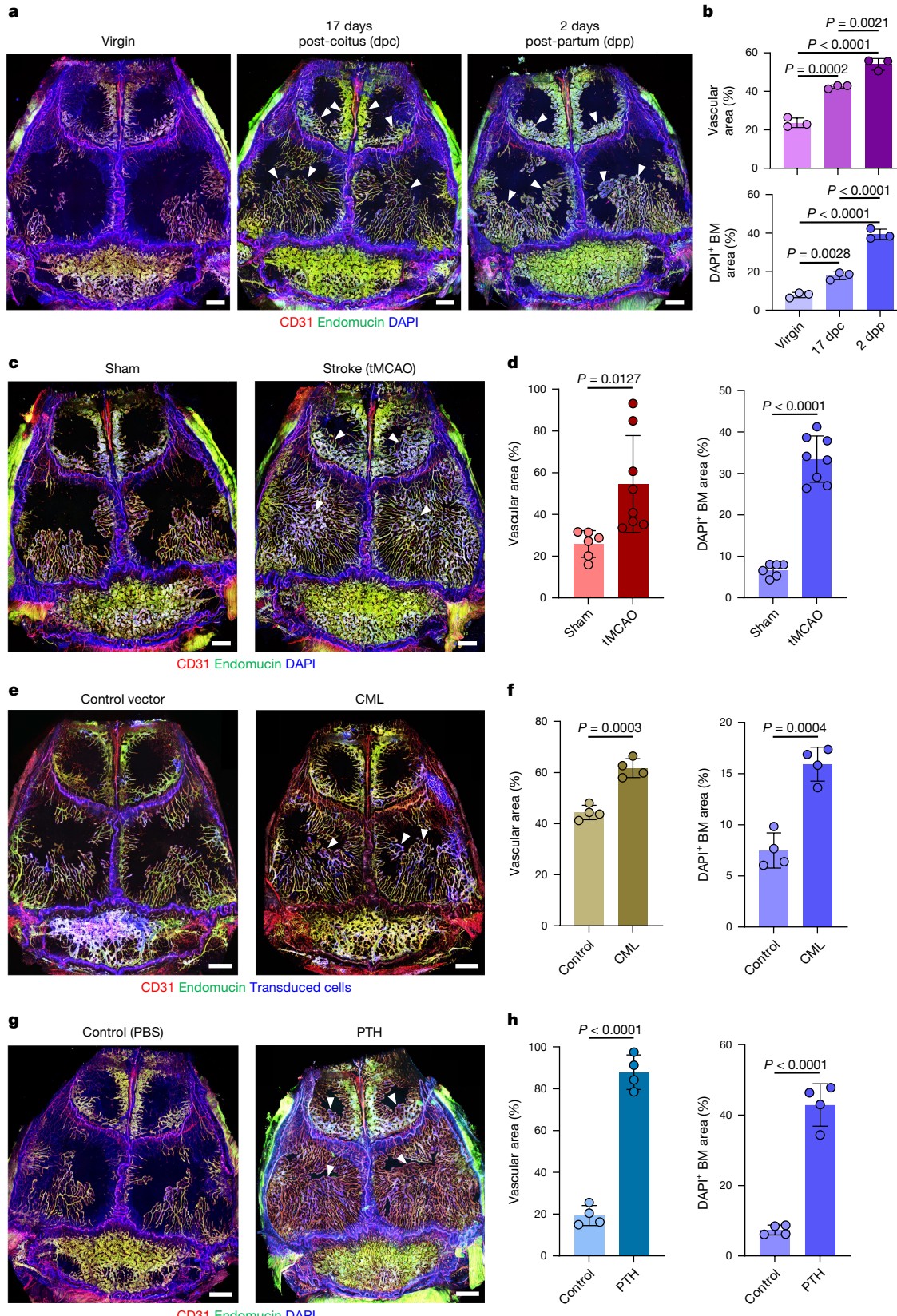

**Fig. 2 | Pathophysiological regulation of vessel growth and BM expansion in adult skull. a,b,** In vivo immunofluorescence (**a**) and quantification (**b**) of skull blood vessels and BM in pregnant (17 dpc) and postpartum (2 dpp) female mice. $n = 3$ mice per group from 3 independent experiments. **c,d,** In vivo immunofluorescence (**c**) and quantification (**d**) of skull blood vessels and BM in mice 7 days after transient mid-cerebral artery occlusion. $n = 6$ (sham) and $n = 8$ (tMCAO) mice per group from 3 independent experiments. **e,f,** In vivo immunofluorescence (**e**) and quantification (**f**) of skull blood vessels and BM in mice with CML. $n = 4$ mice per group from 3 independent experiments. **g,h,** In vivo immunofluorescence (**g**) and quantification (**h**) of skull blood vessels and BM in mice with 28-day sustained PTH treatment. $n = 4$ mice per group from 3 independent experiments. Arrowheads indicate areas of substantial expansion. Scale bars, 1 mm. Data are mean ± s.d. $P$ values by Tukey multiple comparison test (one-way ANOVA) and two-tailed unpaired Student's $t$-test.

in HSCs, HSPCs, certain committed haematopoietic progenitors and stromal cells (Supplementary Data 1), consistent with enhanced blood cell production.

Ischaemic stroke in a transient mid-cerebral artery occlusion model (tMCAO) was shown to activate BM haematopoietic stem cells[28]. Consistent with this finding, tMCAO induced a striking expansion of skull BM and associated vasculature (2.1-fold increase), filling a substantial portion of parietal bone within 7 days (Fig. 2c,d). These changes are accompanied by strong increases in calvarial total haematopoietic cells, HSCs, HSPCs, committed haematopoietic progenitors and stromal cells (Supplementary Data 1). By contrast, vascular density in femoral BM decreased by 26.0% and haematopoietic cells and subpopulations were either unchanged or reduced in the stroke condition (Extended Data Fig. 6c,d and Supplementary Data 1).

Haematological malignancies are known to lead to BM alterations, which, in turn, facilitate disease progression[29]. A chronic myeloid leukaemia (CML) model showed a modest but significant expansion of the skull BM and associated blood vessels (39.1% increase), which involved an increase in haematopoietic and stromal cells (Fig. 2e,f and Supplementary Data 1). By contrast, substantial loss of vascularity (50.4% decrease), severe vascular rarefaction and limited changes in total haematopoietic cell number were observed in CML femoral BM (Extended Data Fig. 6e,f and Supplementary Data 1).

Biologically active fragments of parathyroid hormone (PTH) are used to treat osteoporosis[30], and also lead to HSPC expansion[31]. We found that 4 weeks of daily PTH treatment resulted in substantial skull BM expansion (4.6-fold increase), increased large-calibre vessels and expansion of vessel-associated BM area (Fig. 2g,h). Flow cytometry confirmed the increase in total haematopoietic cell number, HSCs, HSPCs and certain committed progenitors, as well as in endothelial and stromal cells (Supplementary Data 1). By contrast, only very limited changes were observed in femoral BM immunostaining and flow cytometry (Extended Data Fig. 6g,h and Supplementary Data 1). These results demonstrate an unexpected dynamic plasticity of skull BM in a range of pathophysiological contexts and uncover differences between calvarial and femoral BM.

## Skull HSPCs drive BM growth during ageing

HSPCs residing in BM change drastically during ageing, both in quantity and differentiation potential[32]. To address whether such cells might promote skull BM expansion, we isolated lineage-negative (Lin⁻) cells from young or old skull and femur and transplanted them into lethally irradiated young recipients (Fig. 3a). In vivo immunofluorescence staining showed substantial expansion (2.2-fold) of skull BM vessels in recipient mice that received cells from old donor skull BM compared with those that received cells from young donors (Fig. 3b,c). Transplantation of old donor femur BM also resulted in the expansion of skull BM vessels in recipient mice but to a lesser degree than old donor skull BM. Notably, vascular density in femur was not significantly different between recipients of transplants from young and old donors (Extended Data Fig. 7a,b), further suggesting divergent responsiveness of different BM compartments. Flow cytometric analyses of HSPCs showed that both long-term HSCs (LT-HSCs) and Lin⁻Sca1⁺KIT⁺ (LSK) HSPCs increase in number in the ageing skull BM (Fig. 3d,e), similar to reports for ageing long bone[33,34]. To further test whether HSPCs contribute to BM expansion and vascular growth in skull, we treated mice with prostaglandin E₂[35,36] (PGE₂) or AMD3100 (ref. 37) to modulate HSPC number. PGE₂ treatment increased the number of HSPCs in the skull BM by 92.3% (Extended Data Fig. 7c,d), resulting in significant expansion of calvarial vasculature and vessel-associated BM relative to control mice (Fig. 3f,g and Supplementary Data 1). AMD3100 treatment mobilized HSPCs into the peripheral blood by 11.2-fold (Extended Data Fig. 7e,f) and induced a 60.0% reduction of skull BM and vasculature relative to control

mice (Fig. 3h,i). Although both PGE₂ and AMD3100 treatments led to changes in vascular density in femoral BM, these alterations were confined to existing marrow without changing the BM area (Extended Data Fig. 7g–j). These results indicate that treatments acting on HSPCs can dynamically modulate the skull BM and its vasculature.

## VEGFA drives skull BM and vessel growth

VEGFA is a master regulator of vascular growth[38,39], and enzyme-linked immunosorbent assays (ELISAs) showed that levels of the growth factor were significantly increased in the skull BM in response to pregnancy, stroke and PTH treatment, whereas no changes could be detected in the femoral BM under the same conditions (Extended Data Fig. 8a). VEGFA was significantly reduced in the femoral BM of mice with CML but was increased in the skulls of the same mice (Extended Data Fig. 8a).

Previous studies showed that HSPCs isolated from long bone express high levels of *Vegfa* transcripts compared with other BM cell types[40]. We performed quantitative PCR with reverse transcription (RT–qPCR) analyses of various haematopoietic and stromal cell populations after fluorescence-activated cell sorting (FACS) from enzymatically digested BM. Lin⁻Sca1⁺KIT⁺ HSPCs from skull indeed showed higher *Vegfa* expression compared with Lin⁻Sca1⁻KIT⁺ cells (4.72-fold), Lin⁻Sca1⁻KIT⁻ stromal cells (5.97-fold) or Lin⁺ mature haematopoietic cells (5.92-fold) in young mice (Fig. 4a). HSPCs isolated from aged skull show higher *Vegfa* expression compared with Lin⁻Sca1⁻KIT⁺ cells (8.39-fold), Lin⁻Sca1⁻KIT⁻ stromal cells (10.59-fold) and Lin⁺ mature haematopoietic cells (9.86-fold) (Fig. 4a). By contrast, HSPCs isolated from aged femur show significantly lower *Vegfa* expression (5.5-fold) compared with the same population from young mice (Extended Data Fig. 9a). These results are mirrored by the expression of transcripts encoding VEGFA₁₂₀ and VEGFA₁₆₄ isoforms, which are highest in HSPCs from old skull, but also substantially higher in HSPCs from young mice compared with other cell populations (Extended Data Fig. 9b).

At the protein level, ELISA of whole-BM lysates shows that VEGFA levels are initially highest in young femur, probably reflecting expression by non-haematopoietic sources such as chondrocytes, and decrease by 40.2% during ageing (Fig. 4b). By contrast, the amount of VEGFA in calvarial BM increases 2.2-fold from young to geriatric stages (Fig. 4b). Immunofluorescence of skull sections confirms high expression of VEGFA by resident KIT⁺ HSPCs (Fig. 4c). As hypoxia is a critical regulator of VEGFA expression in angiogenesis[41,42], we administered Hypoxyprobe (pimonidazole hydrochloride) to analyse age-related changes in the oxygenation of skull and femur (Extended Data Fig. 9c,d). The Hypoxyprobe signal was significantly higher in skull BM relative to femoral BM at old (3.3-fold) and geriatric stages (4.8-fold) (Fig. 4d,e and Extended Data Fig. 9c,d), consistent with the observed increase of VEGFA in ageing skull.

Next, we manipulated VEGFA-dependent VEGFR2 signalling in vivo to directly address the role of this pathway in the expansion of skull vasculature and BM. For gain-of-function experiments, a cDNA encoding a bone-homing version of VEGFA (Methods) was cloned into the vector pLIVE, which enables constitutive protein expression in liver after hydrodynamic tail vein injection[43]. *Vegfa* overexpression induced a 2.5-fold increase of skull BM, including HSCs and other haematopoietic and stromal cells, and associated vasculature (Fig. 4f,g and Supplementary Data 1). Conversely, inhibition of VEGFR2 signalling with the blocking antibody DC101 inhibited BM expansion and vascular growth during ageing, as shown by immunostaining and flow cytometry (Fig. 4h,i and Supplementary Data 1). By contrast, although we observed vascular density changes within the confines of the femoral BM, these alterations did not involve any substantial volumetric changes of the BM as in the skull (Extended Data Fig. 9e–j). These results demonstrate that VEGFA–VEGFR2 signalling regulates skull BM expansion during ageing and that the increase of HSPCs in ageing skull is likely to be a relevant source of VEGFA.

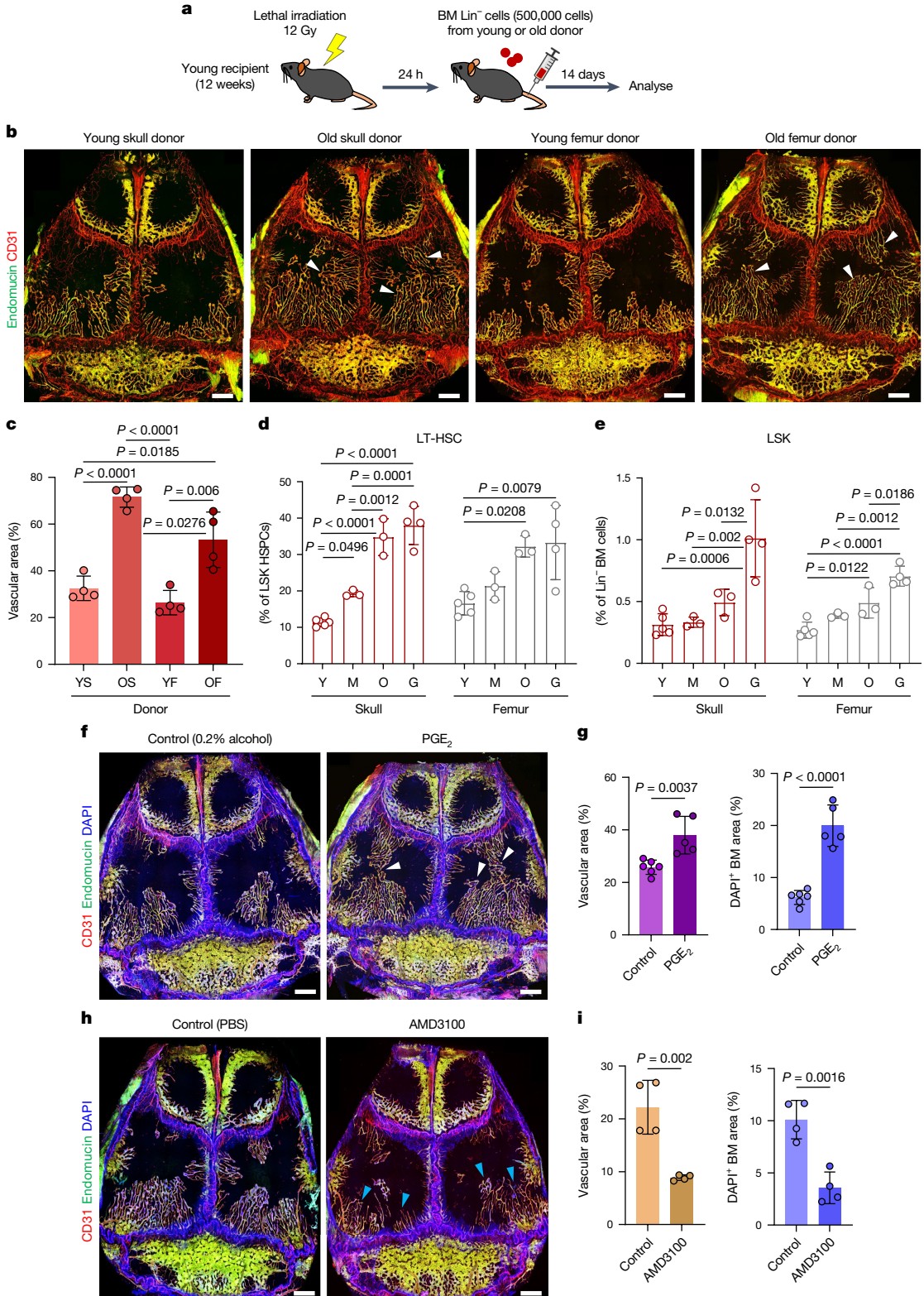

**Fig. 3 | Regulation of calvarial vessels by haematopoietic cells.**
**a**, Experimental scheme for lethal irradiation and transplantation of haematopoietic lineage-depleted BM population isolated from young or old donor mice. **b,c**, In vivo immunofluorescence (**b**) and quantification (**c**) of skull blood vessels and BM in transplanted mice showing increased vascular expansion in recipients receiving Lin⁻ cells from old donors. $n = 4$ mice per group from 3 independent experiments. Arrowheads indicate areas of substantial BM vascular expansion. YS, young adult skull; OS, old skull; YF, young adult femur; OF, old femur. **d,e**, Quantification of HSPCs in young adult, middle-aged, old

and geriatric skull and femur by FACS. $n = 5$ (young adult), $n = 3$ (middle-aged and old) and $n = 4$ (geriatric) mice per group from 3 independent experiments. **f–i**, In vivo immunofluorescence staining (**f,h**) and quantification (**g,i**) of skull blood vessels and BM in mice treated with PGE$_2$ or AMD3100 to promote HSPC expansion or mobilization, respectively. $n = 6$ (PGE$_2$ control), $n = 5$ (PGE$_2$) and $n = 4$ (all other groups) mice per group from 3 independent experiments. Arrowheads indicate areas of substantial BM expansion (white) and regression (blue). Scale bars, 1 mm. Data are mean ± s.d. $P$ values by two-tailed unpaired Student's $t$-test and Tukey multiple comparison test (one-way ANOVA).

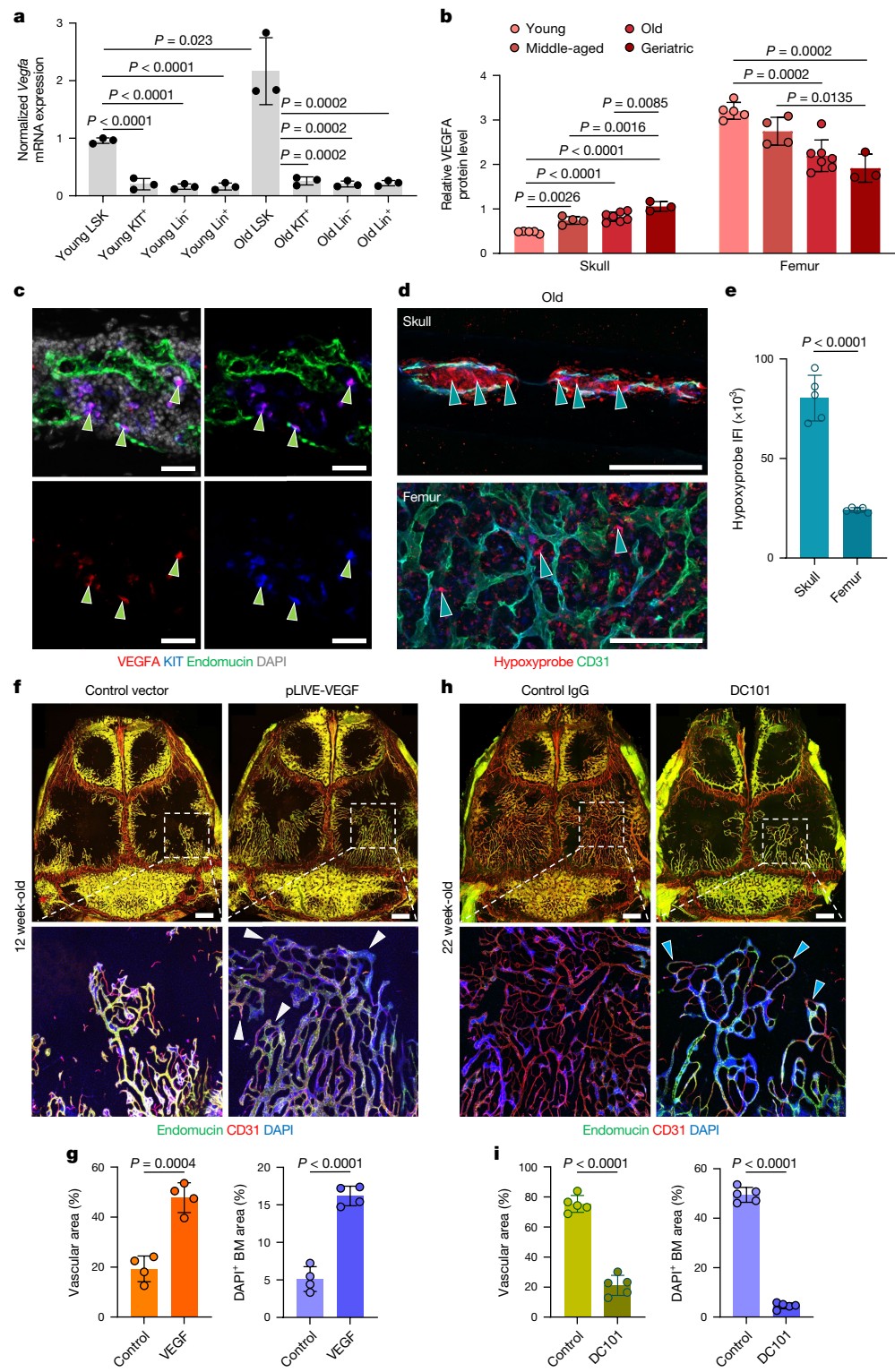

**Fig. 4 | Role of VEGF in calvarial vascular growth. a**, RT−qPCR analyses of *Vegfa* mRNA expression in FACS-sorted LSK (Lin⁻Sca1⁺KIT⁺), KIT⁺ (Lin⁻Sca1⁻KIT⁺), lineage-negative (Lin⁻Sca1⁻KIT⁻) and lineage-positive (Lin⁺) populations isolated from young or old skull BM. *n* = 5 pooled mice per sample from 3 independent experiments. **b**, VEGFA protein concentrations in total BM lysates from skull or femur isolated at various stages of adulthood and ageing. *n* = 5 young, *n* = 4 middle-aged, *n* = 7 old and *n* = 3 geriatric mice per group from 3 independent experiments. **c**, Immunofluorescence of skull BM sections showing high anti-VEGFA signal in KIT⁺ HSPCs (arrowheads). Representative images from three independent experiments. Scale bars, 20 μm. **d,e**, Immunofluorescence (**d**) and quantification (**e**) of intravenously injected Hypoxyprobe in old skull or femoral BM. *n* = 5 mice per group from 2 independent experiments. Arrowheads

indicate labelled cells. Scale bars, 200 μm. IFI, integrated fluorescence intensity. **f,g**, In vivo immunofluorescence (**f**) and quantification (**g**) of calvarial blood vessels in mice expressing bone-homing VEGFA. Note the substantial increase in BM and expansion of vessels (arrowheads) and vascular area. *n* = 4 mice per group from 3 independent experiments. Scale bars, 1 mm. **h,i**, In vivo immunofluorescence (**h**) and quantification (**i**) of calvarial blood vessels in mice treated with VEGFR2 blocking antibody (DC101) for 12 weeks, showing profound inhibition of BM expansion, suppressed vessel growth (arrowheads) and with decrease in vascular diameter. *n* = 5 mice per group from 2 independent experiments. Scale bars, 1 mm. Data are mean ± s.d. *P* values by Tukey multiple comparison test (one-way ANOVA) and two-tailed unpaired Student's *t*-test.

## Resilience of ageing skull BM

The ageing BM microenvironment disrupts normal self-renewal and other essential functions of HSCs[44]. To assess whether hallmarks of ageing affect calvarial and femoral BM equally, we characterized adipogenesis and inflammation in both compartments in young and geriatric mice. Adipocytes have previously been reported to accumulate in the ageing BM and negatively regulate HSC function[45]. Whole-mount or cross-sectional staining of skull or femur, respectively, with BODIPY shows minimal changes in the number of lipid-filled mature adipocytes in the skull, whereas a substantial 5.9-fold increase in adipocytes can be observed in the metaphyseal region of the ageing femoral BM (Fig. 5a,b). Next, we used a bead-based multiplex array to quantify 12 pro-inflammatory cytokines in lysates from skull and femoral BM isolated from young and old mice. Of note, only IFNγ was significantly upregulated in old skull, whereas 7 out of 12 cytokines, including TNF and IL-6, showed increased expression in old femur relative to young femur (Fig. 5c). It should also be noted that the expression of most pro-inflammatory cytokines in old femur was substantially higher than in skull, further supporting the notion that detrimental processes associated with ageing are more prominent in femur.

Ageing HSCs exhibit a myeloid bias at the expense of lymphopoiesis[46]. To assess whether such myeloid bias occurs and determine which stages of the myeloid differentiation cascade are affected in the ageing skull, we analysed myeloid-committed progenitors and mature myeloid cells in the geriatric skull or femoral BM using flow cytometry. We detected significant proportional increases of Lin⁻Sca1⁻KIT⁺ multipotent, common myeloid and granulocyte-monocyte progenitors, but not megakaryocyte-erythroid progenitors, in the femur compared with the skull (Fig. 5d). Furthermore, RT–qPCR analyses of the key myeloid determination factors *Cebpa* and *Spi1* (which encodes the transcription factor PU.1) show substantial upregulation in the geriatric femur BM versus skull BM (Fig. 5e). Finally, to compare the myeloid output from skull versus hindlimb BM into peripheral blood, we selectively shielded these areas during lethal irradiation of young and old mice (Fig. 5f). Whereas there was no difference in myelopoiesis between head or leg shielding in young mice, peripheral blood output from hindlimbs of old mice showed a strong myeloid bias compared with skull (Fig. 5f,g). Together, these results show that calvarial BM is remarkably resilient against key features of ageing and thereby maintains a healthy microenvironment for haematopoiesis even in old mice.

## Old skull preserves functional BM niches

It has been established that skull BM contains functional HSCs that can fully reconstitute the entire haematopoietic system in a transplantation setting[9] and is the source of meningeal immune cells[11,47]. The effect of ageing on haematopoietic output by different BM compartments, however, remains unknown. Following lethal irradiation in a non-transplantation setting, all mice with whole-body exposure died within 12 days of BM failure (Fig. 5h). However, all young adult mice that were protected by shielding of the head or hindlimbs survived past 16 weeks without transplantation of exogenous BM, demonstrating that both bone compartments contain sufficient HSCs for survival (Fig. 5h). Secondary haematopoietic reconstitution following an additional round of lethal irradiation with head or hindlimb shielding also demonstrated the long-term reconstitution potential of HSCs derived from both BM compartments (Extended Data Fig. 10a,b). Whereas shielding of the head of old mice was also fully compatible with long-term survival after irradiation, the same was not the case for hindlimb shielding, and none of the mice in this group survived beyond 200 days (Fig. 5h).

The calvarial vasculature is also highly resilient against irradiation-induced changes. Consistent with our previous findings[40], vascular density (10.2-fold) and diameter (3.3-fold) were substantially increased in femoral BM at 7 days after irradiation. By contrast, vessels in skull showed

minimal changes both with and without BM transplantation (Extended Data Fig. 10c–e). To explore the potential mechanisms that contribute to BM heterogeneity in ageing skull and femur, we performed single-cell RNA sequencing (scRNA-seq) of KIT⁺ HSPCs, and stromal and endothelial cells isolated from either BM compartment in young or geriatric mice (Extended Data Fig. 11a–d). We found significantly higher expression of factors associated with stress and inflammation such as *Jund*, *Fosb* and *Dusp1*, as well as myeloid differentiation factors such as *Ngp*, *Ltf* and *S100a8*, in geriatric femur HSCs compared with skull HSCs (Extended Data Fig. 12a–f). These differentially expressed genes correlate strongly with gene expression changes reported for IL-1β-treated HSCs or the *Tet2*-knockout model of clonal haematopoiesis[23,48].

To further determine whether the observed differences in BM function between ageing skull and femur are owing to HSC- or HSPC-intrinsic or extrinsic, microenvironmental factors, we performed colony-forming unit (CFU) assays of HSPCs isolated from skull or femur of young or geriatric mice (Extended Data Fig. 13a,b). Although HSPCs derived from both geriatric skull and femur initially showed higher colony-forming potential than their younger counterparts in primary CFU plating, there were no significant differences in secondary plating or in HSPC self-renewal or myeloid differentiation potential in both rounds of plating, indicating a role for niche-derived factors. scRNA-seq analysis of endothelial cells from geriatric BM revealed differences in the expression of inflammatory and myeloid differentiation factors such as *Hotairm1*, *Lgals3*, *Il6* and *S100a6*, which were significantly upregulated in geriatric femur compared with skull (Extended Data Fig. 14a–g). Conversely, endothelial cells from skull exhibited higher expression of HSC maintenance factors such as *Ptn*, *Gpr182* and *Spp1* (Extended Data Fig. 14h).

Finally, to assess the haematopoietic function of skull BM in a physiological context devoid of irradiation injury, we expressed the green-to-red photoconvertible fluorescent protein, Kikume green–red[49] (KikGR) in haematopoietic cells under control of *Vav-cre*[21] (Fig. 5i). Following in vivo photoconversion of haematopoietic cells in calvarial BM, flow cytometric analysis of peripheral blood revealed a marked age-dependent increase in systemic haematopoietic contribution from photoconverted skull BM (Fig. 5j). These results show that systemic haematopoietic output from skull increases relative to other BM compartments during ageing.

In summary, our findings reveal an unexpected level of heterogeneity between two major BM compartments that are well-established model systems in experimental studies. In contrast to nearly all other organ systems and long bone, which become fully vascularized during embryonic and postnatal development[50], blood vessel growth in skull persists throughout adult life. Lifelong angiogenesis in skull involves the continued formation of large-calibre sinusoidal vessels, which are strongly associated with a range of haematopoietic cell populations and thus represent a central landmark in the skull BM environment. Vascular growth and BM expansion in skull rely on VEGFA and its receptor VEGFR2, which are key regulators of developmental and regenerative angiogenesis in many organs[39,51,52]. Consistent with previous studies[40], skull BM-resident HSPCs express more VEGFA than Lin⁺ differentiated haematopoietic cells, but other relevant sources of VEGFA might exist, as has been shown for chondrocytes and osteoprogenitors in long bone[53]. Our experiments also establish that circulation-derived VEGFA can promote calvarial vessel growth and BM expansion, which is consistent with the finding that systemic VEGF can increase lifespan and delay ageing processes in mice[54]. Conversely, HSPC behaviour is controlled by the BM endothelium and associated reticular cells, which provide critical niche signals, such as stem cell factor[55] (also known as SCF or KIT ligand) or CXCL12[56] (also known as SDF-1). Several studies have shown that haematopoietic cell production is enhanced in response to various physiological and pathological stimuli, such as pregnancy and ischaemic stroke[57]. Our present findings indicate that these conditions lead to the expansion of calvarial BM but induce only limited alterations in load-bearing long bone, where BM growth may not be compatible with

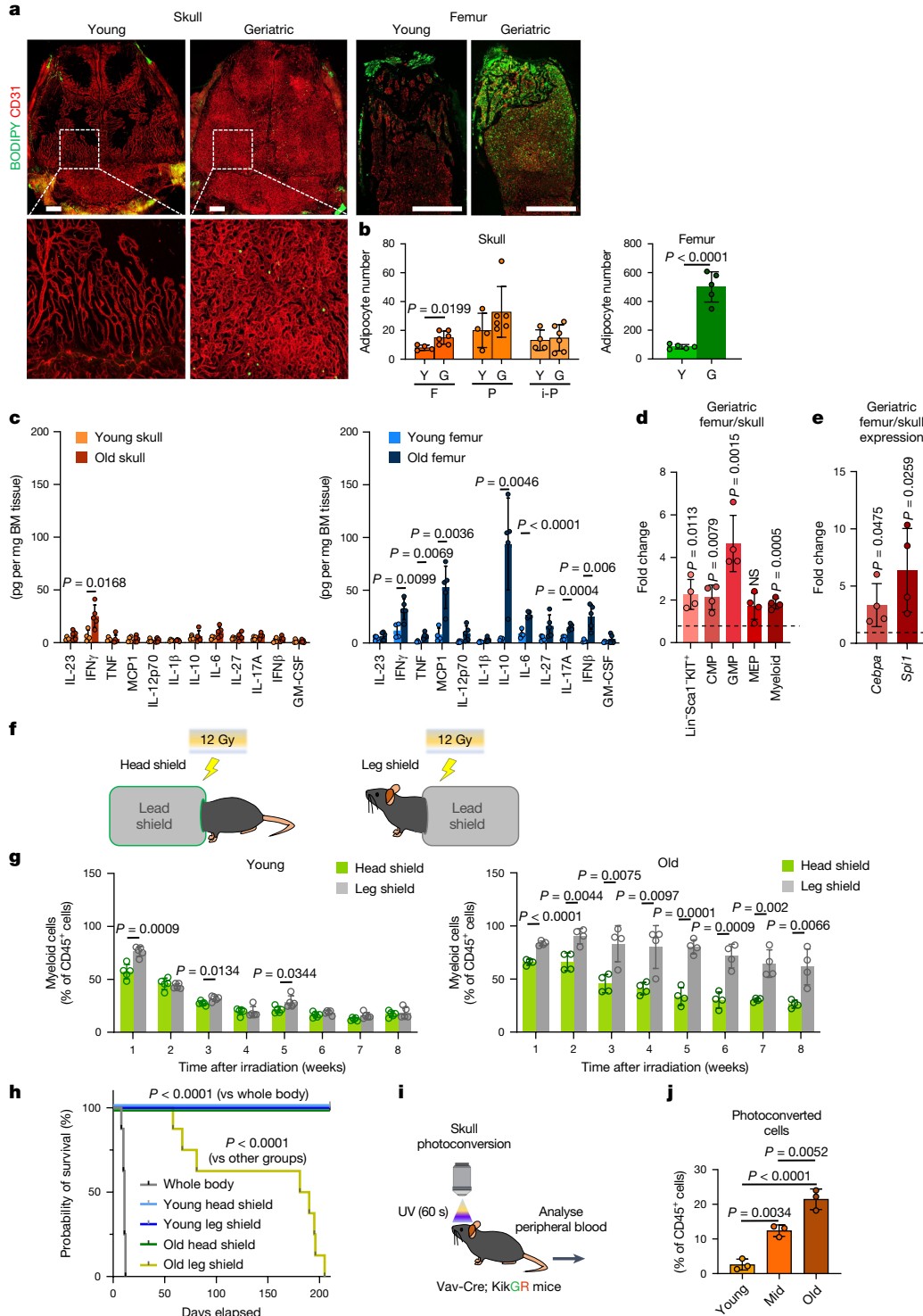

**Fig. 5 | Properties and function of ageing skull marrow. a,b**, Staining (**a**) and quantification (**b**) of adipocytes (BODIPY) and blood vessels (CD31) in skull or femoral BM of young or geriatric mice. Scale bars, 1 mm. *n* = 4 (young skull), *n* = 6 (old skull), *n* = 5 (all other groups) mice per group from 4 independent experiments. F, frontal; P, parietal; i-P, interparietal. **c**, Quantification of inflammatory cytokines by multiplex array analysis of total BM lysates from young or old skull or femur. *n* = 4 (young skull), *n* = 5 (old skull), *n* = 4 (young femur), *n* = 5 (old femur) mice per group from 2 independent experiments. **d,e**, Quantification of myeloid progenitors and progeny by FACS (**d**; difference in percentage of live cells) and RT–qPCR analyses of myeloid determination factors (**e**) of geriatric skull versus femur BM. Lin⁻Sca1⁻KIT⁺, common myeloid progenitor (CMP), granulocyte-monocyte progenitor (GMP), megakaryocyte-erythrocyte progenitor (MEP). Each value is the fold change difference

between geriatric femur sample and corresponding skull sample from the same mouse. *n* = 4 mice per group from 2 independent experiments. **f,g**, Scheme of shielding experiments (**f**) and FACS analysis of CD11b⁺ myeloid cells in peripheral blood (**g**) isolated from mice with head versus leg shielding. *n* = 5 (young) and *n* = 4 (old) mice per group from 3 independent experiments. **h**, Kaplan–Meier survival plot showing survival of mice after whole-body irradiation or shielding skull or hindlimbs. *n* = 8 mice per group from 2 independent experiments. **i**, Schematic showing skull BM photoconversion in *Vav-cre;Rosa26-KikGR* mice. **j**, FACS analysis of photoconverted CD45⁺ haematopoietic cells in peripheral blood. *n* = 3 mice per group from 3 independent experiments. Data are mean ± s.d. *P* values by two-tailed unpaired Student's *t*-test, log-rank test and Tukey multiple comparison test (one-way ANOVA).

mechanical loading and structural integrity. Furthermore, our results establish that calvarial BM is more resistant to ageing-related degenerative processes, namely adipogenesis, the upregulation of inflammatory cytokines and compromised vascular integrity. This might, in part, reflect that calvarial BM is biologically 'younger' owing to its formation during adult life. Alternatively, unknown cellular or molecular features might differentially influence haematopoiesis in different BM compartments. Consistent with the latter, recent work has shown that skull BM has a distinct molecular profile in health and neurological disorders[58]. Notably, our findings also suggest that BM expansion occurs in human skull, which is consistent with previous observations[59,60]. Thus, our findings regarding the plasticity and resilience of the BM in skull may have wider relevance for ageing and disease processes in humans as well as pharmacological treatments that target angiogenic signalling pathways.

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

## Methods

### Animal models

C57BL/6J mice were used for all experiments involving wild-type mice and pharmacological treatments. Mice at the age of 10–14 weeks, 31–37 weeks, 52–75 weeks and >95 weeks were chosen for young adult, middle-aged, old and geriatric groups, respectively. Both female and male mice of all age groups were used for initial BM expansion analyses, while only female mice were used for remaining experiments. *Flk1*-GFP reporter mice[17] were used for initial blood vessel characterization. For genetic labelling of haematopoietic cells, *Vav1-cre* mice[21] were interbred with *ROSA26-mTmG* reporter mice[22] to generate *Vav-mTmG* mice. For photoconversion of haematopoietic cells, *Vav1-cre* mice were interbred with *ROSA26*-CAG-*loxP*-stop-*loxP*-KikGR knock-in mice[49] to generate *Vav-KikGR* mice. For pregnancy experiments, 10-week-old C57BL/6J female mice were paired with 10- to 12-week-old C57BL/6J male mice and the onset of pregnancy was determined by the presence of a vaginal plug in the morning. Ten-week-old mice received daily intraperitoneal injections of PTH (1–34) (Bachem, 0.1 mg kg$^{-1}$ for 28 days), PGE$_2$ (Cayman Chemical, 2 mg kg$^{-1}$ for 7 days), AMD3100 (Abcam, 5 mg kg$^{-1}$ for 14 days) before they were euthanized. For DC101 treatment, 10-week-old mice received intraperitoneal injections of DC101 (BioXCell, 40 mg kg$^{-1}$) every 2 days for 12 weeks.

Mice were kept in individually ventilated cages, with constant access to food and water under a 12 h light and 12 h dark cycle regime. Air flow, temperature (21–22 °C) and humidity (55–60%) were controlled by an air management system. Mice were checked daily and maintained in specific pathogen-free conditions. Sufficient nesting material and environmental enrichment was provided. All animal experiments were performed according to the institutional guidelines and laws, approved by local animal ethical committee and were conducted at the Max Planck Institute for Molecular Biomedicine (84-02.04.2016. A160, 81-02.04.2018.A171, 81-02.04.2020.A212, 81-02.04.2020.A416 and 81-02.04.2022.A198), Universitätsmedizin Berlin (G0220/17), Georg-Speyer-Haus (F123/2017) and the University Medical Center Mainz Institute of Transfusion Medicine (G23-1-067 A1TE) under the indicated permissions granted by the Landesamt für Natur, Umwelt und Verbraucherschutz (LANUV) of North Rhine-Westphalia, the State Office for Health and Social Affairs Berlin, Regierungspräsidium Darmstadt and the Landesuntersuchungsamt Rheinland-Pfalz, Germany.

### Human subjects, CT acquisition and data analysis

The study was approved by the local ethics committee and the institutional review board (IRB) of Asan Medical Center, and the requirement for informed consent was waived due to the retrospective nature of the study (IRB number: 2023-0658). The study population consisted of 36 patients, divided into 4 groups according to age (between 20 and 40 years, over 60 years) and sex (male, female), with 9 individuals in each group. Patients who underwent CT for evaluation of small cerebral aneurysm from April to May 2023 were eligible. Participants were excluded if they had a previous history of surgery or radiation therapy to the head and neck, vascular or bone-related medical implants, or a suspicious disease other than small cerebral aneurysm.

All human patients underwent CT examinations on the same 128-channel multidetector CT system (Somatom Definition Edge; Siemens). Imaging variables were as follows: 100 kV; 100 effective mAs; axial scan mode; section thickness, 0.5 mm; display FOV, 20.5 cm; pitch, 1; gantry rotation time, 0.5 s; pixel matrix, 512 × 512. Images were obtained from the vertex to first cervical spine, without an intravenous injection of contrast media.

The CT data were digitally transferred to a personal computer and processed with ImageJ software (http://rsb.info.nih.gov/ij/). A representative image was selected on a coronal CT image perpendicular to the outermost convex area on an axial CT image. After whole-bone segmentation of the parietal bone, cortical bone and BM were defined by attenuation densities on CT scan: cortical bone as over 850 Hounsfield units and BM as less than 850 Hounsfield units and their areas were calculated.

### Sample processing and immunostaining

Mice were euthanized by transcardial perfusion of PBS and 4% paraformaldehyde (PFA), skulls and femur were collected and fixed immediately in ice-cold 4% PFA for 6–8 h under gentle agitation. Bones were decalcified in 0.5 M EDTA for 3 days (for skulls) or 7 days (for femurs) at 4 °C under gentle shaking agitation, washed 5 times in PBS in 5 min intervals, followed by overnight incubation in cryoprotectant solution (20% sucrose, 2% polyvinylpyrrolidone) and embedding in bone embedding medium (8% gelatin, 20% sucrose, 2% polyvinylpyrrolidone). Samples were stored overnight at −80 °C. 80-μm-thick cryosections were prepared for immunofluorescence staining.

Bone sections were washed in PBS and permeabilized with 0.3% Triton X-100 in PBS for 10 min at room temperature. Samples were incubated in blocking solution (5% heat-inactivated donkey serum in 0.3% Triton X-100) for 1 h at room temperature. Primary antibodies (rat monoclonal anti-endomucin (V.7C7) (Santa Cruz, sc-65495, 1:200 dilution), rabbit monoclonal anti-vATPaseB1/B2 (Abcam, 200839, 1:200 dilution), goat polyclonal anti-osteopontin (R&D Systems, AF808, 1:200 dilution), goat polyclonal anti-CD31 (R&D, AF3628, 1:200 dilution), rabbit polyclonal anti-caveolin-1 (Cell Signaling, 3238, 1:100), goat polyclonal anti-VEGF$_{164}$ (R&D Systems, AF-493-NA, 1:200 dilution), and rat monoclonal APC-conjugated anti-CD117 (KIT) (BD Biosciences, 553356, 1:100 dilution) were diluted in PBS with 5% donkey serum and incubated overnight at 4 °C. Next, slides were washed 3–5 times in PBS in 5 min intervals. Species-specific Alexa Fluor-conjugated secondary antibodies Alexa Fluor 488 (Thermo Fisher Scientific, A21208), Alexa Fluor 594 (Thermo Fisher Scientific, A21209), Alexa Fluor 647 (Thermo Fisher Scientific, A31573 or A21447) diluted 1:500 in PBS with 5% donkey serum were added and incubated overnight at 4 °C. Slides were washed 3–5 times in PBS in 5 min intervals. Nuclei were counterstained with DAPI (Sigma-Aldrich, D9542, 1:1,000 dilution). Coverslips were mounted with FluoroMount-G (Southern Biotech, 0100-01).

### In vivo immunostaining and Evans Blue leakage assay

Rat monoclonal anti-CD31 (BD Biosciences, 553708) was conjugated to Alexa Fluor 647 using the Alexa Fluor 647 Antibody Labeling Kit (Thermo Fisher Scientific, A20186) according to the manufacturer's instructions. For blood vessel immunostaining, the conjugated anti-CD31 antibody and rat monoclonal PE-conjugated anti-endomucin (V.7C7) (Santa Cruz, 65495 PE) were diluted 1:10 in 200 μl PBS and injected intravenously into the tail vein. For haematopoietic cell immunostaining, rat monoclonal FITC-conjugated anti-CD45 (eBioscience, 11-0451-82), hamster monoclonal FITC-conjugated anti-CD3e (eBioscience, 16-0031-82), rat monoclonal PE-conjugated anti-CD45R/B220 (BD Biosciences, 553090), rat monoclonal FITC-conjugated anti-CD11b (BD Biosciences, 553310) were diluted 1:10 in PBS and injected intravenously into the tail vein. Mice were euthanized 1 h after injection with transcardial perfusion with PBS and 4% PFA and bones were collected and fixed immediately in ice-cold 4% PFA for 6–8 h under gentle agitation. The dura mater was carefully removed from the skull with forceps. Bones were decalcified in 0.5 M EDTA for 1 day (for skulls) or 7 days (for femurs) at 4 °C under gentle shaking agitation, and washed 5 times in PBS in 5 min intervals. Skulls were counterstained with DAPI (1:500 dilution) for 1 h and trimmed down to the calvarium before mounting with iSpacers (Sunjin Lab, IS011) in PBS. Femurs were cryosectioned, counterstained and mounted as described above.

For the Evans Blue leakage assay, mice were anaesthetized immediately prior to tail vein injection of 200 μl Evans Blue solution (Sigma-Aldrich, E2129, 1% v/w). Mice were euthanized via transcardial perfusion 5 min after injection as described above. In order to distinguish

vascular leakage in the dura mater from the calvarial BM, dura mater tissues were separated from the calvarial bone before overnight decalcification.

Immunostained samples were imaged with a Zeiss LSM980 (Carl Zeiss). Images were analysed, quantified and processed using ZEN Black (Carl Zeiss, v2.3), ImageJ (NIH, v2.0.0) and IMARIS (Bitplane, v10.0.1). Tilescan overview images of skull BM were superimposed on top of a black background, filling empty corners without image data. Vessel diameter was measured by selecting the $z$-plane image with the widest vessel diameter from the $z$-stack of the individual vessel.

### Scanning electron microscopy

Skull and femur from 12-week-old and 73-week-old mice were isolated and submerged in 4% PFA, 0.5% glutaraldehyde, 2 mM $MgCl_2$, 2 mM $CaCl_2$ in 0.1 M cacodylate buffer, pH 7.4, under agitation for 2 h at room temperature. Samples were fixed further overnight in 2% glutaraldehyde, 2 mM $MgCl_2$, 2 mM $CaCl_2$ in 0.1 M cacodylate buffer, pH 7.4 at 4 °C. Bones were then decalcified over 12 days, changing solution every other day in 5% EDTA in 0.1 M cacodylate buffer, pH 7.4 under rotation at 4 °C. Subsequently, 150 µm sections were generated with a vibratome (VT 1200, Leica). Sections were post-fixed in 1% $OsmO_4$, containing 2.5% PFA–glutaraldehyde mixture buffered with 0.1 M phosphate (pH 7.2) for 5 h and then were placed in graded ethanol for critical-point drying using E3000 (Polaron) critical-point dryer. Critical-point-dried bones were placed on a piece of carbon tape and sputter coated with gold in a SC502 Sputter Coater (Polaron). Specimens were imaged on a Quanta 250 Field Emission Scanning Electron microscope (FEI Quanta 250 FEG, FEI, Hillsboro, OR) installed at the Korea Research Institute of Bioscience and Biotechnology.

### Dura mater whole-mount immunostaining

Mice were euthanized by transcardial perfusion of PBS and 4% PFA, skulls were collected and fixed immediately in ice-cold 4% PFA for 6–8 h under gentle agitation. Skulls were decalcified in 0.5 M EDTA for 24 h at 4 °C under gentle shaking agitation, washed 5 times in PBS in 5 min intervals, trimmed down to the calvarium, and incubated in blocking solution (5% heat-inactivated donkey serum in 0.3% Triton X-100) for 1 h at room temperature. Goat polyclonal anti-CD31 (R&D, AF3628, 1:100 dilution) was diluted in PBS with 5% donkey serum and incubated overnight at 4 °C with gentle agitation. Samples were washed 3–5 times in PBS in 10 min intervals. Alexa Fluor 647 (Thermo Fisher Scientific, A21447) diluted 1:500 in PBS with 5% donkey serum was added and incubated overnight at 4 °C with gentle agitation. Samples were washed 3–5 times in PBS in 10 min intervals and mounted with iSpacers (Sunjin Lab, IS011) in PBS.

### tMCAO

Sixteen-week-old female C57BL6/J mice were used throughout the experiments. Mice were anaesthetized by intraperitoneal injection of a mixture of 10 mg kg$^{-1}$ xylazine (cp-pharma) and 90 mg kg$^{-1}$ ketamine hydrochloride (cp-pharma). Throughout the whole procedure and during recovery, body temperature was maintained at 37 °C via a heating pad. After ligation of the left proximal common carotid artery and external carotid artery, a 7.0-nylon monofilament (Doccol) with a 0.23-mm coated tip was introduced into the distal internal carotid artery via an incision in the ligated common carotid artery. The monofilament was advanced distal to the carotid bifurcation to occlude the middle cerebral artery. Arter topical application of the local anaesthetic lidocaine hydrochloride (Xylocain Spray 2%, Aspen) the neck wound was closed temporarily for a 45 min ischaemic period. At reperfusion, the monofilament was withdrawn from the carotid artery and the wound was stitched with 4-0 non-resorbable sutures (Ethibond Excel, Ethicon) and the single s.c. injection of Penicillin G 20 000 U (Benzylpenicillin-Natrium, InfectoPharm) was given. The mouse was returned to its cage to recover under observation.

### Chronic myeloid leukaemia

Six-week-old female C57BL/6 mice were purchased from Charles River Laboratories and were used as donors and recipients in all transplants. The transplantation experiments were performed as previously described[61]. In brief, to induce CML-like myeloproliferative neoplasia, donor BM cells from donor mice pre-treated with 5-fluorouracil (200 mg kg$^{-1}$ intravenously; 4 days prior to collection) were pre-stimulated overnight in medium containing SCF (50 ng ml$^{-1}$), IL-6 (10 ng ml$^{-1}$) and IL-3 (6 ng ml$^{-1}$) and transduced on two consecutive days with murine stem cell virus (MSCV)-IRES-GFP-BCR-ABL1 to induce CML or MSCV-IRES-GFP control virus. Subsequently, transduced cells were intravenously transplanted ($2.5 \times 10^5$ cells per mouse) into sublethally irradiated (900 cGy) recipient mice. Mice were euthanized 14 days after transplantation.

### Lineage depletion and transplantation

In order to transplant lineage-negative BM cells, young (10- to 14-week-old) or old (52- to 75-week-old) donor mice were euthanized and skulls were collected. The calvarium was first chopped with scissors in FACS buffer (PBS with 2% fetal calf serum), then crushed with a mortar and pestle. Cell suspension was filtered through a 40-µm mesh filter (Falcon, 352340), resuspended in RBC lysing buffer (Sigma-Aldrich, R7757) for red blood cell lysis and washed with FACS buffer. Cells were resuspended in FACS buffer and incubated with a biotinylated anti-haematopoietic lineage antibody cocktail (Miltenyi-Biotec, 130-092-613, 1:10 dilution), followed by washing with FACS buffer and incubation with R-PE-conjugated streptavidin secondary antibody (Invitrogen, S866, 1:50 dilution). DAPI (1:1,000 dilution) was added to resuspended cells to distinguish live and dead cells and were FACS-sorted for live lineage-negative cells on a FACSAria Fusion (BD Biosciences). Sorted cells were intravenously transplanted ($5 \times 10^5$ cells/mouse) into lethally irradiated (12 Gy, Best Theratronics, Gammacell 40 Exactor) recipient mice (12-week-old). Mice were euthanized 14 days after transplantation.

### FACS analysis of BM and peripheral blood

Mice from each age group were euthanized and skull and femur were collected. Skulls were chopped with scissors in FACS buffer before crushed with mortar and pestle; femurs were crushed without chopping. BM stromal samples were dissociated with Collagenase I (Gibco, 17100-017, 2 mg ml$^{-1}$) and Collagenase IV (Gibco, 17104-019, 2 mg ml$^{-1}$) in PBS for 20 min at 37 °C with intermittent shaking. Cell suspensions were strained through a 40-µm mesh filter, resuspended in RBC lysing buffer (when applicable) and washed with FACS buffer. Cells were resuspended and incubated with the following primary antibodies in FACS buffer: biotinylated rat monoclonal anti-haematopoietic lineage antibody cocktail (Miltenyi-Biotec, 130-092-613, 1:50 dilution), APC-conjugated rat monoclonal anti-CD117 (BD Biosciences, 553356, 1:100 dilution), FITC-conjugated rat monoclonal anti-CD117 (Biolegend, 105806, 1:100 dilution), FITC-conjugated rat monoclonal anti-Ly6A/E (Sca1) (eBioscience, 11-5981-85, 1:100 dilution), PerCP-Cy5.5-conjugated rat monoclonal anti-Ly-6A/E (Invitrogen, 45-5981, 1:100), APC-Cy7-conjugated hamster monoclonal anti-CD48 (BD Biosciences, 561242, 1:100 dilution), PE-conjugated rat monoclonal anti-CD150 (SLAM) (Biolegend, 115904, 1:100 dilution), Alexa Fluor 647-conjugated rat monoclonal anti-CD150 (Biolegend, 115918, 1:100 dilution), PE-Cy7-conjugated rat monoclonal anti-CD45 (eBioscience, 25-0451-82, 1:100 dilution), BV421-conjugated rat monoclonal anti-TER-119 (Biolegend, 116234, 1:100 dilution), FITC-conjugated rat monoclonal anti-CD71 (Biolegend, 113806, 1:100 dilution), Alexa Fluor 647-conjugated rat monoclonal anti-CD31 (BD Biosciences, 553708, conjugation described above, 1:200 dilution), PE-conjugated rat monoclonal anti-endomucin (Santa Cruz, 65495 PE, 1:100 dilution), PE-Cy7-conjugated rat monoclonal anti-CD16/32 (eBioscience,

25-0161, 1:100 dilution), eFluor 450-conjugated rat monoclonal anti-CD34 (eBioscience, 48-0341, 1:100 dilution), PE-conjugated rat monoclonal anti-CD127 (eBioscience, 12-1271, 1:100 dilution), BV711-conjugated rat monoclonal anti-CD41 (BD Biosciences, 740712, 1:100 dilution), PE-conjugated rat monoclonal anti-CD105 (eBioscience, 12-1051-82, 1:100 dilution), APC-conjugated hamster monoclonal anti-CD3e (eBioscience, 17-0031, 1:100 dilution), PE-conjugated rat monoclonal anti-CD45R/B220 (BD Biosciences, 553090, 1:100 dilution), FITC-conjugated rat monoclonal anti-CD11b (BD Biosciences, 553310, 1:100 dilution). Cells were washed, resuspended in FACS buffer with Alexa Fluor 405-conjugated (Invitrogen, S32351, 1:100 dilution) or APC-Cy7-conjugated (BD Biosciences, 554063, 1:100 dilution) streptavidin secondary antibody, washed again before analysis with a FACSymphony A5 Cell Analyzer (BD Biosciences).

Peripheral blood was collected from the submandibular vein with lancets (Medipoint) into EDTA-coated tubes. Blood was resuspended in RBC lysing buffer and washed with FACS buffer. Cells were resuspended and incubated with the following primary antibodies in FACS buffer: biotinylated rat monoclonal anti-haematopoietic lineage antibody cocktail (Miltenyi-Biotgec, 130-092-613, 1:50 dilution), APC-conjugated rat monoclonal anti-CD117 (BD Biosciences, 553356, 1:100 dilution), FITC-conjugated rat monoclonal anti-Ly6A/E (Sca1) (eBioscience, 11-5981-85, 1:100 dilution), Pacific Blue-conjugated mouse monoclonal anti-CD45.2 (Biolegend, 109820, 1:100 dilution), APC-conjugated hamster monoclonal anti-CD3e (eBioscience, 17-0031, 1:100 dilution), PE-conjugated rat monoclonal anti-CD45R/B220 (BD Biosciences, 553090, 1:100 dilution), FITC-conjugated rat monoclonal anti-CD11b (BD Biosciences, 553310, 1:100 dilution). Cells were washed and resuspended in FACS buffer before analysis with a FACSymphony A5 Cell Analyzer (BD Biosciences).

## RNA extraction and quantitative PCR

FACS-sorted cells from 10-week-old mouse skulls were lysed and RNA was extracted using a Monarch Total RNA Miniprep Kit (New England BioLabs, T2010S). Extracted RNA concentration was measured with a NanoDrop 8000 Spectrophotometer (Thermo Fisher Scientific) and cDNA was generated with a LuncaScript RT SuperMix Kit (New England BioLabs, E3010L). Quantitative PCR with reverse transcription was performed with a BioRad CFX96 real-time PCR system using FAM-conjugated Taqman probes for *Vegfa* (Mm00437306_m1) or using PowerUp SYBR Green Master Mix (Applied Biosystems, A25742) with primers designed using Pimer-BLAST or adopted from previously published studies: *Vegfa$_{120}$* (5′-AACGATGAAGCCCTGGAGTG-3′; 5′-TGAGAGGTCTGGTTCCCGA-3′); *Vegfa$_{164}$* (5′-AACGATGAAGCCCTG GAGTG; 5′-GACAAACAAATGCTTTCTCCG-3′); *Vegfa$_{188}$* (5′-AACGATG AAGCCCTGGAGTG-3′; 5′-AACAAGGCTCACAGTGAACG-3′). Gene expression levels were normalized to the endogenous VIC-conjugated *Gapdh* probe (44326317E) as control.

## ELISA

Mice from each age group were euthanized, bones were collected. Skulls were chopped before being crushed with a mortar and pestle in ice-cold RIPA lysis buffer; femurs were crushed without chopping. Supernatants of centrifuged lysates were further concentrated using an Ultra-0.5 Centrifugal Filter Unit with a 3 kDa cutoff (Millipore, UFC500396), resulting concentrations were measured using a Pierce BCA Protein Assay Kit (Thermo Fisher Scientific, 23225), and the concentrations of VEGFA in tissue extracts were measured using a Mouse VEGFA Quantikine ELISA Kit (R&D Systems, MMV00-1).

## Hypoxia analysis

Hypoxic cells were detected with the hypoxia probe pimonidazole (Pimo, Hypoxyprobe) according to the manufacturer's instructions. Mice were intraperitoneally injected with 60 mg kg$^{-1}$ 1 h before analysis.

## VEGFA plasmid construction and overexpression

To generate the pLIVE-VEGFA$_{165}$-HA-MP-Asp$_{8x}$ bone-homing protein containing VEGF$_{165}$ fused to a HA tag, metalloprotease and 8x Asp peptide sequences, a cDNA fragment encoding amino acids 1–191 of human *VEGFA* was amplified via PCR using the following oligonucleotide primers: VEGFA-AscI-Fwd: 5′-ATGAACTTTCTGCTGTCT-3′ and VEGFA-XhoI-Rev: 5′-CCGCCTCGGCTTGTCACATCTGCA-3′ and annealed with the NEBuilder Assembly Cloning Kit.

Ten-week-old mice were used for hydrodynamic tail vein injection. Mice were injected with 0.5 μg g$^{-1}$ (plasmid/body weight) pLIVE-Vegfa plasmid suspended in TransIT-EE hydrodynamic delivery solution (Mirus, MIR5340). The appropriate amount of plasmid was suspended in an injection volume of 10% of the body weight and injected into each individual mouse via the tail vein in 5–7 s as previously reported[62].

## Adipocyte analysis

To stain for neutral lipids, the entire calvarium or femur cryosections were incubated in BODIPY 493/503 (Invitrogen, D3922; 1:1,000 dilution) for 1 h at room temperature with gentle agitation (only calvarium). Samples were washed with PBS 3–5 times at 5 min intervals before mounting.

## Analysis of inflammatory cytokines

Mice from each age group were euthanized and bones were collected. Skulls were chopped before being crushed with a mortar and pestle in ice-cold RIPA lysis buffer; femurs were crushed without chopping. Supernatants of centrifuged lysates were further concentrated using an Ultra-0.5 Centrifugal Filter Unit with a 3 kDa cutoff (Millipore, UFC500396), resulting concentrations were measured using a Pierce BCA Protein Assay Kit (Thermo Fisher Scientific, 23225), and concentrations of inflammatory cyotokines were measured with LEGENDplex Mouse Inflammation Panel (13-plex) with V-bottom plates (Biolegend, 740446). Analysis on a FACSymphony (BD Biosciences) and quantification were performed according to the manufacturer's protocol. Data analysis was performed using software provided by Biolegend. Manual gating was used to define beads A and B, and automatic gating strategy was used to gate individual cytokines in the APC–PE plot.

## Irradiation with partial shielding

Mice were anaesthetized with ketamine (100 mg kg$^{-1}$) and xylazine (10 mg kg$^{-1}$) prior to irradiation. For partial shielding, the entire head or both legs of a mouse were inserted into the opening of the cylindrical 1-inch-thick lead shield (JRT Associates, PTI-50-P) and exposed to lethal irradiation (12 Gy). The mouse was returned to its cage to recover under observation.

## Sample preparation for scRNA-seq

Mice from each age group were euthanized and skull and femur were collected. Skulls were chopped with scissors in FACS buffer before crushed with mortar and pestle; femurs were crushed without chopping. BM stromal samples were dissociated with Collagenase I (Gibco, 17100-017, 2 mg ml$^{-1}$) and Collagenase IV (Gibco, 17104-019, 2 mg ml$^{-1}$) in PBS for 20 min at 37 °C with intermittent shaking. Cell suspensions were strained through a 40-μm mesh filter and washed with FACS buffer. Cells were resuspended and incubated with biotinylated rat monoclonal anti-haematopoietic lineage antibody cocktail (Miltenyi-Biotec, 130-092-613, 1:50 dilution). Cells were washed, resuspended in FACS buffer with mouse monoclonal anti-Biotin MicroBeads (Miltenyi-Biotec, 130-105-637, 1:50 dilution) and incubated before being loaded into a magnetic-associated cell sorting (MACS) column (Miltenyi-Biotec, 130-042-201) for lineage depletion. Lin$^-$ cells were further incubated with rat monoclonal anti-CD45 MicroBeads (Miltenyi-Biotec, 130-052-301, 1:50 dilution), rat monoclonal anti-CD117 MicroBeads (Miltenyi-Biotec, 130-091-224, 1:50 dilution), biotinylated rat monoclonal anti-CD71 (Biolegend, 113803, 1:100). Cells were washed, resuspended in FACS

buffer with mouse monoclonal anti-Biotin MicroBeads (Miltenyi-Biotec, 130-105-637, 1:50 dilution) and incubated before being loaded into a magnetic-associated cell sorting (MACS) column (Miltenyi-Biotec, 130-042-201) for further haematopoietic depletion. Single-cell suspensions were processed with BD Rhapsody and scRNA-seq libraries were evaluated and quantified by Agilent Bioanalyzer using High Sensitivity DNA Kit (Agilent Technologies, 5067-4626) and Qubit (Thermo Fisher Scientific, Q32851). Individual libraries were diluted to 4 nM and pooled for sequencing. Pooled libraries were sequenced by using High Output Kit (Illumina, TG-160-2002) with a NextSeq500 sequencer (Illumina).

### scRNA-seq

Preprocessing: STAR version 2.7.10a (PMID: 23104886) was used to generate a reference genome index for GRCm39, with Gencode annotations vM29, subset to lncRNA and protein-coding genes.

FASTQ reads were mapped against the reference genome index using STAR with the settings "--soloType CB_UMI_Complex --soloCellFilter None --outSAMtype BAM SortedByCoordinate --soloFeatures GeneFull_Ex50pAS --soloCBmatchWLtype 1MM --soloUMIlen 8 --soloCBwhitelist BD_CLS1.txt BD_CLS2.txt BD_CLS3.txt --runRNGseed 1 --soloMultiMappers EM --readFilesCommand zcat --outSAMattributes NH HI AS nM NM MD jM jI MC ch CB UB GX GN sS CR CY UR UY". Libraries using standard BD Rhapsody beads were mapped using the adapter parameters "--soloAdapterSequence NNNNNNNNNACTGG CCTGCGANNNNNNNNNNGGTAGCGGTGACA --soloCBposition 2_0_2_8 2_21_2_29 3_1_3_9 --soloUMIposition 3_10_3_17", libraries with BD Rhapsody enhanced beads with --soloAdapterSequence NNNNNNNNNNG TGANNNNNNNNNNGACA --soloCBposition 2_0_2_8 2_13_2_21 3_1_3_9 --soloUMIposition 3_10_3_17.

Raw counts were imported as AnnData[63] objects. We removed low complexity barcodes with the knee plot method, and further filtered out cells with a mitochondrial mRNA content, as well as unusually high total and gene counts using manually determined cutoffs for each sample. Doublets were scored with scrublet[64]. Finally, each sample's gene expression matrix was normalized using scran[65] (1.22.1) with Leiden clustering[66] input at resolution 0.5.

G2M and S phase scores were assigned to each cell using gene lists from ref. 67 and the scanpy[68] (1.9.6) sc.tl.score_genes_cell_cycle function.

Embedding, clustering and annotation: different combinations of samples and cell populations (all, ECs, HSCs), were used as input for 2D embedding and clustering: the corresponding expression matrix was subset to the 2,000 most highly variable genes (sc.pp.highly_variable_genes, flavour "seurat"). The top 50 principal components were calculated, and batch-corrected using Harmony[69] (0.0.9). The principal components served as basis for $k$-nearest neighbour calculation (sc.pp.neighbors, n_neighbors=30), which were used as input for UMAP[70] layout (sc.tl.umap, min_dist=0.3). Cell populations were clustered using scanpy.tl.leiden, and a suitable resolution was chosen for a first-pass annotation. Here, contaminating cell populations, including multiplet clusters, were removed, and clustering was repeated. Cluster marker genes were calculated using a pseudobulk approach, comparing aggregate counts with 2 pseudoreplicates for each cluster to all remaining cells (pyDeSEQ2 0.4.8). Finally, expression of select marker genes was plotted using Matplotlib[71] (3.8.4) imshow, and clusters were annotated accordingly.

Differential expression analysis: Differentially expressed genes were calculated using a pseudobulk approach, comparing aggregate counts with two pseudoreplicates for each condition (pyDeSEQ2 0.4.8).

### Skull BM photoconversion

*Vav1-KikGR* mice were anaesthetized with ketamine (100 mg kg⁻¹) and xylazine (10 mg kg⁻¹). A skin flap was generated to expose the calvarium, as previously described[72]. Each exposed area of the calvarium was then exposed to UV light from a Zeiss Axio Imager (Zeiss Microscopy) for 60 s, confirmed for photoconversion from green to red fluorescence, before exposing another area. The skin flap was sutured back together and peripheral blood was analysed by flow cytometry, as described above, to check for the presence of photoconverted cells, which were non-existent in the peripheral blood immediately after photoconversion. One week after photoconversion, peripheral blood was drawn, stained for Alexa Fluor-conjugated rat monoclonal anti-CD45, and was analysed by flow cytometry for CD45⁺ photoconverted haematopoietic cells derived from the skull BM.

### Statistical analysis

No statistical methods were used to predetermine sample size. The experiments were randomized and investigators were blinded to allocation during experiments and outcome analyses. All values are presented as mean ± s.d. Statistical significance was determined by the two-tailed unpaired Student's $t$-test between two groups or the Tukey multiple comparison test (one-way ANOVA) for multiple-group comparison. Statistical analyses were performed using GraphPad Prism 9.0 (GraphPad Software). Statistical significance was set at $P < 0.05$.

### Reporting summary

Further information on research design is available in the Nature Portfolio Reporting Summary linked to this article.

### Data availability

The scRNA-seq data generated in this study have been deposited in the Gene Expression Omnibus under accession number GSE275179. The mouse reference genome GRCm39 with GENCODE M26 annotation (https://www.gencodegenes.org/mouse/release_M26.html) was used for mapping the reads in this study. All individual mouse lines used in this study are commercially available at The Jackson Laboratory. Plasmid constructs are available, upon request, through the corresponding author. Source data are provided with this paper.

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

**Acknowledgements** The authors thank M. Stehling (MPI Münster), J. Kirsch (FCCF, DRFZ) and C. Conche (HI-TRON Mainz, DKFZ) for FACS technical assistance; F. Berkenfeld for supporting animal colony maintenance; and K. Müller for sequencing technical assistance. V.M. is a part of the Cells in Motion-International Max Planck Research School (CiM-IMPRS) and University of Münster, Germany. R.H.A. is supported by the Max Planck Society, the European Research Council (AdG 101139772, PROTECT), the DFG (CRC 1366, project no. 394046768), and the Leducq Foundation.

**Author contributions** B.I.K. and R.H.A. designed the study and interpreted results. B.I.K. designed, conducted and analysed the majority of the experiments, including all pharmacological treatments, skull sample processing, immunostainings, transplantations, FACS sorting and analyses, RNA and protein analyses, and photoconversion surgeries. V.M. processed femur samples and provided technical assistance. H.-W.J. and H.P. provided technical assistance. K.K. performed scRNA-seq analyses. Y.J.C. conducted and analysed human head CT scans. M.N.-K. and P.V. designed and performed transient middle cerebral

artery occlusion surgeries. R.K., R.S.P. and D.S.K. generated the CML mouse model and contributed to project design. S.A. generated pLIVE plasmids. H.J.L. conducted scanning electron microscopy sample processing and imaging. M.G.B. contributed to project design and data interpretation. R.H.A. supervised the project.

**Funding** Open access funding provided by Max Planck Society.

**Competing interests** The authors declare no competing interests.

**Additional information**
**Correspondence and requests for materials** should be addressed to Bong Ihn Koh or Ralf H. Adams.

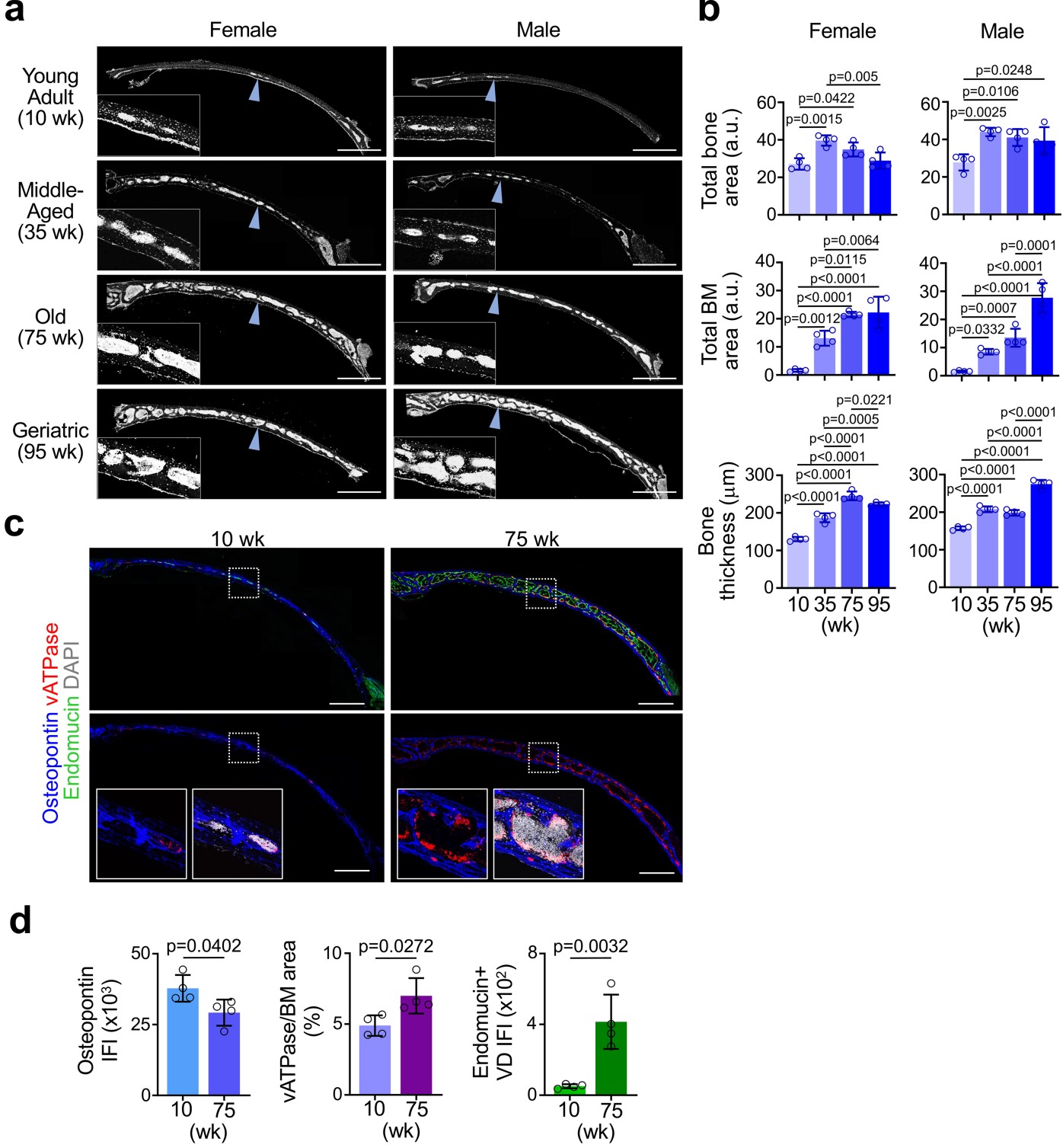

**Extended Data Fig. 1 | Age-dependent expansion of BM in skull. a, b.** DAPI staining and quantification of mouse skull coronal cryosections showing expansion of BM cellular content during adulthood and aging (n = 4 mice/group from three independent experiments). Blue arrowheads indicate location of magnified inset. **c, d**. IF staining and quantification of skull coronal cryosections showing increased vATPase[+] activated osteoclasts attached to Osteopontin[+] bone surfaces in old versus young specimen (n = 4 mice/group from two independent experiments). Scale bars, 1 mm. Vertical bars indicate mean ± SD. *P* values were calculated using Tukey multiple comparison test (one-way ANOVA) and two-tailed unpaired Student's *t*-test.

## a

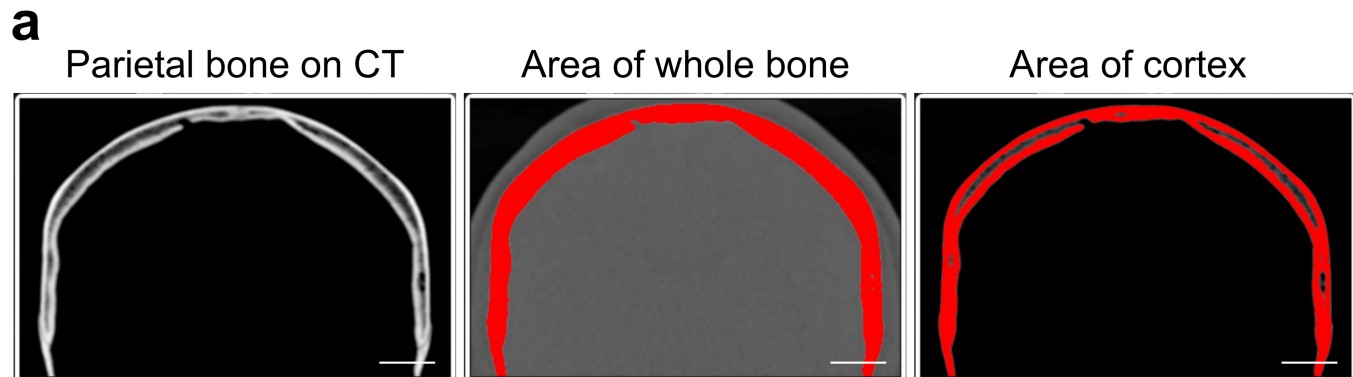

| Parietal bone on CT | Area of whole bone | Area of cortex |

*Area of whole bone – Area of cortex = Area of bone marrow

## b

|  | 38 yr | 68 yr |
| Female | | |

|  | 37 yr | 69 yr |
| Male | | |

## c

**Extended Data Fig. 2 | Age-related expansion of diploic space in human skull. a**. Method for quantification of whole bone, cortex and bone marrow areas in human patient CT scans. **b, c**. Representative coronal CT images (**b**) and quantification (**c**) of young (21–40 years old) and old (61–69 years old) female and male human skulls showing enlargement of diploic space with aging (n = 9 patients/group). Scale bars, 2 cm. Vertical bars indicate mean ± SD. *P* values were calculated using two-tailed unpaired Student's *t*-test.

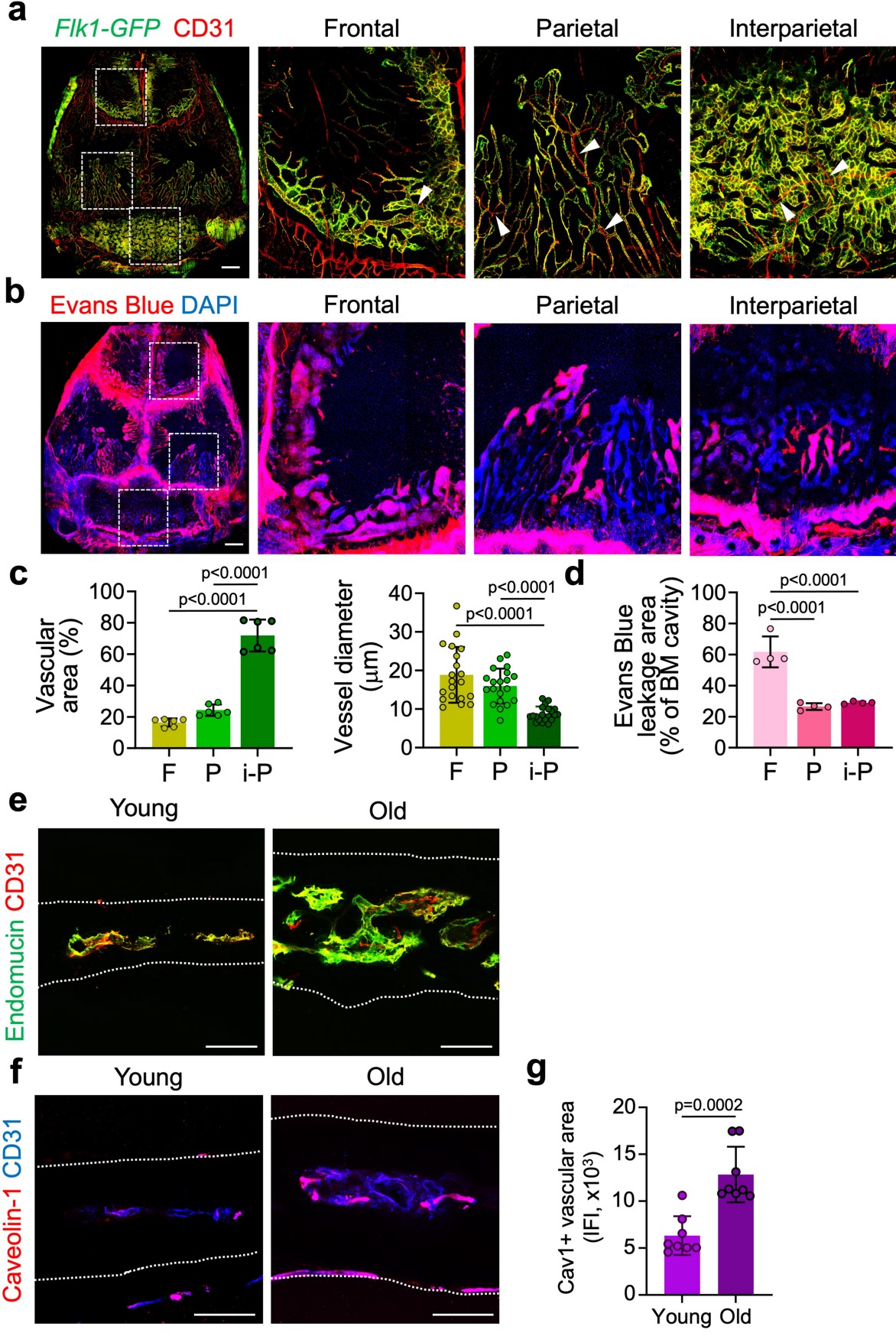

**Extended Data Fig. 3** | See next page for caption.

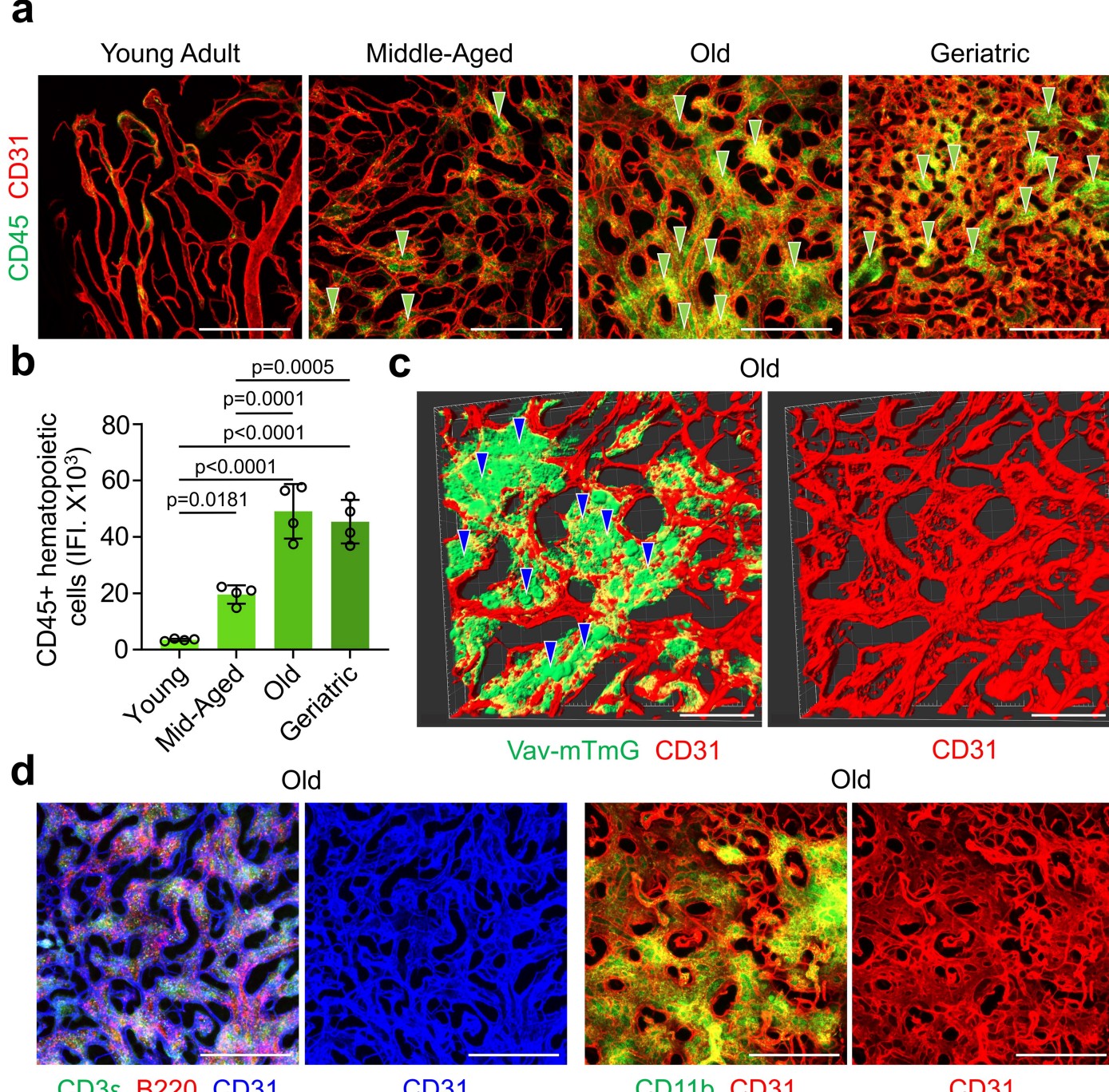

**a**

| Young Adult | Middle-Aged | Old | Geriatric |

CD45 CD31

**b**

CD45+ hematopoietic cells (IFI. X10³)

p=0.0005
p=0.0001
p<0.0001
p<0.0001
p=0.0181

Young  Mid-Aged  Old  Geriatric

**c**

Old

Vav-mTmG CD31          CD31

**d**

Old                                Old

CD3ε B220 CD31      CD31          CD11b CD31      CD31

**Extended Data Fig. 4 | Association of vessels and hematopoietic cells in skull BM. a, b.** In vivo IF staining (**a**) and quantification (**b**) of CD45⁺ hematopoietic cells in skull BM during aging (n = 4 mice/group from three independent experiments). Green arrowheads indicate dense clusters of CD45⁺ cells. Scale bars, 500 μm. **c.** Genetic labeling of hematopoietic cells in *Vav-Cre Rosa26-mTmG* mice (Vav-mTmG) shows association (arrowheads) of GFP-expressing cells with large CD31⁺ vessels. Representative images from three independent experiments. Scale bars, 150 μm. **d.** Confocal images showing association of CD3ε⁺ T lymphocytes (green), B220⁺ B lymphocytes (red) and CD11b⁺ cells (green) with large-caliber CD31⁺ BM vessel in aged skull. Representative images from three independent experiments. Vertical bars indicate mean ± SD. *P* values were calculated using Tukey multiple comparison test (one-way ANOVA). Scale bars, 500 μm.

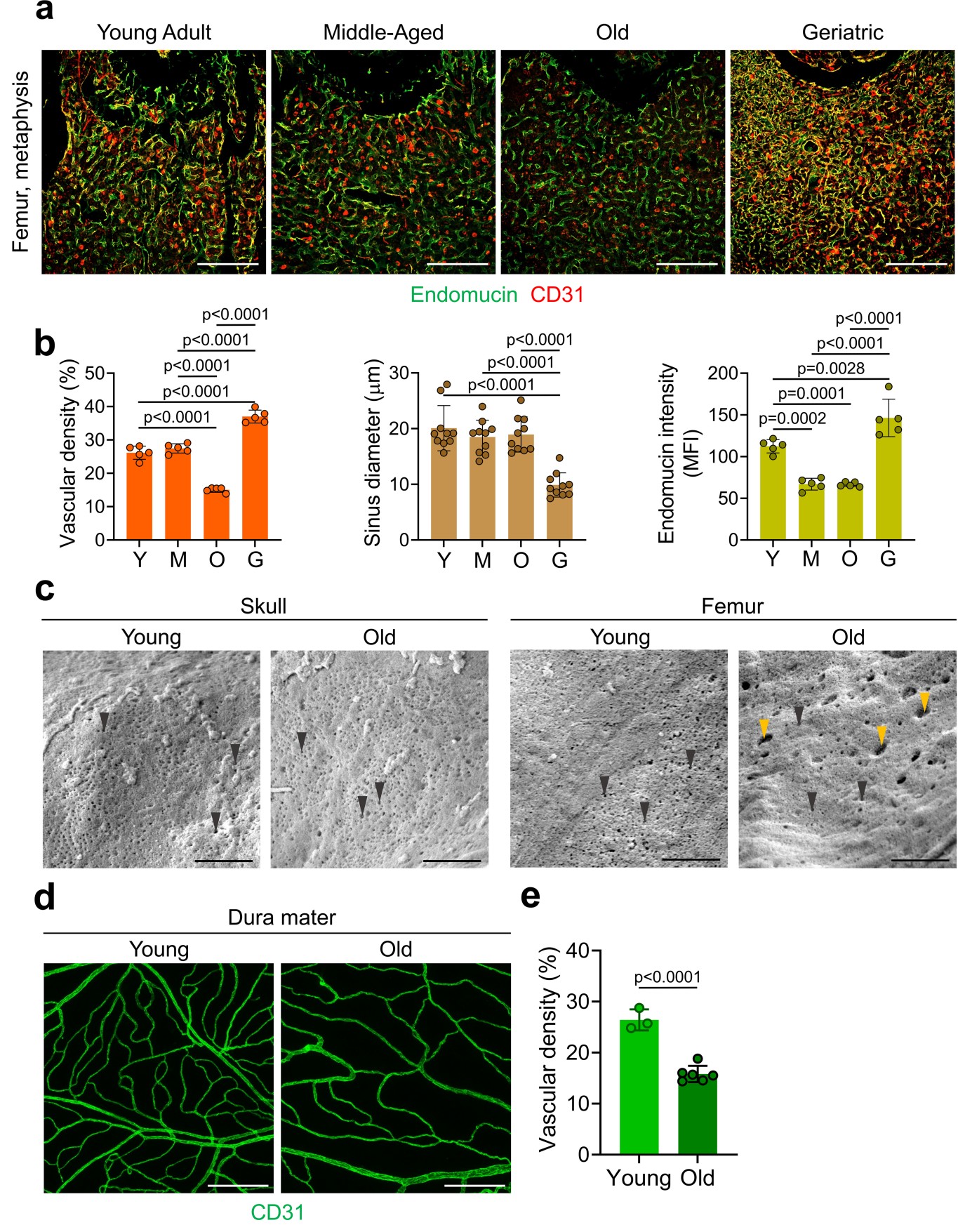

**Extended Data Fig. 5 |** See next page for caption.

**Extended Data Fig. 5 | Age-related changes in the BM vasculature.**
**a, b**. In vivo IF staining (**a**) and quantification (**b**) of femoral BM vessels showing vascular deterioration during aging by comparing young (Y), middle-aged (M), old (O), and geriatric (G) specimen (n = 5 mice/group for vascular density and Endomucin intensity, n = 10 randomly selected vessels from all samples/ group from three independent experiments). Scale bars, 500 μm. **c**. Electron micrographs showing the surface of BM endothelial cells in young vs. old skull or femur, as indicated. Note regular pattern of fenestrations (black arrowheads) in young and old skull but disorganized pattern with larger gaps (yellow arrowheads) in old femur. Representative images from three independent experiments. Scale bars, 1 μm. **d, e**. IF staining (**d**) and quantification (**e**) of dural blood vessels revealing substantial decrease in vascular density with aging. n = 3 (young), n = 6 (old) mice/group from three independent experiments. Scale bars, 250 μm. Vertical bars indicate mean ± SD. *P* values were calculated using Tukey multiple comparison test (one-way ANOVA) and two-tailed unpaired Student's *t*-test.

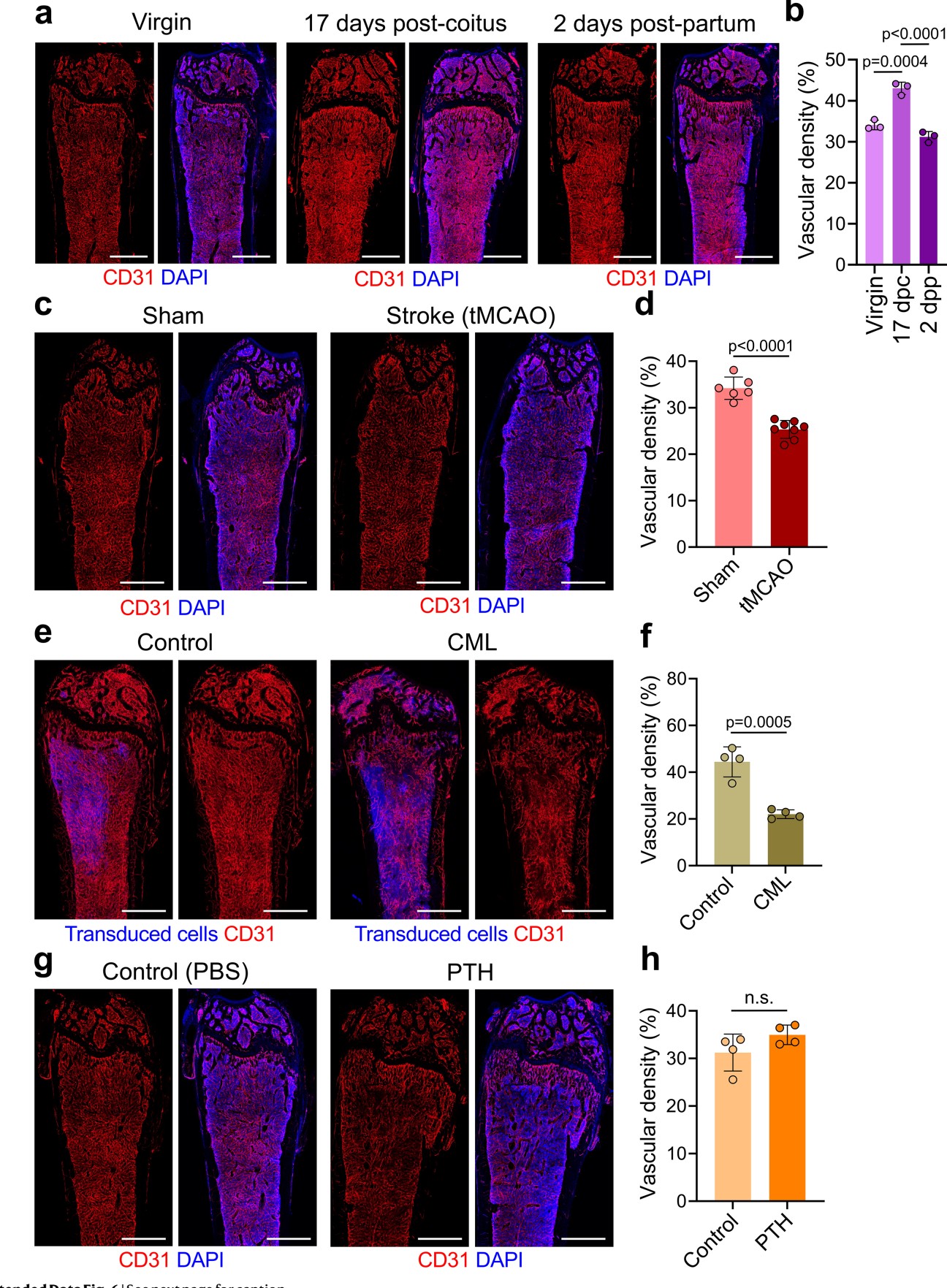

**Extended Data Fig. 6** | See next page for caption.

**Extended Data Fig. 6 | Pathophysiological changes in the femoral BM vasculature. a, b.** In vivo IF staining (**a**) and quantification (**b**) of femoral BM blood vessels in pregnant (17 dpc) and post-partum (2 dpp) female mice. n = 3 mice/group from three independent experiments. **c, d.** In vivo IF staining (**c**) and quantification (**d**) of femoral BM blood vessels in mice 7 days after transient mid-cerebral artery occlusion (tMCAO). n = 6 (sham), n = 8 (tMCAO) from three independent experiments. **e, f.** In vivo IF staining (**e**) and quantification (**f**) of femoral BM blood vessels in mice with chronic myeloid leukemia. n = 4 mice/group from three independent experiments. **g, h.** In vivo IF staining (**g**) and quantification (**h**) of femoral BM blood vessels in mice with 28-day sustained parathyroid hormone (PTH) treatment. n = 4 mice/group from three independent experiments. Scale bars, 1 mm. Vertical bars indicate mean ± SD. *P* values were calculated using Tukey multiple comparison test (one-way ANOVA) and two-tailed unpaired Student's *t*-test.

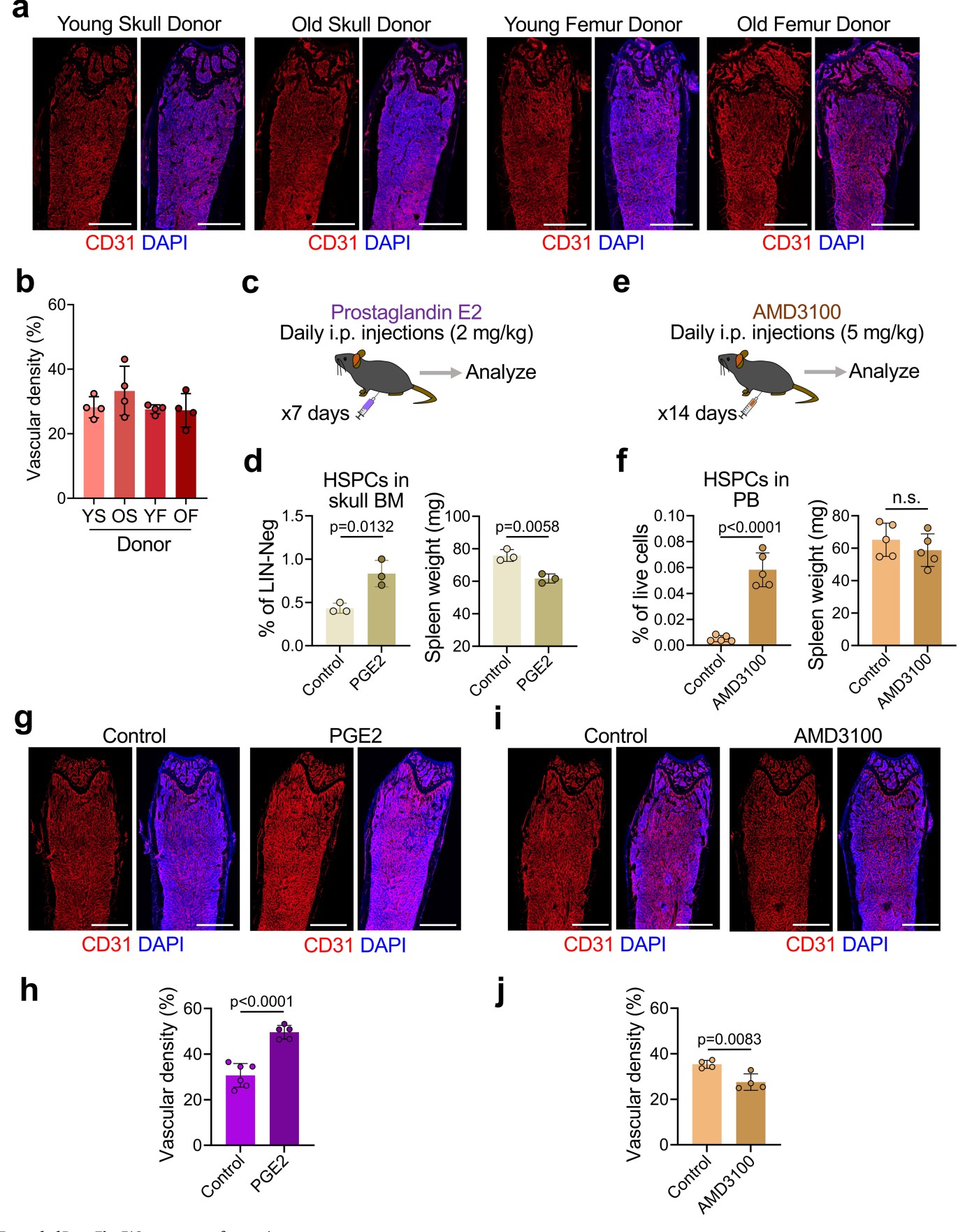

**Extended Data Fig. 7** | See next page for caption.

**Extended Data Fig. 7 | Effect of treatments affecting HSPCs. a, b**. In vivo IF staining (**a**) and quantification (**b**) of femoral BM blood vessels in recipient mice receiving hematopoietic lineage-negative cells isolated from young or old donors (young skull (YS), old skull (OS), young femur (YF), old femur (OF)). n = 4 mice/group from three independent experiments. Scale bars, 1 mm. **c**, **d**. Diagram showing experimental scheme for PGE2 treatment for HSPC expansion (**c**) and quantification (**d**) of HSPCs in skull BM and spleen weight with PGE2 treatment (n = 3 mice/group from three independent experiments). **e**, **f**. Diagram showing experimental scheme for AMD3100 treatment for HSPC mobilization (**e**) and quantification (**f**) of HSPCs in peripheral blood and spleen weight with AMD3100 treatment (n = 5 mice/group from three independent experiments). **g-j**. In vivo IF staining (**g, i**) and quantification (**h, j**) of femoral BM blood vessels in mice treated with PGE2 (**g, h**) or AMD3100 (**i, j**) for HSPC expansion or mobilization, respectively. n = 6 (PGE2 Control), n = 5 (PGE2), n = 4 (all other groups) mice/group from three independent experiments. Scale bars, 1 mm. Vertical bars indicate mean ± SD. *P* values were calculated using two-tailed unpaired Student's *t*-test.

**a**

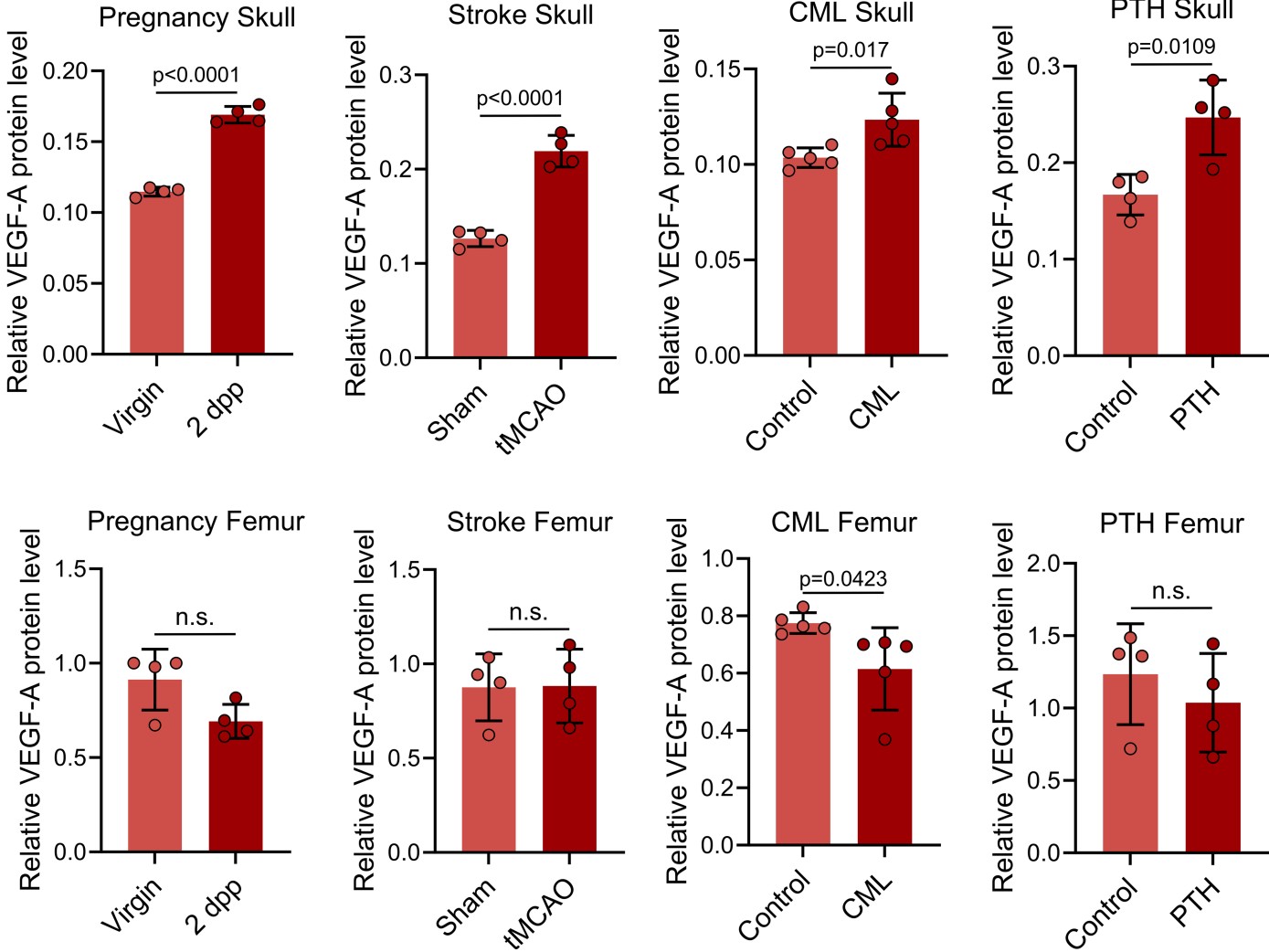

**Extended Data Fig. 8 | VEGF-A response to pathophysiological conditions.** **a**. VEGF-A protein concentrations in total BM lysates from skull or femur isolated from mice in various pathophysiological conditions, including pregnancy, stroke, chronic myeloid leukemia and sustained PTH treatment.

n = 4 (Pregnancy, Stroke and PTH) and n = 5 (CML) mice/group from two independent experiments. Vertical bars indicate mean ± SD. *P* values were calculated using two-tailed unpaired Student's *t*-test.

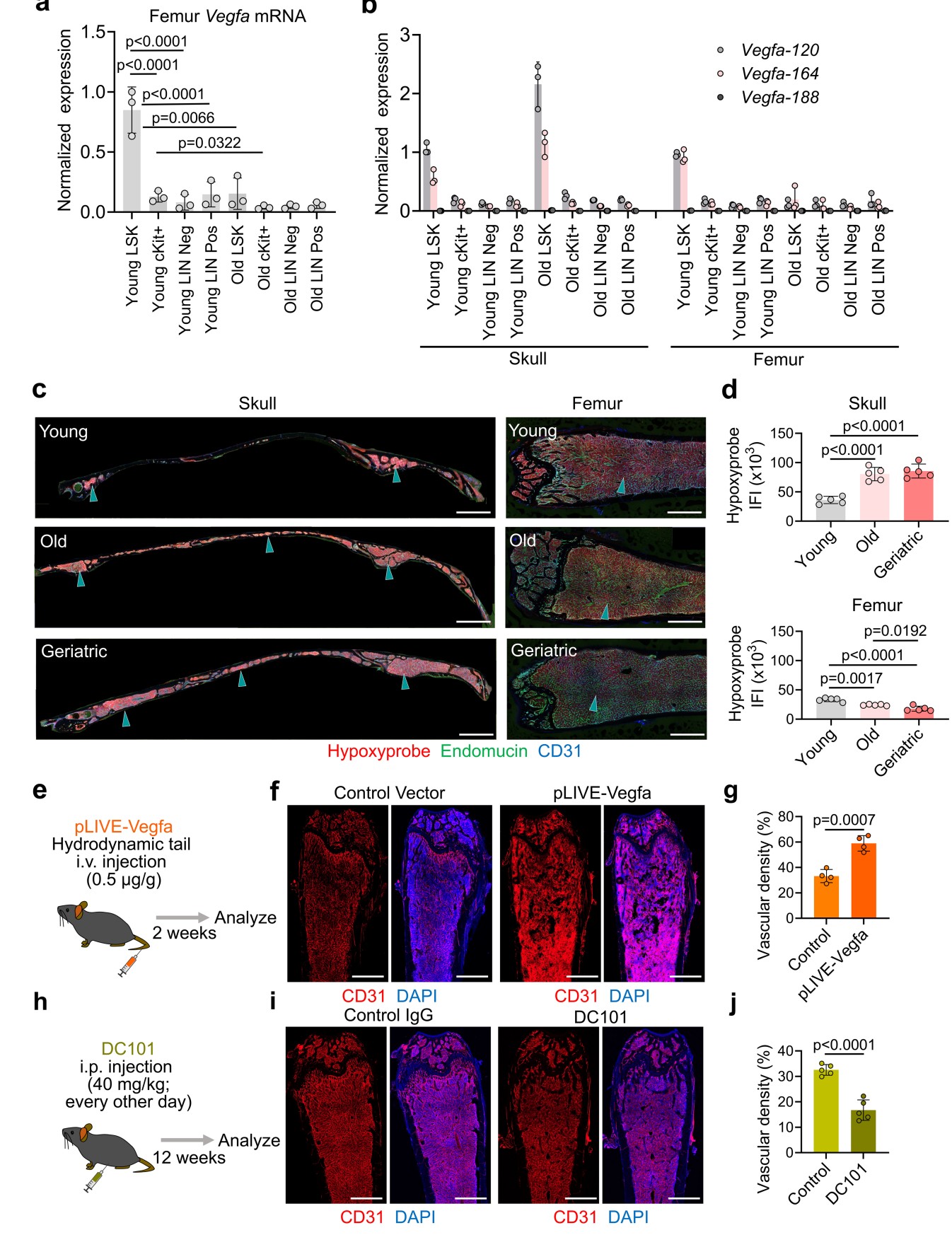

**Extended Data Fig. 9** | See next page for caption.

**Extended Data Fig. 9 | VEGF expression and hypoxia in BM compartments.**
**a**. qRT-PCR analyses of *Vegfa* mRNA in FACS-sorted LSK (LIN⁻ Sca1⁺ cKit⁺), cKit⁺ (LIN⁻ Sca1⁻ cKit⁺), Lin neg (LIN⁻ Sca1⁻ cKit⁻) and Lin pos (LIN⁺) populations isolated from femur BM of young or old mice (n = 5 mice pooled/sample from three independent experiments). **b**. qRT-PCR analyses of *Vegfa* mRNA isoforms in populations shown in (**a**) isolated from skull or femur BM of young or old mice (n = 5 mice pooled/sample from three independent experiments). **c, d**. IF staining (**c**) and quantification (**d**) of IV-injected Hypoxyprobe in young, old and geriatric skull and femoral BM, as indicated. Note increase of Hypoxyprobe signal in skull but not in femur (arrowheads, n = 5 mice/group from two independent experiments). Scale bars, 1 mm. Vertical bars indicate mean ± SD.

\*\*$P < 0.01$, \*\*\*$P < 0.001$ versus Young LSK (a) or Young (d), #$P < 0.05$ versus Old by Tukey multiple comparison test (one-way ANOVA). **e, h**. Diagram showing experimental scheme for *Vegfa* overexpression or anti-VEGFR2 blocking antibody DC101 treatment. **f-j**. In vivo IF staining (**f, i**) and quantification (**g, j**) of femoral BM blood vessels in mice after hydrodynamic injection of bone-homing *Vegfa* construct (n = 4 mice/group from three independent experiments) (**f, g**) or treatment with anti-VEGFR2 blocking antibody (DC101) for 12 weeks (n = 5 mice/group from two independent experiments) (**i, j**). Scale bars, 1 mm. Vertical bars indicate mean ± SD. *P* values were calculated using two-tailed unpaired Student's *t*-test.

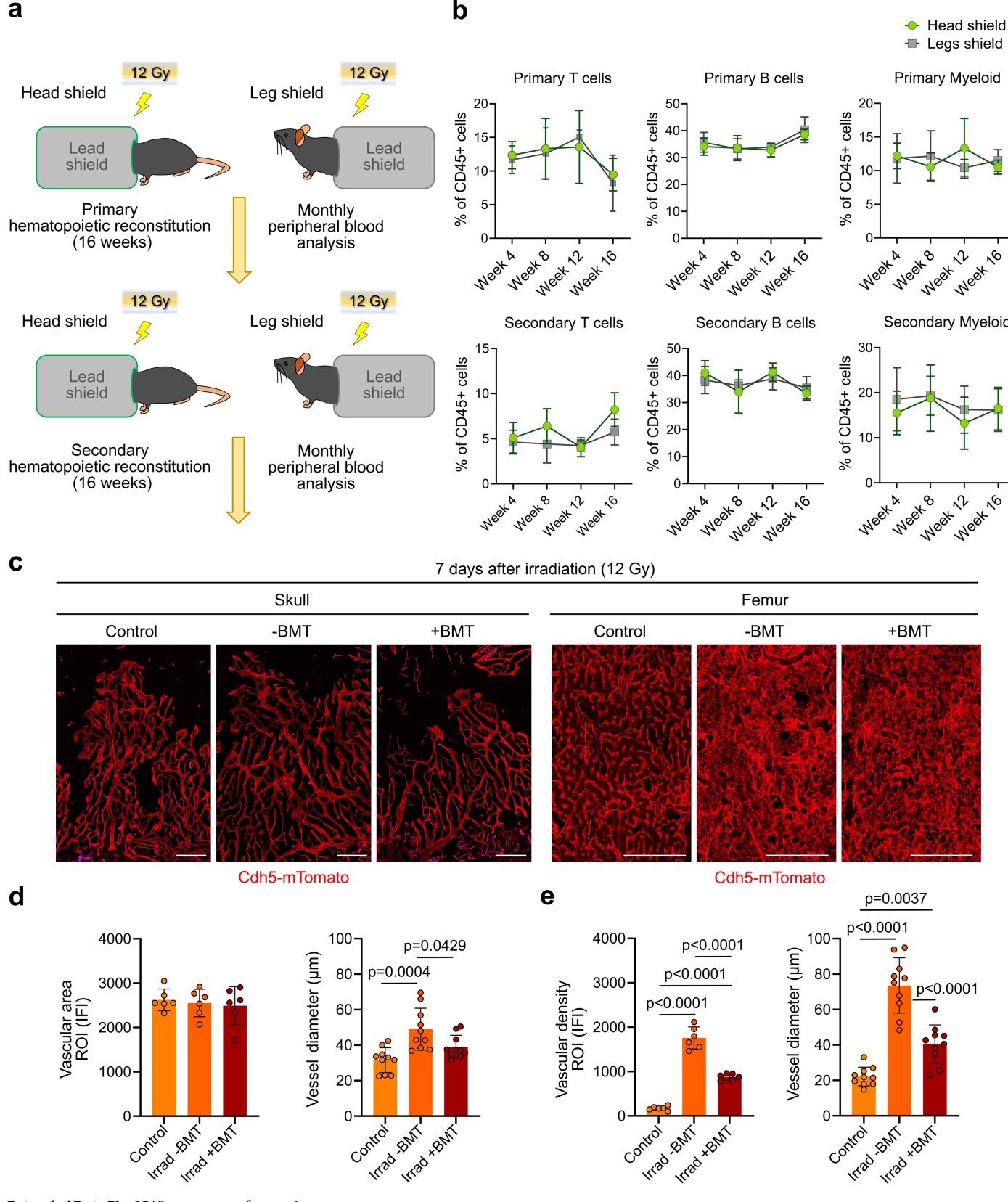

**Extended Data Fig. 10** | See next page for caption.

Extended Data Fig. 10 | Long-term reconstitution potential of skull HSCs. a. Diagram showing experimental scheme for serial hematopoietic reconstitution following serial partial irradiation with shielding. b. Monthly FACS analysis of T, B, and myeloid cells in peripheral blood isolated from animals with head versus leg shielding during primary and secondary hematopoietic reconstitution (n = 5 (Head shield), n = 4 (Leg shield) mice/group from two independent experiments). Vertical bars indicate mean ± SD. c-e. Fluorescence imaging (c) and quantification (d, e) of skull or femoral BM blood vessels in Cdh5-mTnG (mTomato, red) reporter mice at 7 days after lethal irradiation with (+) or without (-) bone marrow transplantation (BMT), as indicated. Note minimal vascular alterations in the skull BM compared to substantial vascular changes in femoral BM (n = 6 mice/group for vascular area/density, n = 10 randomly selected vessels from all samples/group for vessel diameter from three independent experiments). Vertical bars indicate mean ± SD. P values were calculated using Tukey multiple comparison test (one-way ANOVA).

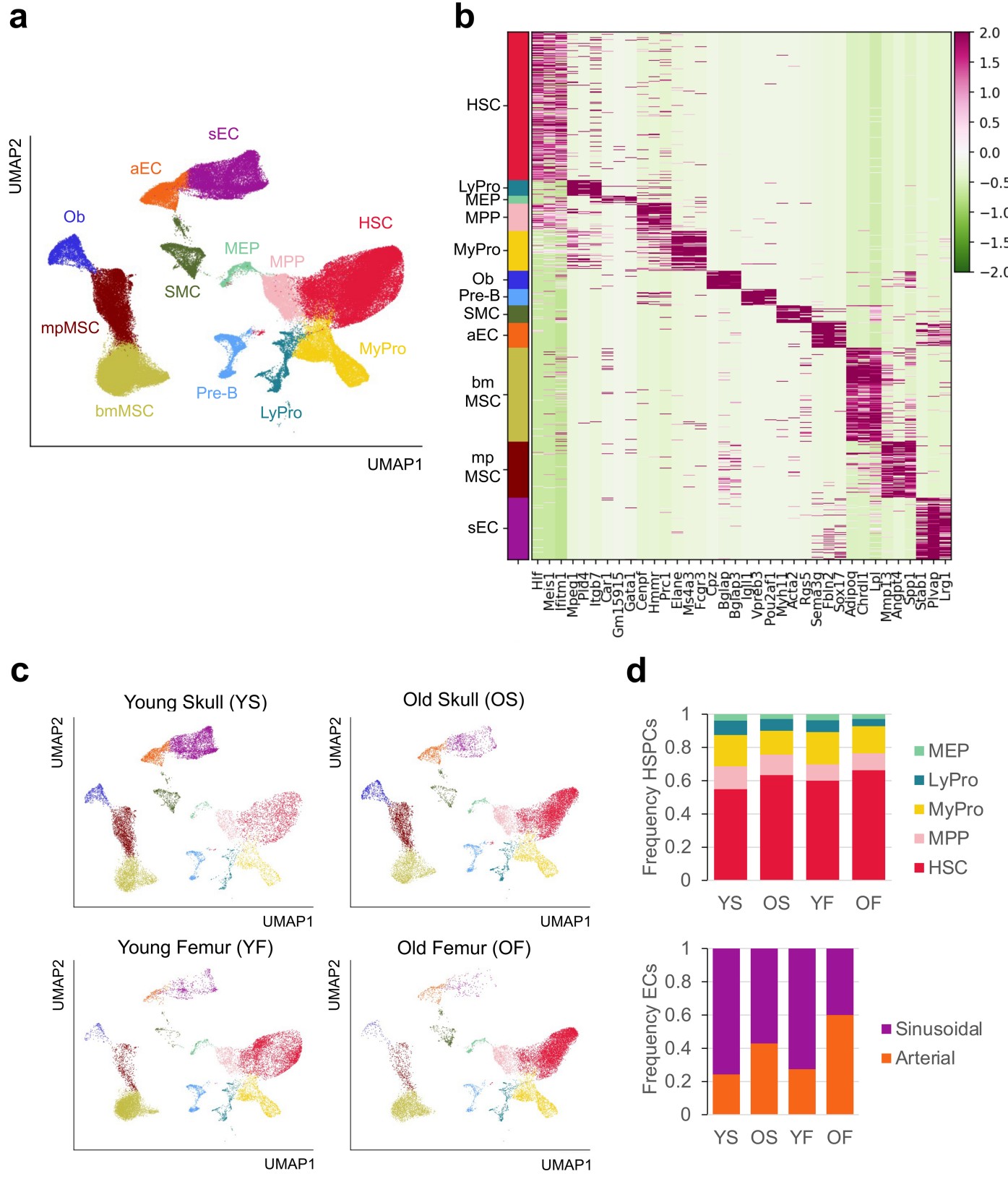

**Extended Data Fig. 11 | Single-cell RNA-sequencing analysis of HSPCs and stromal cells in skull and femur bone marrow during aging. a**. UMAP plot showing color-coded merged cell clusters of FACS-sorted cKit⁺ HSPCs and hematopoietic lineage-depleted bone marrow cells. Hematopoietic stem cell (HSC), multi-potent progenitor (MPP), myeloid progenitor (MyPro), lymphoid progenitor (LyPro), megakaryocyte-erythrocyte progenitor (MEP), pre-B-cell (Pre-B), sinusoidal endothelial cell (sEC), arterial endothelial cell (aEC), bone marrow mesenchymal stromal cell (bmMSC), metaphyseal mesenchymal stromal cell (mpMSC), smooth muscle cell (SMC), osteoblast (Ob). **b**. Heat map showing top cell marker genes of each cell population shown in (**a**). **c**. Group-selected color-coded cell clusters from young or old, skull or femur as indicated. **d**. Frequency plots of color-coded HSPCs (top) and endothelial cells (bottom) from young skull (YS), old skull (OS), young femur (YF), old femur (OF).

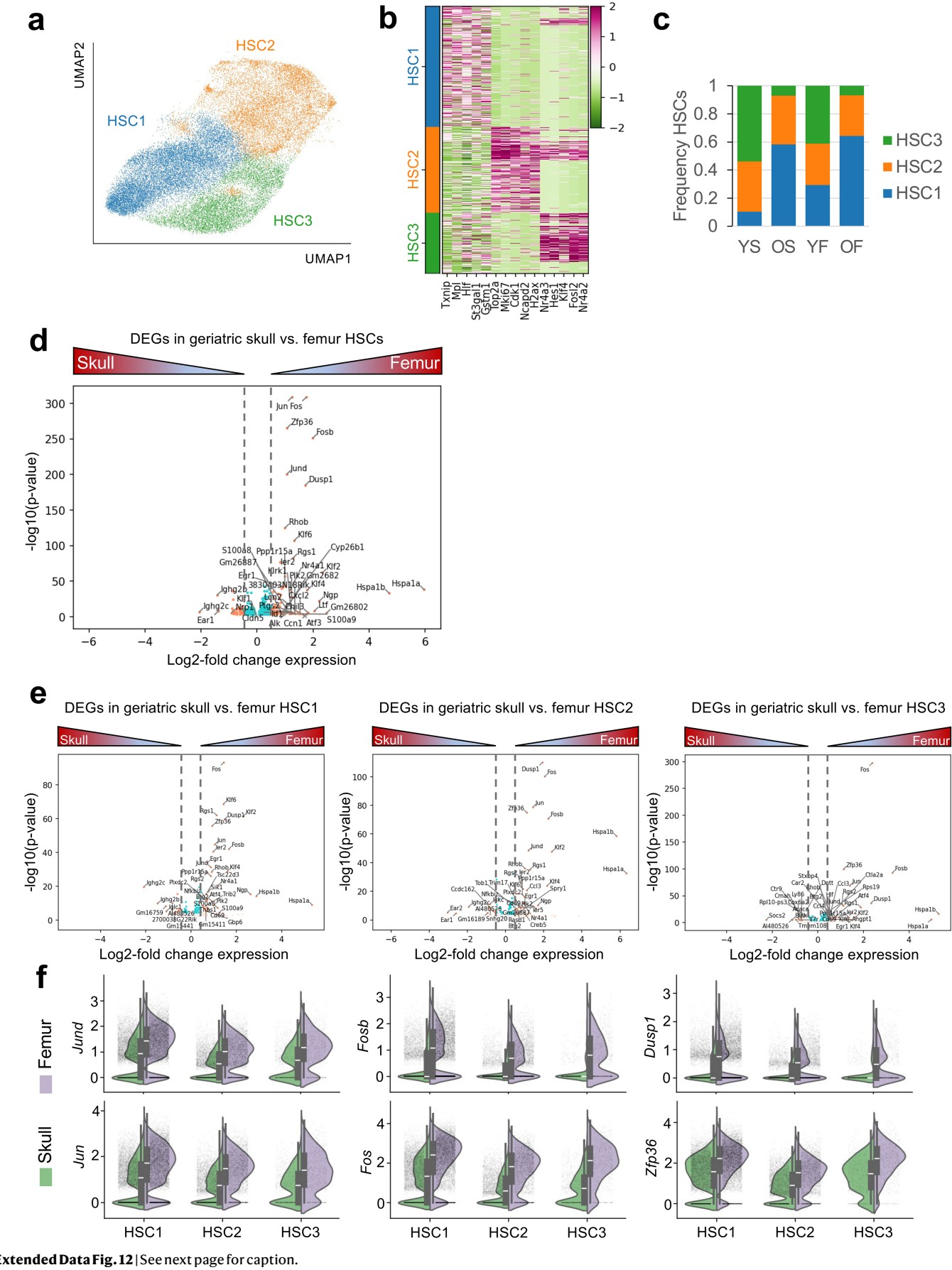

**Extended Data Fig. 12** | See next page for caption.

**Extended Data Fig. 12 | Single-cell RNA-sequencing analysis of HSCs in skull and femur bone marrow during aging. a**. UMAP plot showing color-coded merged cell clusters of HSCs identified in Extended Data Fig. 11. **b**. Heat map showing top cell marker genes of each cell population shown in (**a**). **c**. Frequency plots of color-coded HSCs from young skull (YS), old skull (OS), young femur (YF), old femur (OF). **d**. Differentially expressed genes (DEG) analysis comparing HSCs from geriatric skull versus femur. **e**. DEG analysis comparing each HSC subcluster from geriatric skull versus femur. **f.** Violin plots showing the expression of selected genes with log normalized values for inflammatory and myeloid determination factors in each HSC subcluster identified in (**a**). Skull HSC1 (n = 6216 cells), HSC2 (n = 3713 cells), HSC3 (n = 768 cells) and Femur HSC1 (n = 6726 cells), HSC2 (n = 3006 cells), HSC3 (n = 719 cells) isolated from n = 5 mice from two independent experiments. The box of each boxplot starts in the first quartile and ends in the third, with the line inside representing the median. Blue dots, $p$-adjusted value < 0.01 and $Log_2$ fold change <0.5; orange dots, $p$-adjusted value < 0.01 and $Log_2$ fold change > 0.5 by two-tailed unpaired Student's $t$-test.

## a

## b

**Extended Data Fig. 13 | HSPCs are dependent on microenvironment during aging. a, b.** Brightfield image (**a**) and quantification (**b**) of primary and secondary colony forming unit (CFU) assays of 500 FACS-sorted LSK cells isolated from young or geriatric skull or femur bone marrow (n = 4 mice/group from two independent experiments). Scale bar, 5 mm. Vertical bars indicate mean ± SD. *P* values were calculated using Tukey multiple comparison test (one-way ANOVA).

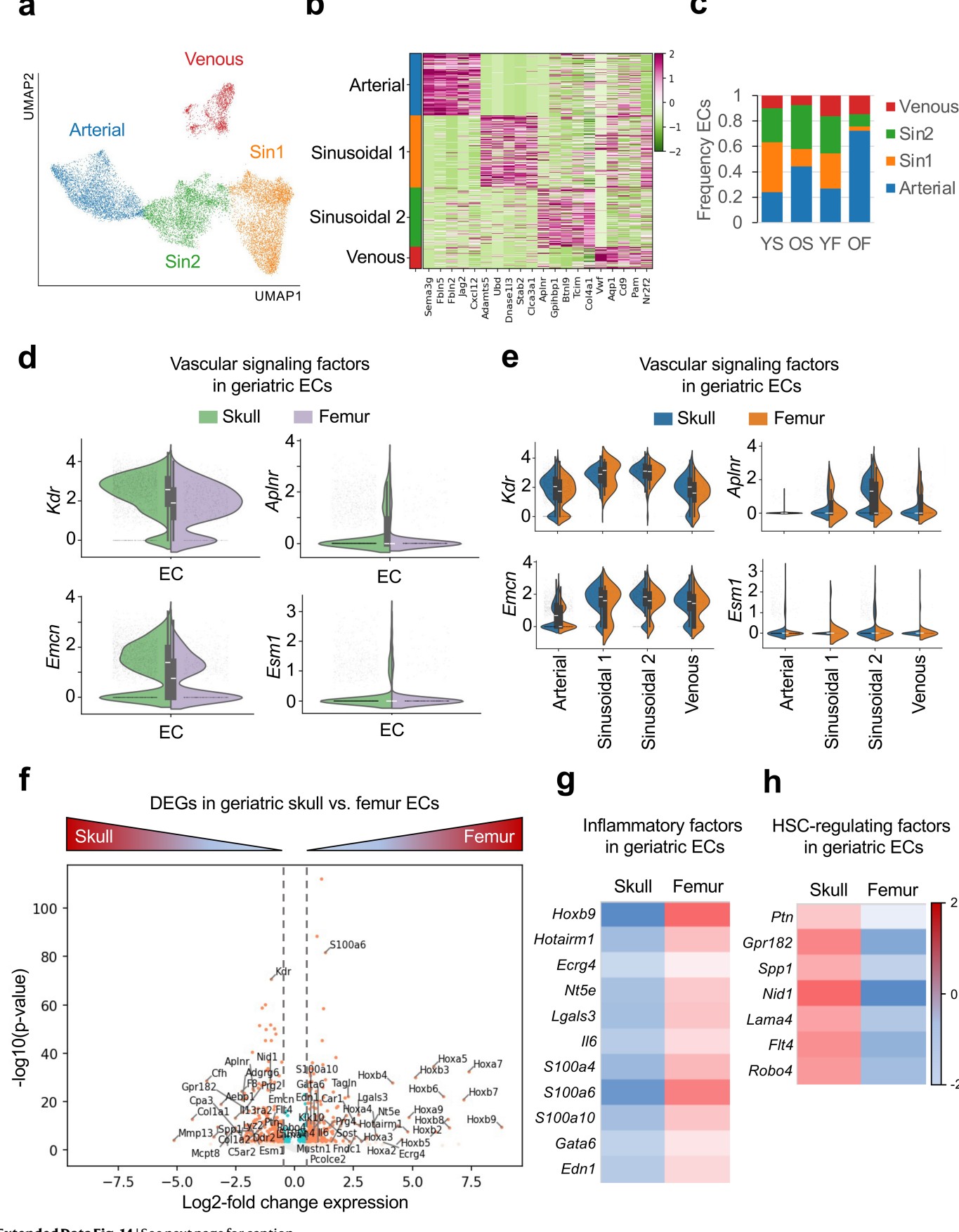

**Extended Data Fig. 14** | See next page for caption.

**Extended Data Fig. 14 | Single-cell RNA-sequencing analysis of ECs in skull and femur bone marrow during aging. a**. UMAP plot showing color-coded merged cell clusters of ECs identified in Extended Data Fig. 11. **b**. Heat map showing top cell marker genes of each cell population shown in (**a**). **c**. Frequency plots of color-coded ECs from young skull (YS), old skull (OS), young femur (YF), old femur (OF). **d, e**. Violin plots showing the expression of selected genes with log normalized values for vascular signaling factors in ECs (**d**) and each EC subcluster (**e**) from geriatric skull versus femur. Skull ECs (n = 2414 cells), Arterial (n = 1070 cells), Sinusoidal 1 (n = 329 cells), Sinusoidal 2 (n = 833 cells), Venous (n = 182 cells) and Femur ECs (n = 594 cells), Arterial (n = 430 cells), Sinusoidal 1 (n = 19 cells), Sinusoidal 2 (n = 58 cells), Venous (n = 87 cells) isolated from n = 5 mice from two independent experiments. The box of each boxplot starts in the first quartile and ends in the third, with the line inside representing the median. **f**. Differentially expressed genes (DEG) analysis comparing ECs from geriatric skull versus femur. **g, h**. Heat map showing the expression of selected genes with log normalized values for inflammatory factors (**g**) and HSC-regulating factors (**h**) identified in (**f**). Blue dots, $p$-adjusted value < 0.01 and $\log_2$ fold change <0.5; orange dots, $p$-adjusted value < 0.01 and $\log_2$ fold change > 0.5 by two-tailed unpaired Student's $t$-test.

|  |  |
|---|---|

# Reporting Summary

## Statistics

For all statistical analyses, confirm that the following items are present in the figure legend, table legend, main text, or Methods section.

| n/a | Confirmed | |
|---|---|---|
| ☐ | ☒ | The exact sample size (*n*) for each experimental group/condition, given as a discrete number and unit of measurement |
| ☐ | ☒ | A statement on whether measurements were taken from distinct samples or whether the same sample was measured repeatedly |
| ☐ | ☒ | The statistical test(s) used AND whether they are one- or two-sided<br>*Only common tests should be described solely by name; describe more complex techniques in the Methods section.* |
| ☐ | ☒ | A description of all covariates tested |
| ☐ | ☒ | A description of any assumptions or corrections, such as tests of normality and adjustment for multiple comparisons |
| ☐ | ☒ | A full description of the statistical parameters including central tendency (e.g. means) or other basic estimates (e.g. regression coefficient) AND variation (e.g. standard deviation) or associated estimates of uncertainty (e.g. confidence intervals) |
| ☐ | ☒ | For null hypothesis testing, the test statistic (e.g. *F*, *t*, *r*) with confidence intervals, effect sizes, degrees of freedom and *P* value noted<br>*Give P values as exact values whenever suitable.* |
| ☒ | ☐ | For Bayesian analysis, information on the choice of priors and Markov chain Monte Carlo settings |
| ☒ | ☐ | For hierarchical and complex designs, identification of the appropriate level for tests and full reporting of outcomes |
| ☒ | ☐ | Estimates of effect sizes (e.g. Cohen's *d*, Pearson's *r*), indicating how they were calculated |

*Our web collection on statistics for biologists contains articles on many of the points above.*

## Software and code

Policy information about availability of computer code

| Data collection | ZEN (Blue) 3.7 software (Carl Zeiss)<br>BD FACSDiva 9.1 software (BD Biosciences)<br>CFX Maestro 2.3 software (Bio-Rad) |
|---|---|
| Data analysis | ZEN (Black) 2.3 software (Carl Zeiss)<br>ImageJ 2.0.0 software (NIH)<br>IMARIS 10.0.1 software (Bitplane)<br>Prism 8.0 software (GraphPad)<br>FlowJo 10.3.0 (BD) |

For manuscripts utilizing custom algorithms or software that are central to the research but not yet described in published literature, software must be made available to editors and reviewers. We strongly encourage code deposition in a community repository (e.g. GitHub). See the Nature Portfolio guidelines for submitting code & software for further information.

## Data

Policy information about availability of data

All manuscripts must include a data availability statement. This statement should provide the following information, where applicable:
- Accession codes, unique identifiers, or web links for publicly available datasets
- A description of any restrictions on data availability
- For clinical datasets or third party data, please ensure that the statement adheres to our policy

The source data underlying all Figs. and Supplementary Figs. are provided as a Source Data file. All other data that support the findings of this study are available from the corresponding author upon reasonable request.

## Research involving human participants, their data, or biological material

Policy information about studies with human participants or human data. See also policy information about sex, gender (identity/presentation), and sexual orientation and race, ethnicity and racism.

| | |
|---|---|
| Reporting on sex and gender | Data from both sexes were used and consent to use this information was obtained. |
| Reporting on race, ethnicity, or other socially relevant groupings | All patients are Asian (Korean). |
| Population characteristics | 9 females (ages 29-39 for young, 61-68 for old) and 9 males (ages 21-39 for young, 61-69 for old). |
| Recruitment | Patients who underwent CT for evaluation of small cerebral aneurysm from April to May 2023 were eligible. Patients were excluded if they had a previous history of surgery or radiation therapy to the head and neck, vascular or bone-related medical implants, or a suspicious disease other than small cerebral aneurysm. |
| Ethics oversight | Asan Medical Center, Seoul, Republic of Korea |

Note that full information on the approval of the study protocol must also be provided in the manuscript.

# Field-specific reporting

Please select the one below that is the best fit for your research. If you are not sure, read the appropriate sections before making your selection.

☒ Life sciences        ☐ Behavioural & social sciences        ☐ Ecological, evolutionary & environmental sciences

For a reference copy of the document with all sections, see nature.com/documents/nr-reporting-summary-flat.pdf

# Life sciences study design

All studies must disclose on these points even when the disclosure is negative.

| | |
|---|---|
| Sample size | No specific statistical methods were used to predetermine sample size. |
| Data exclusions | No samples were excluded from analysis. |
| Replication | Experiments were replicated at least once for all analyses and number of reproductions of each experimental finding is described in each figure legend. All attempts at experimental replication were successful. |
| Randomization | Animals from different cages, but within the same experimental group, were selected to assure randomization. |
| Blinding | N/A |

# Reporting for specific materials, systems and methods

We require information from authors about some types of materials, experimental systems and methods used in many studies. Here, indicate whether each material, system or method listed is relevant to your study. If you are not sure if a list item applies to your research, read the appropriate section before selecting a response.

## Materials & experimental systems

| n/a | Involved in the study |
|---|---|
| ☐ | ☒ Antibodies |
| ☒ | ☐ Eukaryotic cell lines |
| ☒ | ☐ Palaeontology and archaeology |
| ☐ | ☒ Animals and other organisms |
| ☒ | ☐ Clinical data |
| ☒ | ☐ Dual use research of concern |
| ☒ | ☐ Plants |

## Methods

| n/a | Involved in the study |
|---|---|
| ☒ | ☐ ChIP-seq |
| ☐ | ☒ Flow cytometry |
| ☒ | ☐ MRI-based neuroimaging |

# Antibodies

**Antibodies used**

The following primary antibodies were used in the immunostaining of mouse samples: rat monoclonal anti-Endomucin (V.7C7) (Santa Cruz, Cat# sc-65495, 1:200 dilution), rabbit monoclonal anti-vATPaseB1/B2 (Abcam, Cat# 200839, 1:200 dilution), goat polyclonal anti-Osteopontin (R&D Systems, Cat# AF808, 1:200 dilution), goat polyclonal anti-CD31 (R&D, Cat# AF3628, 1:200 dilution), rabbit polyclonal anti-Caveolin 1 (Cell Signaling, Cat# 3238, 1:100), goat polyclonal anti-VEGF164 (R&D Systems, Cat# AF-493-NA, 1:200 dilution), rat monoclonal APC-conjugated anti-CD117 (c-Kit) (BD Biosciences, Cat# 553356, 1:100 dilution).

Species-specific secondary antibodies: Alexa Fluor 488 (Thermo Fischer Scientific, Cat# A21208), Alexa Fluor 594 (Thermo Fischer Scientific, Cat# A21209), Alexa Fluor 647 (Thermo Fischer Scientific, Cat# A31573 or Cat# A21447).

The following primary antibodies were used in in vivo immunostaining: rat monoclonal anti-CD31 (BD Biosciences, Cat# 553708. 1:10 dilution), rat monoclonal PE-conjugated anti-Endomucin (V.7C7)(Santa Cruz, Cat# 65495 PE, 1:10 dilution), rat monoclonal FITC-conjugated anti-CD45 (eBioscience, Cat# 11-0451-82, 1:10 dilution), hamster monoclonal FITC-conjugated anti-CD3e (eBioscience, Cat# 16-0031-82, 1:10 dilution), rat monoclonal PE-conjugated anti-CD45R/B220 (BD Biosciences, Cat# 553090, 1:10 dilution), rat monoclonal FITC-conjugated anti-CD11b (BD Biosciences, Cat# 553310, 1:10 dilution).

For lineage depletion, FACS sorting and analyses, the following antibodies were used: biotinylated rat monoclonal anti-hematopoietic lineage antibody cocktail (Miltenyi-Biotec, Cat# 130-092-613, 1:50 dilution), APC-conjugated rat monoclonal anti-CD117 (BD Biosciences, Cat# 553356, 1:100 dilution), FITC-conjugated rat monoclonal anti-CD117 (Biolegend, Cat# 105806, 1:100 dilution), FITC-conjugated rat monoclonal anti-Ly-6A/E (Sca-1) (eBioscience, Cat# 11-5981-85, 1:100 dilution), PerCP-Cy5.5-conjugated rat monoclonal anti-Ly-6A/E (Invitrogen, Cat# 45-5981, 1:100), APC-Cy7-conjugated hamster monoclonal anti-CD48 (BD Biosciences, Cat# 561242, 1:100 dilution), PE-conjugated rat monoclonal anti-CD150 (SLAM) (Biolegend, Cat# 115904, 1:100 dilution), Alexa Fluor 647-conjugated rat monoclonal anti-CD150 (Biolegend, Cat# 115918, 1:100 dilution), PE-Cy7-conjugated rat monoclonal anti-CD45 (eBioscience, Cat# 25-0451-82, 1:100 dilution), BV421-conjugated rat monoclonal anti-TER-119 (Biolegend, Cat# 116234, 1:100 dilution), FITC-conjugated rat monoclonal anti-CD71 (Biolegend, Cat# 113806, 1:100 dilution), Alexa Fluor 647-conjugated rat monoclonal anti-CD31 (BD Biosciences, Cat# 553708, conjugation described above, 1:200 dilution), PE-conjugated rat monoclonal anti-Endomucin (Santa Cruz, Cat# 65495 PE, 1:100 dilution), PE-Cy7-conjugated rat monoclonal anti-CD16/32 (eBioscience, Cat# 25-0161, 1:100 dilution), eFluor 450-conjugated rat monoclonal anti-CD34 (eBioscience, Cat# 48-0341, 1:100 dilution), PE-conjugated rat monoclonal anti-CD127 (eBioscience, Cat# 12-1271, 1:100 dilution), BV711-conjugated rat monoclonal anti-CD41 (BD Biosciences, Cat# 740712, 1:100 dilution), PE-conjugated rat monoclonal anti-CD105 (eBioscience, Cat# 12-1051-82, 1:100 dilution), APC-conjugated hamster monoclonal anti-CD3e (eBioscience, Cat# 17-0031, 1:100 dilution), PE-conjugated rat monoclonal anti-CD45R/B220 (BD Biosciences, Cat# 553090, 1:100 dilution), FITC-conjugated rat monoclonal anti-CD11b (BD Biosciences, Cat# 553310, 1:100 dilution), Alexa Fluor 405-(Invitrogen, Cat# S32351, 1:100 dilution) or APC-Cy7-(BD Biosciences, Cat# 554063, 1:100 dilution) conjugated streptavidin secondary antibody.

**Validation**

All the antibodies were validated for the species and applications (immunohistochemistry and FACS) by the corresponding manufacturer, which is described in the manufacturer's website. Our usage is described in the Methods section of the manuscript.

# Animals and other research organisms

Policy information about studies involving animals; ARRIVE guidelines recommended for reporting animal research, and Sex and Gender in Research

**Laboratory animals**

C57BL/6J mice were used for all experiments involving wild-type mice. Flk1-GFP reporter (Xu et al., 2010), Vav1-Cre, ROSA26-mTmG reporter, ROSA26-CAG-loxP-stop-loxP-KikGR knock-in mice were transferred, established, and bred in the SPF animal facility at Max Planck Institute for Molecular Biomedicine. All of these mice were maintained in the C57BL/6 background. Mice at the age of 10-14 weeks, 31-37 weeks, 52-75 weeks and >95 weeks were chosen for young, middle-aged, old and geriatric groups, respectively.

Mice were kept in individually ventilated cages (IVC), with constant access to food and water under a 12h light and 12h dark cycle regime. Air flow, temperature (21-22□C) and humidity (55-60%) were controlled by an air management system. Animals were checked daily and maintained in specific pathogen-free (SPF) conditions. Sufficient nesting material and environmental enrichment was provided.

**Wild animals**

The study did not involve wild animals.

**Reporting on sex**

*Indicate if findings apply to only one sex; describe whether sex was considered in study design, methods used for assigning sex.*

| | |
|---|---|
| Reporting on sex | *Provide data disaggregated for sex where this information has been collected in the source data as appropriate; provide overall numbers in this Reporting Summary. Please state if this information has not been collected. Report sex-based analyses where performed, justify reasons for lack of sex-based analysis.* |
| Field-collected samples | The study did not involve samples collected from the field. |
| Ethics oversight | All animal experiments were performed according to the institutional guidelines and laws, approved by local animal ethical committee and were conducted at the Max Planck Institute for Molecular Biomedicine (84-02.04.2016.A160, 81-02.04.2018.A171, 81-02.04.2020.A212, 81-02.04.2020.A416 and 81-02.04.2022.A198), Universitätsmedizin Berlin (G0220/17), Georg-Speyer-Haus (F123/2017) and the University Medical Center Mainz Institute of Transfusion Medicine (G23-1-067 A1TE) under the indicated permissions granted by the Landesamt für Natur, Umwelt und Verbraucherschutz (LANUV) of North Rhine-Westphalia, the State Office for Health and Social Affairs Berlin, Regierungspräsidium Darmstadt and the Landesuntersuchungsamt Rheinland-Pfalz, Germany. |

Note that full information on the approval of the study protocol must also be provided in the manuscript.

# Flow Cytometry

## Plots

Confirm that:

☒ The axis labels state the marker and fluorochrome used (e.g. CD4-FITC).

☒ The axis scales are clearly visible. Include numbers along axes only for bottom left plot of group (a 'group' is an analysis of identical markers).

☒ All plots are contour plots with outliers or pseudocolor plots.

☒ A numerical value for number of cells or percentage (with statistics) is provided.

## Methodology

| | |
|---|---|
| Sample preparation | Mice from each age group were euthanized and skull and femur were harvested. Skulls were chopped with scissors in FACS buffer before crushed with mortar and pestle; femurs were crushed without chopping. BM stromal samples were dissociated with Collagenase I (Gibco, Cat# 17100-017, 2 mg/ml) and Collagenase IV (Gibco, Cat# 17104-019, 2 mg/ml) in PBS for 20 minutes at 37ºC with intermittent shaking. Cell suspensions were strained through a 40 ⬚m mesh filter, resuspended in RBC lysing buffer (when applicable) and washed with FACS buffer. |
| Instrument | FACSAria Fusion (BD Biosciences), FACSymphony A5 Cell Analyzer (BD Biosciences) |
| Software | FACSDiva 8.0.2 software (BD Biosciences), FACSDiva 9.1 software (BD Biosciences) |
| Cell population abundance | Sorted LIN-negative cells were 1.5~2.0% of single cells. Purity was achieved at 95~98% and was confirmed by immediate analysis on a FACS analyzer. |
| Gating strategy | Preliminary FSC/SSC gating was established based on the exclusion of cellular debris and erythrocytes. Single cells were gated based on a single linear cluster within FSC-W vs. FSC-A. Positive populations (>10^3 fluorescence intensity) were gated based on clear population separation from the negative population (<10^2 fluorescence intensity). |

☒ Tick this box to confirm that a figure exemplifying the gating strategy is provided in the Supplementary Information.

