## [Peer Review File · Nature]

Manuscript Title: Adult skull bone marrow is an expanding and resilient hematopoietic reservoir

Reviewer Comments & Author Rebuttals

Reviewer Reports on the Initial Version:

Referees' comments:

Referee #1 (Remarks to the Author):

Key results: Please summarize what you consider to be the outstanding features of the work.

1. The skull exhibits age-associated expansion of bone marrow and vasculature, distinct from the aging characteristics of the femoral bone marrow.
2. Reduction of adipocytes and inflammatory cytokines in the skull as compared to the femur in aging, along with increasing contributions from the skull to overall hematopoietic output and decreased myeloid bias, point to age-related functional differences and unique resilience in the bone marrow of the skull.
3. The skull bone marrow exhibits vascular changes in response to physiological and pharmacological events that are distinct from the femoral bone marrow.

Validity: Does the manuscript have flaws which should prohibit its publication? If so, please provide details.

None.

Originality and significance: If the conclusions are not original, please provide relevant references. On a more subjective note, do you feel that the results presented are of immediate interest to many people in your own discipline, and/or to people from several disciplines?

Anatomical differences between different bone marrow compartments and their associated functional consequences are of interest to the field. The characterization of skull bone marrow and vascular expansion with age is novel and significant. While others have examined distinct properties of the skull bone marrow at the cellular level (ref 76 in paper: PMID 37562402), this work uses an imaging-centered approach to reveal large scale differences in how skull bone marrow structure changes in response to aging and other diverse challenges and conditions. The functional consequences of age-related skull bone marrow and vascular expansion are unexpected and highly consequential: that older skull bone marrow HSPC can rescue an irradiated animal where HSPC from other sites cannot. That points to unique niche-specific modification of an important aging phenotype.

Data & methodology: Please comment on the validity of the approach, quality of the data and quality of presentation. Please note that we expect our reviewers to review all data, including any extended data and supplementary information. Is the reporting of data and methodology sufficiently detailed and transparent to enable reproducing the results?

In demonstrating that the calvarial bone marrow is fundamentally different from the bone marrow of the long bone, the authors rely extensively on evidence of a unique response in calvarial bone marrow and vasculature to physiological challenges. However, the authors exclusively used female mice for experiments other than the initial imaging of bone marrow expansion in aging. Given that the authors show a more rapid expansion of calvarial bone marrow with age in female as compared to male mice, findings of differential bone marrow vascular expansion in the calvarial bone marrow in response to other biologically relevant events (i.e. pregnancy excepted) are of interest. Limiting the study to females is understandable given the technical demands of the study but should at least be justified and commented upon in the text.

The Control Vector skull in figure 4f shows decreased vasculature in comparison to the Control IgG skull in figure 4h. How were these controls chosen? Use of equally vascularized control skulls between these two experiments would strengthen the authors' argument. Additionally, were the experiments shown in 4f and 4h both done in equally aged mice? The authors only specify that the animals in the DC101 and Control IgG groups (Fig 4.h) were old, and they do not note the age of the animals in Fig. 4f.

Appropriate use of statistics and treatment of uncertainties:

The authors rely on the Tukey multiple comparison test (one-way ANOVA) for comparison between multiple groups and the two-tailed unpaired Student's t test for comparison between two groups. These statistical tests are appropriately used. Error bars and probability values are accurately described.

Suggested improvements: Please list additional experiments or data that could help strengthening the work in a revision.

1. The finding that bone marrow in the older animal from the skull is capable of rescuing mice from lethal irradiation while marrow from the femur is not and that the marrow contributes to >20% of blood cells in homeostasis in older animals is fascinating and unexpected. That those cells are less myeloid in nature is similarly surprising. The authors rightly point out that the bone marrow of the skull is therefore special and perhaps 'resilient' to the effects of aging. But that begs the question of what is different about the stem cells in aged skull bone marrow compared with the femur? I realize this is likely the subject of future studies, but the field (and therefore the impact of the paper) would greatly benefit from scRNAseq data on hematopoietic cells or bulk RNAseq of specific subsets of HSPC in those two tissues.

2. The authors demonstrate the very intriguing finding of older calvarial bone marrow transplantation resulting in increased vascularity of calvarial (but not femoral) bone marrow in the recipient young mice.

But I don't see the control of older femoral bone marrow as the donor cells. That is necessary to define that the effect is calvarial bone marrow cell autonomous and to justify the studies on VEGF.

3. While increased Vegfa expression by HSPCs in old mice is shown to drive the expansion of skull bone marrow and vasculature with age, the authors do not investigate whether Vegfa and bone marrow expansion is mechanistically linked to the resilience of skull-derived HSPCs during aging.

4. Additionally, to further explore the role of VEGF as a driver of bone marrow expansion, the authors could examine the VEGF-A protein levels and mRNA expression in the context of pregnancy, stroke, and CML. Experiments investigating these mechanisms would grant insight into why the skull bone marrow exhibits a vascular response distinct from the femur under various physiological conditions.

5. One of the key findings of this paper is that skull-derived HSPCs maintain their functional integrity with age. It is not evident whether this is a consequence of the dynamic skull microenvironment the authors have identified. The authors could perform colony forming unit assays to examine whether any cell-intrinsic proliferative features persist after their removal from the niche.

Clarity and context: Is the abstract clear, accessible? Are abstract, introduction and conclusions appropriate?

The abstract presents a clear outline of the findings in the paper. All conclusions in the abstract are well-supported by the data presented, and the authors highlight their novel findings.

In the introduction, the authors discuss previous literature that describes the functional role of crosstalk between structural elements of the skull in terms of immunity as well as dysfunctional myelopoiesis in aging in an effort to introduce the possibility of functional differences between long and flat bones. While they demonstrate that the calvarial bone marrow microenvironment exerts protective effects during aging, mechanistic connections between large-scale bone marrow vascular expansion and the ability of skull-resident HSPCs to resist aging are not explored, leaving the questions posed in the introduction unanswered.

The authors provide substantial evidence for their claims, supporting each conclusion with sufficient data.

Inflammatory material: Does the manuscript contain any language that is inappropriate or potentially libelous?

No.

Please indicate any particular part of the manuscript, data, or analyses that you feel is outside the scope of your expertise, or that you were unable to assess fully.

Referee #2 (Remarks to the Author):

In Koh et al., the authors examine the vascular compartment in the calvarium and observed differences in the aging associated changes in skeletal vasculature between calvarium and long bones. They also investigate whether these changes in turn translate into differences in calvarial leukopoiesis. The topic of whether calvarial marrow is fundamentally different than other marrow sites is timely and important, especially given very recent work such as Kolabas et al. Cell 2023 and other work as appropriately cited in the introduction to the present manuscript identifying differences between calvarial marrow and other marrow sites. This manuscript also makes some novel observations regarding differences in aging associated changes in calvarial vs long bone vasculature. However, the work overall is extremely descriptive and the depth of investigation into the areas considered, especially the links to the calvarial marrow and hematopoiesis, is shallow and lacking sufficient rigor. These issues greatly reduce the potential impact of the study.

Major points:

1. The most important overall point is that the vast majority of the experiments are descriptive in nature, observing increases and decreases largely in calvarial vasculature with various treatments. Investigation into the mechanisms is very limited, and where performed relies on very well-established pathways regulating skeletal angiogenesis (see additional comments below). There is little overall data making the case that the differences observed carry physiologic or pathophysiologic importance.
2. The whole-mount areal measurements of DAPI+ cells in Fig 2-4 are not a compelling way to make the case that there is expansion of total calvarial marrow. It is important that this analysis be conducted with a combination of histology and volumetric analysis, such as by uCT. Additionally, an analysis of marrow composition and absolute cell numbers of specific fractionated hematopoietic populations and endothelial populations should be demonstrated by flow cytometry. Areal measurements DAPI+ cells and vessel immunostaining are insufficient to make a case about marrow dynamics. In particular, using DAPI+ cells on whole mount to demonstrate marrow area is not seen as a rigorous approach, as the identity of the DAPI+ cells is unclear. A combination of flow cytometry and immunostaining of frozen sections for hematopoietic cell type defining markers is needed to demonstrate changes in total marrow amount and composition.
3. Is hydroxyprobe staining a linear indicator of oxygen tension? The fold comparisons may be misleading if this signal does not have a linear relationship between oxygen tension and probe signal within the measured range. The increase in hypoxia in skull vs femur bone marrow with aging would appear on its face to be counter to the increase in calvarial vascular density documented elsewhere in this manuscript, though it is appreciated that the flow dynamics involved may be more complex than simply scaling with vessel area. Some resolution of this apparent discrepancy is needed.
4. Data cataloguing VEGFA expression in calvarial vs long bone cells is descriptive, and no evidence is offered that these differences are determinative of the vascular phenotypes observed. Additionally, invoking differences in the HSPC compartment between skeletal sites as driving local vascular

phenotypes would need to account for the ability of these HSPCs to circulate.

5. Related to comment 4 above, it isn't compelling to show that anti-VEGFA decreases calvarial vasculature and VEGFA expression decreases calvarial vasculature, as these relationships were already well established within the skeleton by Olsen and others many years ago. Instead, the potential novelty here lies in showing that fundamental local differences in VEGFA signaling are determinative of differences in skeletal vasculature between long bones and other sites, otherwise the VEGFA data doesn't strongly advance the conclusions of the manuscript.

6. The finding of a potential difference in myelopoiesis between calvarium and other skeletal sites is interesting and potentially important, but not rigorously established. The shielding studies in Fig 5e are difficult to interpret, as differences in the secondary responses to the irradiation induced reductions in total body hematopoietic output could alter the results. Direct investigation of myeloid precursors and the discrete steps in the myeloid differentiation cascade in the calvarium vs long bone is essential to make the desired conclusions. Application of the already established photoconversion system in Fig 5f could be helpful towards demonstrating that, if site specific differences in myeloid composition are observed, that these translate into differential contributions to the peripheral myeloid pool.

7. As an extension of comment 1 above, the increase in vascular area in pregnancy, stroke, CML and PTH treatment are all interesting findings, but not impactful or strongly significant unless the specific physiologic/pathophysiologic importance of the differential properties of the calvarial vs long bone vascular response can be demonstrated, at least for some subset of these conditions. Simply cataloging vascular changes by site is insufficient for a report at this level.

8. Similar issues apply to the cytokine studies in Fig 5c—it is not enough to profile these, but there needs to be a deeper investigation of the responsible cellular sources and determination that some element of these distinct cytokine profiles impacts the aging of the calvarial vs long bone marrow.

9. The studies shown describe differences in the phenotypic aging of the stromal compartment, but do not make a strong case for differences in phenotypic aging within the HSC compartment other than investigating relative myeloid output. Key molecular hallmarks of aging should be investigated within the HSC compartment. As above, any investigation of this topic needs to account for circulation of HSCs and deconvolute the degree to which phenotypes observed represent niche vs HSC-lineage intrinsic effects. Do any hematopoietic effects observed relate to the vascular phenotypes studied in the rest of the manuscript? No data is presented to demonstrate that this is the case, so these studies of hematopoiesis seem disconnected from the rest of the manuscript and lack any mechanistic basis.

Minor points

1. The vATPase imaging in Sup Fig 1c is not ideal as the stained cells are not clearly demarcated as individual cells but more as a swath of staining near the bone.
2. Given recent reports on skeletal lymphatics and the topic of the current report, it would seem to be an obvious and related question whether lymphatics differ between calvarium and long bones.
3. Is the measurement of sinusoid diameter in Fig 1 derived from the whole mount imaging? This analysis would likely be best conducted on histologic sections given the possibility that vessels "stacked" and overlapping the z-plane could give a misleading sense of vessel diameter.
4. Considering stromal cells as total lin-sca1-kit- cells for the purpose of gene expression analysis in 242 is somewhat problematic. Some stromal cells do express Sca1, so the exclusion of the Sca1+ stromal

cells here is puzzling and appears arbitrary. Also, this approach lumps a large number of stromal cells, each with likely very different VEGFA expression, together and makes interpretation difficult.

5. It is recommended that qPCR studies in Ext Fig 8a also be conducted in long bones for comparison.

6. Discussing the femoral bone marrow as being "more inflammatory" on the basis of cytokine profiling seems overbroad.

Referee #3 (Remarks to the Author):

Ralph Adam's group has employed sophisticated genetic models and functional assays to show that in the skull of mice there are continued formation of VEGFR2+ large caliber sinusoidal endothelial cells that support hematopoietic stem and progenitors (HSPCs) survival and expansion. They show that the release of Vgfa120, and Vegfa165 by the hematopoietic cells or the parathyroid hormone turns on VEGFR2 thereby activating angiogenesis in the skull sinusoidal endothelial cells expanding and sustaining the LSK HSPC populations. They also show that the endothelial cells formed within the skull compose of younger endothelial cells with less inflammation.

Comments:

This a very well-executed, well-controlled and well-designed and important paper demonstrating the heterogeneity of the marrow vascular niches for the support of the adult and aging hematopoiesis. The characterization of the various vascular beds within the skull are well performed and the genetic models clearly highlights the significance of VEGF-A and VEGFR2 in mediating angiogenesis and LKS expansion.

The only concerns I have are as follows:

1) To prove that indeed true LKS hematopoietic stem cells are populating sinusoidal endothelial cells, the authors might consider performing secondary transplantation of the primary engrafted LKS populations.

2) Inclusion some type of human data to prove that same phenomenon might be applicable to humans aging or young skull could be revealing.

3) Does DC101 the anti-VEGFR2 neutralizing antibody also shuts down angiogenesis in the marrow thereby impairing hematopoiesis or only targets skull angiogenic endothelial cells in aging mice?

Author Rebuttals to Initial Comments:

Referees' comments:

Referee #1 (Remarks to the Author):

Overall Q&A-1: Key results: Please summarize what you consider to be the outstanding features of the work.

1. The skull exhibits age-associated expansion of bone marrow and vasculature, distinct from the aging characteristics of the femoral bone marrow.
2. Reduction of adipocytes and inflammatory cytokines in the skull as compared to the femur in aging, along with increasing contributions from the skull to overall hematopoietic output and decreased myeloid bias, point to age-related functional differences and unique resilience in the bone marrow of the skull.
3. The skull bone marrow exhibits vascular changes in response to physiological and pharmacological events that are distinct from the femoral bone marrow.

Validity: Does the manuscript have flaws which should prohibit its publication? If so, please provide details.

None.

Originality and significance: If the conclusions are not original, please provide relevant references. On a more subjective note, do you feel that the results presented are of immediate interest to many people in your own discipline, and/or to people from several disciplines?

Anatomical differences between different bone marrow compartments and their associated functional consequences are of interest to the field. The characterization of skull bone marrow and vascular expansion with age is novel and significant. While others have examined distinct properties of the skull bone marrow at the cellular level (ref 76 in paper: PMID 37562402), this work uses an imaging-centered approach to reveal large scale differences in how skull bone marrow structure changes in response to aging and other diverse challenges and conditions. The functional consequences of age-related skull bone marrow and vascular expansion are unexpected and highly consequential: that older skull bone marrow HSPC can rescue an irradiated animal where HSPC from other sites cannot. That points to unique niche-specific modification of an important aging phenotype.

Data & methodology: Please comment on the validity of the approach, quality of the data and quality of presentation. Please note that we expect our reviewers to review all data, including any extended data and supplementary information. Is the reporting of data and methodology sufficiently detailed and transparent to enable reproducing the results?

In demonstrating that the calvarial bone marrow is fundamentally different from the bone marrow of the long bone, the authors rely extensively on evidence of a unique response in calvarial bone marrow and vasculature to physiological challenges. However, the authors exclusively used female mice for experiments other than the initial imaging of bone marrow expansion in aging. Given that the authors show a more rapid expansion of calvarial bone marrow with age in female as compared to

male mice, findings of differential bone marrow vascular expansion in the calvarial bone marrow in response to other biologically relevant events (i.e. pregnancy excepted) are of interest. Limiting the study to females is understandable given the technical demands of the study but should at least be justified and commented upon in the text.

Response: Thank you very much for your summary and feedback. While we provide evidence for skull bone marrow expansion in females and males of both mice and human subjects in aging, which is a key finding in our study, it was indeed technically infeasible to investigate sex-specific differences for each pathophysiological condition and in the mechanistic/functional studies. As suggested by the reviewer, the revised manuscript provides justification for our focus on female mice, which was also caused by the notion that the BM expansion is more pronounced in females at 35 and 75 weeks (Extended Data Fig. 1a, b). Moreover, some conditions (pregnancy and lactation) can obviously not be studied in males.

The Control Vector skull in figure 4f shows decreased vasculature in comparison to the Control IgG skull in figure 4h. How were these controls chosen? Use of equally vascularized control skulls between these two experiments would strengthen the authors' argument. Additionally, were the experiments shown in 4f and 4h both done in equally aged mice? The authors only specify that the animals in the DC101 and Control IgG groups (Fig 4.h) were old, and they do not note the age of the animals in Fig. 4f.

Response: All animals used in Figure 4f and 4h were 10 weeks old at the beginning of treatment. Animals injected with the *pLIVE* vector in Fig. 4f were sacrificed after 2 weeks of *Vegfa* overexpression, whereas animals injected with DC101 in Fig. 4h were sacrificed after 12 weeks of sustained VEGFR2 inhibition. Therefore, there is a 10-week age difference between the animals in these two experiments at the time of analysis due to their respective treatment schemes. We address this important question by including experimental schemes with treatment lengths in Extended Data Figure 10 and by indicating the age of the analyzed animals in Fig. 4f and 4h.

Experimental scheme for Figure 4f

Experimental scheme for Figure 4h

Appropriate use of statistics and treatment of uncertainties:

The authors rely on the Tukey multiple comparison test (one-way ANOVA) for comparison between multiple groups and the two-tailed unpaired Student's t test for comparison between two groups. These statistical tests are appropriately used. Error bars and probability values are accurately described.

Response: We appreciate the reviewer's overall positive comments and excellent suggestions for improving the manuscript.

Suggested improvements: Please list additional experiments or data that could help strengthening the work in a revision.

Comment 1: The finding that bone marrow in the older animal from the skull is capable of rescuing mice from lethal irradiation while marrow from the femur is not and that the marrow contributes to >20% of blood cells in homeostasis in older animals is fascinating and unexpected. That those cells are less myeloid in nature is similarly surprising. The authors rightly point out that the bone marrow of the skull is therefore special and perhaps 'resilient' to the effects of aging. But that begs the question of what is different about the stem cells in aged skull bone marrow compared with the femur? I realize this is likely the subject of future studies, but the field (and therefore the impact of the paper) would greatly benefit from scRNAseq data on hematopoietic cells or bulk RNAseq of specific subsets of HSPC in those two tissues.

Response: Thank you very much for your comment. A recent study (Kolabas et al., PMID: 37562402) has proposed that mature immune cells of the skull bone marrow are molecularly distinct from BM in other parts of the murine but also human skeletal system. Furthermore, it has been shown that skull BM is functionally specialized for the immune surveillance of the central nervous system (Brioschi et al., PMID: 34083450; Cugarra et al., PMID: 34083447). In the context of these reports, our new findings regarding the dynamic expansion of skull BM in response to aging and other pathophysiological stimuli will be of great interest to the scientific community.

To specifically address whether HSC heterogeneity between the skull and femur could contribute to hematopoietic resilience against aging, we performed scRNA-seq of FACS-sorted cKit⁺ HSPCs and hematopoietic lineage-depleted stromal/endothelial cells isolated from either bone marrow compartment in young or geriatric mice (Extended Data Fig. 13, 14, 16 and below). Sub-clustering analysis identified distinct HSC, intermediate, myeloid, lymphoid, and megakaryocytic progenitors within the HSPC samples and various subtypes of endothelial, mesenchymal, and mural cells in the stromal samples (which are further interpreted in response to a later comment). Although no substantial proportional increase in the HSC subcluster could be observed in this lineage-depleted cell atlas, further sub-clustering of HSCs yielded 3 distinct subpopulations, of which HSC1 showed the highest expression of quiescence/stem cell maintenance markers (*Txnip*, *Mpl*, *Hlf*). This cluster also had the lowest expression of proliferation markers (*Top2a*, *Mki67*), which were most highly expressed by HSC2. Notably, HSC1 proportion rose drastically with aging in both the skull and in the femur.

scRNA-seq analysis of bone marrow

scRNA-seq analysis of HSCs

Strikingly, DEG analyses of HSCs between geriatric skull vs geriatric femur showed significant differences in the expression of stress-induced inflammatory genes. Many of these genes, such as *Hspa1b*, *Jund*, *Fosb*, *Dusp1*, *Klf1*, *Klf2*, and *Klf4* were higher in HSCs from the geriatric femur relative to age-matched skull. Gene expression of factors regulating myeloid differentiation and response, such as *Jun*, *Fos*, *Zfp36*, *Ngp*, *Ltf*, *Chil3*, *Egr1*, *S100a8*, and *S100a9*, was also substantially higher in HSCs isolated from the geriatric femur. Since clonal hematopoiesis is a major hallmark of aging which exponentially exposes aging HSPCs to detrimental mutations, we performed gene set enrichment analysis (GSEA) of our geriatric femur HSC DEGs with that of Tet2-knockout (KO) HSCs, which exhibit clonal hematopoiesis with severe myeloid-biased inflammation (Moran-Crusio et al., PMID: 21723200; Izzo et al., PMID: 32203468) that is instigated and further exacerbated by increased IL-1 signaling (Caiado et al., PMID: 36379023; McClatchy et al., PMID: 38062031). This analysis shows that HSCs from geriatric femur show strong enrichment of the aforementioned inflammatory pathways occurring in aging-associated clonal hematopoiesis (see below, reviewers only).

scRNA-seq analysis of HSCs

Taken together, our targeted analysis of HSCs clearly shows that the skull bone marrow maintains hematopoietic resilience against aging at the most primitive HSC level by evading inflammatory stress-induced damage and the resulting myeloid bias. We have also generated a valuable resource and established the important groundwork for future studies of HSCs and progenitors with respect to regulatory and crosstalk mechanisms of HSC differentiation that are unique to the aging skull.

Comment 2. The authors demonstrate the very intriguing finding of older calvarial bone marrow transplantation resulting in increased vascularity of calvarial (but not femoral) bone marrow in the recipient young mice. But I don't see the control of older femoral bone marrow as the donor cells. That is necessary to define that the effect is calvarial bone marrow cell autonomous and to justify the studies on VEGF.

Response: We appreciate this suggestion and have performed new transplantation experiments to assess whether HSPCs isolated from old femur bone marrow can also affect skull bone marrow expansion. Transplantation of lineage-negative cells from the femur of old donors interestingly also increased vascularity of calvarial (but not femoral) bone marrow in the young recipient mice but to a lesser degree compared to old skull donors (Figure 3a-c, Extended Data Fig. 7a, b and below).

We also assessed *Vegfa* mRNA expression in the aging femur and observed decreased levels in HSPCs isolated from old femur compared to young femur (Extended Data Fig. 9a, b and below).

This data, along with our finding that there is higher hypoxia and less expression of inflammation cytokines in the old skull microenvironment, indicates that VEGF-A expression in HSPCs is governed by the type of skeletal element.

Skull vasculature of young vs. old HSPC transplantation

Femur vasculature of young vs. old HSPC transplantation

Femur Vegfa mRNA

Comment 3: While increased Vegfa expression by HSPCs in old mice is shown to drive the expansion of skull bone marrow and vasculature with age, the authors do not investigate whether Vegfa and bone marrow expansion is mechanistically linked to the resilience of skull-derived HSPCs during aging.

Response: We thank the reviewer for this comment. In addition to showing that VEGF-A is driving the expansion of skull vasculature and bone marrow, we would like to highlight that the newly added scRNA-seq data show higher expression of stress-induced inflammatory genes and a myeloid-biased program in HSCs from geriatric femur relative to geriatric skull (Extended Data Fig. 14d-f). Furthermore, the revised manuscript shows that head shielding is fully sufficient for long-term survival (beyond 200 days) after lethal irradiation whereas the same is not the case for leg shielding (Fig. 5h). Taken together, there is now compelling evidence indicating that BM from the skull is functionally of better quality and therefore more resilient than femoral bone marrow during aging.

To gain deeper insight into the relevance of VEGF-A in this context, we performed additional DC101 treatment (VEGFR2 inhibition) experiments. We analyzed hematopoietic, stromal, and endothelial cells in the skull and femur bone marrow of animals treated with the VEGFR2 blocking antibody DC101 for 12 weeks (Supplementary Data File 1 and smaller selection of graphs below). Compared to IgG controls, DC101-treated animals showed severe suppression of hematopoietic growth in the skull, which exceeded the regression of hematopoietic content in the femoral bone marrow. These results clearly show a particularly strong functional reliance of hematopoietic cells on VEGFR2 signaling in the skull.

Skull bone marrow FACS analysis after DC101 treatment

DC101

Femur bone marrow FACS analysis after DC101 treatment

DC101

To determine whether vascular heterogeneity contributes to the differences between BM in the aging skull and femur, we conducted a transcriptomic analysis of endothelial cells from geriatric skull and geriatric femur (Extended Data Fig. 16 and below).

scRNA-seq analysis of ECs

Consistent with the finding that the skull vasculature continues to expand during aging, we observed higher expression of vascular signaling receptors and angiogenic markers, such as *Kdr* (encoding VEGFR2), *Aplnr*, *Emcn*, *Flt4* and *Esm1*, in the geriatric skull. *Aplnr*, *Esm1* and *Flt4* (which encodes VEGFR3) are of particular significance because it is well established that their expression is regulated by VEGFR2 signaling (Kälin et al. 2007, PMID: 17412318; Tammela et al. 2008, PMID: 18594512; Rennel et al. 2007, PMID: 17362927; Rocha et al. 2014, PMID: 25057127). Thus, the scRNA-seq results are consistent with the notion that elevated levels of signaling by VEGF-A and VEGFR2 is one of the factors that drives the expansion of calvarial BM and distinguishes skull and femur.

Furthermore, in accordance with our general characterization of the BM in aging femur, we found that multiple pro-inflammatory and myeloid determination factors, such as *Hoxb9*, *Hotairm1*, *Ecrq4*, *Nt5e*, *Lgals3*, *Il6*, *S100a4*, *S100a6*, *S100a10*, *Gata6*, and *Edn1*, are significantly upregulated in ECs from aging femur relative to skull. Furthermore, skull bone marrow ECs had higher expression of several HSC-regulating factors such as *Ptn*, *Spp1*, *Gpr182*, *Nid1*, *Lama4*, *Robo4*, and *Flt4* (Extended Data Fig. 16f-h and below).

Inflammatory factors in geriatric ECs

HSC-regulating factors in geriatric ECs

Taken together, these data show that the BM compartment in skull relies strongly on VEGFA-VEGFR2 signaling and provides a comparably healthy microenvironment for hematopoietic cells.

Comment 4: Additionally, to further explore the role of VEGF as a driver of bone marrow expansion, the authors could examine the VEGF-A protein levels and mRNA expression in the context of pregnancy, stroke, and CML. Experiments investigating these mechanisms would grant insight into why the skull bone marrow exhibits a vascular response distinct from the femur under various physiological conditions.

Response: To assess whether there is a tissue-specific differential regulation of VEGF-A expression in response to the various pathophysiological conditions assessed in our study, ELISA for VEGF-A was performed on whole bone marrow

lysates of the skull and femur. Pregnancy, stroke, CML, and sustained PTH treatment all showed higher VEGF-A protein expression in the skull, whereas the only significant change (lower expression) in the femur was detected in mice with CML (Extended Data Fig. 8 and below). This data further highlights the high degree of vascular plasticity in the calvarial bone marrow. This explains the highly dynamic responses to changes in VEGF-A and probably other stimuli in a range of pathophysiological contexts.

ELISA for VEGF-A in various pathophysiological conditions

Comment 5: One of the key findings of this paper is that skull-derived HSPCs maintain their functional integrity with age. It is not evident whether this is a consequence of the dynamic skull microenvironment the authors have identified. The authors could perform colony forming unit assays to examine whether any cell-intrinsic proliferative features persist after their removal from the niche.

Response: Thank you for this comment. To distinguish whether cell-autonomous or microenvironmental factors enhance hematopoietic reconstitution potential in the skull, we performed serial colony forming unit (CFU) assays on FACS-sorted HSPCs. Primary and secondary CFU-F assays showed no significant difference in colony forming potential and myeloid differentiation potential between HSPCs from old skull and old femur (Extended Data Fig. 15 and below). These results suggest that the differences between skull and femur predominantly reflect the properties of the local microenvironment.

Colony forming unit assays of HSPCs

Overall Q&A-2: Clarity and context: Is the abstract clear, accessible? Are abstract, introduction and conclusions appropriate?

The abstract presents a clear outline of the findings in the paper. All conclusions in the abstract are well-supported by the data presented, and the authors highlight their novel findings.

In the introduction, the authors discuss previous literature that describes the functional role of crosstalk between structural elements of the skull in terms of immunity as well as dysfunctional myelopoiesis in aging in an effort to introduce the possibility of functional differences between long and flat bones. While they demonstrate that the calvarial bone marrow microenvironment exerts protective effects during aging, mechanistic connections between large-scale bone marrow

vascular expansion and the ability of skull-resident HSPCs to resist aging are not explored, leaving the questions posed in the introduction unanswered.

Response: Thank you very much for your feedback. We appreciate that our findings raise many new questions and will therefore lead to follow-up studies for many pathophysiological conditions. Future work, inspired by our findings, will also include a deeper exploration of the molecular interactions, which distinguish different BM compartments and enable local resilience in the skull. Our revised manuscript, however, provides first fundamental insights into the molecular processes driving BM expansion in response to aging and other stimuli. Blocking vascular growth via inhibition of VEGFR2, which is an essential component of the VEGF-A signaling pathway, leads to substantially less skull bone marrow. We also show by extensive flow cytometric analysis that most hematopoietic cell subpopulations and their supporting stromal cells in the skull bone marrow cannot expand without vascular growth, and we thereby establish that hematopoietic cell expansion depends on local angiogenesis (Supplementary Data File 1). Furthermore, we have characterized bone marrow endothelial and stromal subpopulations by scRNA-seq and identified several differentially regulated hematopoietic and inflammatory factors that qualify as promising candidates for future studies. DEG analyses identified a wide array of differentially regulated pathways between the two bone marrow compartments, highlighting tissue-specific heterogeneity of bone marrow ECs. Briefly, molecular pathways related to inflammation are significantly upregulated in the old femoral bone marrow, whereas the expression of several hematopoietic factors, as well as of angiogenic molecules, is significantly higher in the old skull. Therefore, our study lays solid groundwork into new and unexpected aspects of bone marrow heterogeneity and plasticity. We also establish a critical role of VEGF-A/VEGFR2 signaling in this process and identify further promising candidate regulators for follow-up studies.

The authors provide substantial evidence for their claims, supporting each conclusion with sufficient data.

Inflammatory material: Does the manuscript contain any language that is inappropriate or potentially libelous?

No.

Please indicate any particular part of the manuscript, data, or analyses that you feel is outside the scope of your expertise, or that you were unable to assess fully.

Referee #2 (Remarks to the Author):

In Koh et al., the authors examine the vascular compartment in the calvarium and observed differences in the aging associated changes in skeletal vasculature between calvarium and long bones. They also investigate whether these changes in turn translate into differences in calvarial leukopoiesis. The topic of whether calvarial marrow is fundamentally different than other marrow sites is timely and important, especially given very recent work such as Kolabas et al. Cell 2023 and other work as appropriately cited in the introduction to the present manuscript identifying differences between calvarial marrow and other marrow sites. This manuscript also makes some novel observations regarding differences in aging associated changes in calvarial vs long bone vasculature. However, the work overall is extremely descriptive and the depth of investigation into the areas considered, especially the links to the calvarial marrow and hematopoiesis, is shallow and lacking sufficient rigor. These issues greatly reduce the potential impact of the study.

Response: We appreciate the reviewer's comments and feedback. It is obvious that an individual manuscript cannot resolve all physiological and mechanistic aspects of an important fundamental finding. To strengthen the depth of our analysis and add further results supporting our main conclusions, we have performed a substantial amount of new experiments, such as an extensive flow cytometric analysis of key hematopoietic, stromal and endothelial cell populations in various pathobiological contexts (Supplementary Data File 1). These data prove unambiguously that dynamic changes in the calvarial vasculature are accompanied by clear-cut alterations in hematopoietic and BM stromal cells.

Major points:

Comment 1: The most important overall point is that the vast majority of the experiments are descriptive in nature, observing increases and decreases largely in calvarial vasculature with various treatments. Investigation into the mechanisms is very limited, and where performed relies on very well-established pathways regulating skeletal angiogenesis (see additional comments below). There is little overall data making the case that the differences observed carry physiologic or pathophysiological importance.

Response: Thank you very much for your comment. A recent study (Kolabas et al., PMID: 37562402) has proposed that mature immune cells of the skull bone marrow are molecularly distinct from BM in other parts of the murine but also human skeletal system. Furthermore, it has been shown that skull BM is functionally specialized for the immune surveillance of the central nervous system (Brioschi et al., PMID: 34083450; Cugarra et al., PMID: 34083447). In the context of these reports, our new findings regarding the dynamic expansion of skull BM in response to aging and other pathophysiological stimuli will be of great interest to the scientific community.

To strengthen the aspect of the physiological relevance, we have performed extensive flow cytometric analyses of 19 hematopoietic, 2 endothelial, and stromal cell subpopulations. This approach uncovered interesting quantitative changes and directly support that the changes in calvarial marrow cellularity reflect alterations in

hematopoietic, endothelial and stromal cells. These results also prove that the skull and femur bone marrow respond differently in a range of conditions.

With regard to the role of signaling by VEGF-A and VEGFR2, it is true that this pathway is known to be a master regulator of angiogenic growth. Nevertheless, there are instances where VEGF-A and its receptor are linked to unexpected biological functions such as the regulation longevity (Grunewald et al. 2021, PMID: 34326210). In our view, the expansion of calvarial bone marrow during adult life and aging, but also its dynamic response to a range of pathophysiological stimuli is clearly an unexpected and very fundamental discovery with far-reaching implications. Linking this expansion of bone marrow to the function of VEGF-A and VEGFR2 is therefore providing valuable insights into the molecular regulation of an important process. Finally, we have also strengthened this part by the analysis of VEGF-A levels in skull and femur for a number of key conditions (Fig. 4b and Extended Data Fig. 8). This new data reveals unexpected differences in VEGF-A levels in skull and femur, which helps to explain the dynamic striking responses in calvarial bone marrow. Finally, we also show how VEGF-A overexpression and VEGFR2 inhibition affect different hematopoietic, stromal and endothelial subpopulations (Supplementary Data File 1), which also proves that calvarial BM expansion requires VEGF-A/VEGFR2 signaling.

Comment 2: The whole-mount areal measurements of DAPI+ cells in Fig 2-4 are not a compelling way to make the case that there is expansion of total calvarial marrow. It is important that this analysis be conducted with a combination of histology and volumetric analysis, such as by uCT. Additionally, an analysis of marrow composition and absolute cell numbers of specific fractionated hematopoietic populations and endothelial populations should be demonstrated by flow cytometry. Areal measurements DAPI+ cells and vessel immunostaining are insufficient to make a case about marrow dynamics. In particular, using DAPI+ cells on whole mount to demonstrate marrow area is not seen as a rigorous approach, as the identity of the DAPI+ cells is unclear. A combination of flow cytometry and immunostaining of frozen sections for hematopoietic cell type defining markers is needed to demonstrate changes in total marrow amount and composition.

Response: Thank you for your comments and suggestion. To address this important issue and to gain further insight into how various hematopoietic and stromal cell types are differentially regulated in various pathophysiological conditions, we performed extensive flow cytometric analyses of 19 hematopoietic, 2 endothelial, and stromal cell subpopulations. As mentioned in our response to the first question, we acquired absolute cell numbers from calvarial and femoral bone marrow, which uncovered interesting quantitative changes and directly support that the changes in calvarial marrow cellularity reflect alterations in hematopoietic, endothelial and stromal cells. These results also prove that the skull and femur bone marrow respond differently in a range of conditions.

While we provide only the quantification related to Figure 1 below, all data pertaining to Figures 2-4 are provided within a separate supplementary file in the revised manuscript.

FACS quantification of aging skull bone marrow

Aging

FACS quantification of aging femur bone marrow

Aging

Comment 3: Is hydroxyprobe staining a linear indicator of oxygen tension? The fold comparisons may be misleading if this signal does not have a linear relationship between oxygen tension and probe signal within the measured range. The increase in hypoxia in skull vs femur bone marrow with aging would appear on its face to be counter to the increase in calvarial vascular density documented elsewhere in this manuscript, though it is appreciated that the flow dynamics involved may be more complex than simply scaling with vessel area. Some resolution of this apparent discrepancy is needed.

Response: We appreciate that hypoxyprobe is a surrogate marker for tissue hypoxia, which is nevertheless widely used for comparative measurements in the literature. A few relevant examples in the context of bone marrow are Parmar et al. 2007 (PMID: 17374716), Lévesque et al. 2007 (PMID: 17478585), Guarnerio et al. 2014 (PMID: 24936467), Nombela-Arrieta 2013 (PMID: 23624405), Rytelowski et al. 2020 (PMID: 32695673), and Ramalingam et al. 2023 (PMID: 37037837).

In contrast to other organs, the presence of sinusoidal vessels in the bone marrow is a poor predictor of local oxygenation. It has, for example, been shown that local oxygen tension is lowest in peri-sinusoidal regions (Spencer et al. 2014, PMID: 24590072; Nombela-Arrieta 2013, PMID: 23624405), which reflects poor perfusion of the sinusoidal vessel network (Parmar et al. 2007, PMID: 17374716; Lassailly et al. 2013, PMID: 23814020; Bixel et al. 2017, PMID: 28199850). This well-established issue is fully consistent with our new finding that local hypoxia in the skull is one of the drivers of the age-associated increase in VEGF-A expression (Fig. 4a-e and Extended Data Fig. 9c, d) and thereby promotes expansion of the calvarial bone marrow.

Comment 4: Data cataloguing VEGFA expression in calvarial vs long bone cells is descriptive, and no evidence is offered that these differences are determinative of the vascular phenotypes observed. Additionally, invoking differences in the HSPC compartment between skeletal sites as driving local vascular phenotypes would need to account for the ability of these HSPCs to circulate.

Response: Thank you for this comment. First of all, we would like to point out that we provide functional data showing that VEGF-A overexpression promotes expansion of the skull vasculature, whereas VEGFR2 inhibition (DC101) has the opposite effect (Fig. 4f-i). In the revised manuscript, we have expanded this data substantially (see below and Supplementary Data File 1) by showing that the alterations observed by imaging of the skull are supported by flow cytometric analyses of key cell populations. For example, VEGF-A overexpression leads to a strong expansion of hematopoietic, endothelial and stromal cell populations in skull but has no significant effect on hematopoietic cell content in femur. VEGFR2 inhibition leads to profound reductions in calvarial hematopoietic, endothelial and stromal cells, which are much stronger than the changes seen in femur. Thus, the observed changes in calvarial vasculature are a strong indicator of alterations affecting the local bone marrow.

As the reviewer points out correctly, HSCs can be isolated from multiple organs and evidence suggests that circulating HSCs maintain homeostasis of the hematopoietic system and participate in innate immune responses (Morita et al. 2011, PMID: 21185906; Mazo et al. 2011, PMID: 21802990; Mende et al. 2022, PMID: 35073399; Quaranta et al. 2024, PMID: 38446574). Nevertheless, it is also clear that microenvironments in different organs are not fully equivalent, leading, for example, to faster cycling of HSCs in spleen than in bone marrow (Morita et al. 2011, PMID: 21185906). In this context, our finding that HSCs in different BM compartments exhibit age-related functional differences is straightforward and, in hindsight, perhaps not totally surprising.

FACS analysis after *Vegfa* overexpression

FACS analysis after VEGFR2 inhibition

Comment 5: Related to comment 4 above, it isn't compelling to show that anti-VEGFA decreases calvarial vasculature and VEGFA expression decreases calvarial vasculature, as these relationships were already well established within the skeleton by Olsen and others many years ago. Instead, the potential novelty here lies in showing that fundamental local differences in VEGFA signaling are determinative of differences in skeletal vasculature between long bones and other sites, otherwise the VEGFA data doesn't strongly advance the conclusions of the manuscript.

Response: Thank you very much for this comment. We greatly appreciate that Bjorn Olsen has made seminal contributions to the bone field, including the findings that VEGF-A plays important roles in various stages of fracture repair or that autocrine, osteoblast-derived VEGF regulates bone formation. With regard to hematopoiesis, we would also like to highlight important findings by Hans-Peter Gerber and Napoleone Ferrara, who showed in 2002 that hematopoietic stem cell survival is controlled by an internal autocrine loop mechanism involving signaling by VEGF and VEGFR1 (PMID: 12087404). This finding is consistent with our new data showing elevated *Vegfa* expression in LSK cells. All these previous findings and the well-established role of VEGF-A as a master regulator of angiogenesis, however, do not diminish the novelty of our new findings, which show that VEGF-driven BM expansion in the skull occurs throughout life and in response to a range of pathophysiological stimuli. Furthermore, new data in the revised manuscript shows that skull BM shows strong condition-dependent fluctuations in VEGF-A level, whereas the same is not the case in femur (Extended Data Fig. 8). Moreover, qPCR analysis of sorted BM subpopulations shows that *Vegfa* expression is strongly elevated in LSK cells from aging skull but significantly decreased in femur (see below and Fig. 4a, Extended Data Fig. 9a).

qRT-PCR analysis for *Vegfa* expression
in FACS-sorted BM subpopulations

scRNA-seq transcriptomic analysis of BM endothelial cells also revealed higher expression of vascular signaling receptors and angiogenic markers, such as *Kdr* (encoding VEGFR2), *Aplnr*, *Emcn*, *Flt4* and *Esm1*, in the geriatric skull relative to age-matched femur (see below and Extended Data Fig. 16d, e). *Aplnr*, *Esm1* and *Flt4* (which encodes VEGFR3) are of particular significance because it is well established that their expression is regulated by VEGFR2 signaling (Kälin et al. 2007, PMID: 17412318; Tammela et al. 2008, PMID: 18594512; Rennel et al. 2007, PMID: 17362927; Rocha et al. 2014, PMID: 25057127). Thus, the scRNA-seq results are consistent with the conclusion that elevated levels of signaling by VEGF-A and VEGFR2 is one of the factors that drives the expansion of calvarial BM and distinguishes skull and femur.

scRNA-seq analysis of ECs

Comment 6: The finding of a potential difference in myelopoiesis between calvarium and other skeletal sites is interesting and potentially important, but not rigorously established. The shielding studies in Fig 5e are difficult to interpret, as differences in the secondary responses to the irradiation induced reductions in total body hematopoietic output could alter the results. Direct investigation of myeloid precursors and the discrete steps in the myeloid differentiation cascade in the calvarium vs long bone is essential to make the desired conclusions. Application of the already established photoconversion system in Fig 5f could be helpful towards demonstrating that, if site specific differences in myeloid composition are observed, that these translate into differential contributions to the peripheral myeloid pool.

Response: Thank you for this feedback. To provide direct evidence that HSPCs and committed progenitors of the skull are more resistant to pathological skewing toward a myeloid lineage during aging, we performed extensive flow cytometric analyses to comprehensively compare myeloid precursors and progeny in skull and femoral bone marrow during aging. Specifically, L⁻S⁺K⁺ progenitors, common myeloid progenitors (CMP), granulocyte-monocyte progenitors (GMP), and CD11b⁺ myeloid cells were significantly increased in the old femur compared to skull bone marrow

within the same animal. We also found substantial increases in mRNA expression of myeloid determination factors, *Cebpa* and *Spi1*, in the old femur versus age-matched skull bone marrow (see below and Figure 5d, e). This data clearly demonstrates that the myeloid determination/differentiation program is significantly upregulated in the entire myeloid hierarchy of the aging femur compared to skull, which also strongly supports our observation that the ultimate peripheral blood contribution is also severely affected.

Quantification of myeloid progenitor populations and myeloid determination factors

Comment 7 As an extension of comment 1 above, the increase in vascular area in pregnancy, stroke, CML and PTH treatment are all interesting findings, but not impactful or strongly significant unless the specific physiologic/pathophysiologic importance of the differential properties of the calvarial vs long bone vascular response can be demonstrated, at least for some subset of these conditions. Simply cataloging vascular changes by site is insufficient for a report at this level.

Response: Thank you for this comment. We have substantially extended our analyses of several conditions, namely pregnancy, stroke, leukemia, and PTH treatment, to strengthen our argument that the skull bone marrow microenvironment behaves differently than that of femur. In the original submission, we had already shown that various conditions and treatments affecting HSPCs lead to differential changes in the vasculature of skull and femur. The revised manuscript contains extensive flow cytometric analyses of critical cell populations, including total hematopoietic cells together with LSK cells as well as various HSC/HSPC and progenitor subpopulations (see a short summary for some of the populations below or the full results in Supplementary Data File 1). This extensive data set provides further compelling evidence for the differential behavior of bone marrow in skull and femur in response to pregnancy, stroke, CML or PTH treatment. All these conditions have in common that they elicit a strong expansion of hematopoietic cells,

HSCs/HSPCs, endothelial and stromal cells in the skull, whereas the alterations in femur are comparably less profound (pregnancy), affect fewer subpopulations (CML, PTH) or even lead to the reduction of several critical cell populations (stroke). Taken together, the newly added results clearly show that skull bone marrow is a highly dynamic hematopoietic compartment that expands strongly in response to a range of pathophysiological challenges and behaves very differently from femur.

Pregnancy

Stroke

CML

PTH treatment

Comment 8: Similar issues apply to the cytokine studies in Fig 5c—it is not enough to profile these, but there needs to be a deeper investigation of the responsible cellular sources and determination that some element of these distinct cytokine profiles impacts the aging of the calvarial vs long bone marrow.

Thank you for your feedback. First of all, we would like to point out that there are excellent studies on the topic of stromal niche inflammation during aging, which involves changes in osteoprogenitors, central marrow LepR⁺ mesenchymal stromal cells and deterioration of the sinusoidal vasculature (Mitchell et al. 2023, PMID: 36650381). In addition, there are intrinsic changes in hematopoietic stem cells (e.g. autophagy, metabolic stress, skewed differentiation and chromosomal instability) and their progeny but also extrinsic alterations (infections and metabolic changes), which together promote the expression of pro-inflammatory cytokines and other detrimental processes (Dellorusso et al. 2024, PMID: 38754428; Batsivari et al. 2020, PMID: 31907409; Mitroulis et al. 2020, PMID: 32849521; Kovtonyuk et al. 2016, PMID: 27895645).

Given the complexity of these processes, it is obvious that questions such as specific cellular sources of cytokines, their relevance for BM aging and the differences between skull and femur require a separate investigation and cannot be adequately addressed within the current manuscript. Nevertheless, we have added new scRNA-seq results for FACS-sorted cKit⁺ HSPCs and hematopoietic lineage-depleted stromal/endothelial cells isolated from either bone marrow compartment in young or geriatric mice (see below and Extended Data Fig. 13, 14, 16). Sub-clustering analysis identified distinct HSC, intermediate, myeloid, lymphoid, and megakaryocytic progenitors within the HSPC samples and various subtypes of endothelial, mesenchymal, and mural cells in the stromal samples (which are further interpreted in response to a later comment). Although no substantial proportional increase in the HSC subcluster could be observed in this lineage-depleted cell atlas, further sub-clustering of HSCs yielded 3 distinct subpopulations, of which HSC1 showed the highest expression of quiescence/stem cell maintenance markers (*Txnip*, *Mpl*, *Hlf*). This cluster also had the lowest expression of proliferation markers (*Top2a*, *Mki67*), which were most highly expressed by HSC2. Notably, HSC1 proportion rose drastically with aging in both the skull and in the femur. Strikingly, DEG analysis shows significant differences in the expression of stress-induced inflammatory genes. Many of these genes, such as *Hspa1b*, *Jund*, *Fosb*, *Dusp1*, *Klf1*, *Klf2*, and *Klf4* are higher in HSCs from the geriatric femur relative to age-matched skull. Gene expression of factors regulating myeloid differentiation and response, such as *Jun*, *Fos*, *Zfp36*, *Ngp*, *Ltf*, *Chil3*, *Egr1*, *S100a8*, and *S100a9*, is also substantially higher in HSCs isolated from geriatric femur. Since clonal hematopoiesis is a major hallmark of aging which exponentially exposes aging HSPCs to detrimental mutations, we performed gene set enrichment analysis (GSEA) of our geriatric femur HSC DEGs with that of Tet2-knockout (KO) HSCs, which exhibit clonal hematopoiesis with severe myeloid-biased inflammation (Moran-Crusio et al., PMID: 21723200; Izzo et al., PMID: 32203468) that is instigated and further exacerbated by increased IL-1 signaling (Caiado et al., PMID: 36379023; McClatchy et al., PMID: 38062031). This analysis revealed that femoral HSCs show strong enrichment of the aforementioned inflammatory pathways occurring in aging-associated clonal hematopoiesis.

scRNA-seq analysis of bone marrow

scRNA-seq of cKit⁺ HSPCs and Lin⁻ stromal cells

scRNA-seq analysis of HSCs

scRNA-seq analysis of HSCs

Therefore, our targeted analysis of HSCs clearly demonstrates that the skull bone marrow helps to maintain HSC properties by evading inflammatory stress-induced damage and the resulting myeloid bias. We have also generated a valuable resource and established the important groundwork for future studies of HSCs and progenitors with respect to regulatory and crosstalk mechanisms of HSC differentiation that are unique to the aging skull.

Comment 9: The studies shown describe differences in the phenotypic aging of the stromal compartment, but do not make a strong case for differences in phenotypic aging within the HSC compartment other than investigating relative myeloid output. Key molecular hallmarks of aging should be investigated within the HSC compartment. As above, any investigation of this topic needs to account for circulation of HSCs and deconvolute the degree to which phenotypes observed represent niche vs HSC-lineage intrinsic effects. Do any hematopoietic effects observed relate to the vascular phenotypes studied in the rest of the manuscript? No data is presented to demonstrate that this is the case, so these studies of hematopoiesis seem disconnected from the rest of the manuscript and lack any mechanistic basis.

Response: Thank you for your comments. Some of the mentioned issues were already discussed in our response to earlier questions. Without repeating ourselves, we would like to refer to the flow cytometric analysis of hematopoietic and stromal cell populations from various conditions including aging (Supplementary Data File 1), the single cell RNA-sequencing analysis of HSCs/HSPCs and bone endothelial cells (Extended Data Fig. 13, 14, 16), and colony formation assays (Extended Data Fig. 15).

scRNA-seq analysis of ECs

DEGs in geriatric skull vs. femur ECs

Inflammatory factors in geriatric ECs

HSC-regulating factors in geriatric ECs

Thus, our imaging data showing dynamic changes of the bone vasculature and surrounding marrow is now directly supported by a detailed flow cytometric analysis of hematopoietic cell subpopulations, which confirms that vascular growth and bone marrow expansion in the calvarium are directly linked.

These findings are also fully consistent with our partial irradiation/shielding experiments, which show that head shielding is sufficient for long-term survival whereas all animals died within 200 days after irradiation with shielded hindlimbs (Fig. 5f-h). The expansion of skull marrow is also directly supported by photoconversion experiments, which show that marrow from older mice has a significantly larger contribution to CD45+ cells in peripheral blood than middle-aged animals. In turn, skull from middle-aged mice, has a significantly larger hematopoietic output relative to young animals (Fig. 5i, j).

As already discussed in the context of question 4, the reviewer is obviously correct in stating that HSCs can circulate. Indeed, HSCs can be isolated from multiple organs and evidence suggests that circulating HSCs maintain homeostasis of the hematopoietic system and participate in innate immune responses (Morita et al. 2011, PMID: 21185906; Mazo et al. 2011, PMID: 21802990; Mende et al. 2022, PMID: 35073399; Quaranta et al. 2024, PMID: 38446574). Nevertheless, it is also clear that microenvironments in different organs are not fully equivalent, leading, for example, to faster cycling of HSCs in spleen than in bone marrow (Morita et al. 2011, PMID: 21185906). The sum of our data argues that microenvironmental conditions are more favorable in aging skull, which leads to detectable transcriptomic and functional differences in HSCs/HSPCs.

Minor points:

Comment 1: The vATPase imaging in Sup Fig 1c is not ideal as the stained cells are not clearly demarcated as individual cells but more as a swath of staining near the bone.

Response: Images were reacquired with different settings to avoid excessive signal. High-magnification images were also reacquired to show distinguishable osteoclasts with distinct nuclei (Extended Data Fig. 1c and 1d).

Immunofluorescent staining of activated osteoclasts in aging skull

Comment 2: Given recent reports on skeletal lymphatics and the topic of the current report, it would seem to be an obvious and related question whether lymphatics differ between calvarium and long bones.

Response: The presence (or absence) of lymphatic vessels in bone is a highly controversial issue that requires careful examination and is not within the scope of the current manuscript. In our data from different ongoing projects, we have so far failed to find evidence for lymphatic vessels inside bone with transgenic *Prox1* reporter mice, immunostaining and scRNA-seq so that a comparison of different skeletal elements is futile.

Comment 3: Is the measurement of sinusoid diameter in Fig 1 derived from the whole mount imaging? This analysis would likely be best conducted on histologic

sections given the possibility that vessels "stacked" and overlapping the z-plane could give a misleading sense of vessel diameter.

Response: Vessel diameter in Fig. 1 was measured by selecting the z-plane image with the widest vessel diameter from the z-stack of the individual vessel. This method excludes the possibility of measuring the diameter of overlapping vessels.

We describe the morphometric analysis in the Methods.

Comment 4: Considering stromal cells as total lin-sca1-kit- cells for the purpose of gene expression analysis in 242 is somewhat problematic. Some stromal cells do express Sca1, so the exclusion of the Sca1+ stromal cells here is puzzling and appears arbitrary. Also, this approach lumps a large number of stromal cells, each with likely very different VEGFA expression, together and makes interpretation difficult.

Response: To address this issue, we performed scRNA-seq analysis and found that, in hematopoietic progenitors and all bone marrow stromal/endothelial cells, *Vegfa* expression is highly restricted to HSPCs, as originally stated in our manuscript.

Vegfa expression in bone marrow

Comment 5: It is recommended that qPCR studies in Ext Fig 8a also be conducted in long bones for comparison.

Response: Thank you for this comment. We have generated the qPCR data for femur, which shows that *Vegfa-120* and *Vegfa-164* are the predominant isoforms (see below and Extended Data Fig. 9b in the revised manuscript).

qRT-PCR analysis of *Vegfa* isoforms

Comment 6: Discussing the femoral bone marrow as being "more inflammatory" on the basis of cytokine profiling seems overbroad.

Response: Most of the cytokines increased in the aging femur have been extensively characterized as pro-inflammatory cytokines. Numerous studies, including a recent publication on stromal-mediated inflammation in the aging bone marrow (Mitchell et al., PMID: 36650381) come to similar conclusions based on tissue cytokine array analysis. Nevertheless, we have toned down our statements and refer to our findings as higher levels of pro-inflammatory cytokines.

Referee #3 (Remarks to the Author):

Ralph Adam's group has employed sophisticated genetic models and functional assays to show that in the skull of mice there are continued formation of VEGFR2+ large caliber sinusoidal endothelial cells that support hematopoietic stem and progenitors (HSPCs) survival and expansion. They show that the release of Vgfa120, and Vegfa165 by the hematopoietic cells or the parathyroid hormone turns on VEGFR2 thereby activating angiogenesis in the skull sinusoidal endothelial cells expanding and sustaining the LSK HSPC populations. They also show that the endothelial cells formed within the skull compose of younger endothelial cells with less inflammation.

Comments:

This a very well-executed, well-controlled and well-designed and important paper demonstrating the heterogeneity of the marrow vascular niches for the support of the adult and aging hematopoiesis. The characterization of the various vascular beds within the skull are well performed and the genetic models clearly highlights the significance of VEGF-A and VEGFR2 in mediating angiogenesis and LKS expansion.

Response: We appreciate the reviewer's overall positive comments and excellent suggestions for improving the manuscript.

Major points:

The only concerns I have are as follows:

Comment 1: To prove that indeed true LKS hematopoietic stem cells are populating sinusoidal endothelial cells, the authors might consider performing secondary transplantation of the primary engrafted LKS populations.

Response: We appreciate the reviewer raising this issue. It has already been shown that skull HSPCs are capable of secondary hematopoietic reconstitution, which is indistinguishable from the reconstitution potential of femur HSPCs (Lassailly et al. 2014, PMID: 23814020). To provide a timely response to this important question and to get conclusive insight into differences hematopoietic reconstitution comparison between skull and femur HSPC with partial shielding, we re-irradiated the animals that previously reconstituted the entire hematopoietic system from partially shielded bone marrow compartments with the same partial shielding against lethal irradiation (Extended Data Fig. 11). This experimental scheme allowed us to directly challenge HSPCs from the skull and femur for a second round of full reconstitution without the complications/artifacts of transplantation. Comparison between peripheral blood derived from primary and secondary reconstitution showed no significant difference between the shielded head or shielded legs, demonstrating the long-term reconstitution potential of skull HSPCs (see below and Extended Data Fig. 11).

Experimental scheme for serial partial irradiation

FACS analysis of peripheral blood reconstitution

2) Inclusion some type of human data to prove that same phenomenon might be applicable to humans aging or young skull could be revealing.

Response: As shown in original Extended Data Fig. 2, we provide CT scan data of young vs old human subjects to show increasing diploe space during aging. Direct imaging of tissue specimens for hematopoietic and stromal cells, as done by Kolabas et al. (PMID: 37562402), would indeed be informative but requires major logistics and access to human tissue. Nevertheless, Kolabas et al. show beyond any doubt that the diploe space is filled by bone marrow in human skull.

3) Does DC101 the anti-VEGFR2 neutralizing antibody also shuts down angiogenesis in the marrow thereby impairing hematopoiesis or only targets skull angiogenic endothelial cells in aging mice?

Response: This important question was addressed by performing FACS analyses in the DC101-treated mice. The results showed that endothelial cells and all hematopoietic cell types were substantially decreased in the skull and femur (see Supplementary Data File 1 and a selection of graphs below). In addition, peripheral blood analysis did not show a bias toward a specific lineage (see below).

Skull bone marrow FACS analysis after DC101 treatment

DC101

Femur bone marrow FACS analysis after DC101 treatment

DC101

Peripheral Blood

Reviewer Reports on the First Revision:

Referees' comments:

Referee #1 (Remarks to the Author):

The authors have addressed all concerns and are to be congratulated for outstanding work.

Referee #2 (Remarks to the Author):

The revisions to the Koh et al. manuscript offer a good amount of additional data, addressing several comments though only partially addressing the broader concern that the report is descriptive. A strength of the revision is that additional flow cytometry data profiling marrow cellular content was added and appears robust and appropriate, clearly addressing comment 2. This now makes the finding of the report that there are cellular compositional differences in the marrow between calvarial and long bones solid. The requested improvements to osteoclast staining are also satisfactory. The scRNA-seq data provided in response to comment 8 addresses the request for investigation of the cellular sources for cytokine differences between long bone and calvarial marrow. However, the concern raised across several points on the initial submission that the report is descriptive was only partially addressed. In general, the response to these points have added additional descriptive data, which is generally helpful and improve the rigor of the manuscript, but do not fundamentally address the larger concern that the report is descriptive and observational. An exception to this criticism should be noted, that the VEGFA overexpression and VEGFR2 inhibition data and summarized in the rebuttal letter are helpful in building towards functional demonstration of the associated mechanisms, though it is seen as important that controls for the degree of VEGFA overexpression in long bones vs calvaria be provided, such controls were not immediately evident in Supp Data File 1. In its current state the manuscript makes a robust case that there are differences between long bones and vertebrae marrow, but does not make a very strong case that these differences are important and only a limited functional case for the associated mechanisms.

Referee #3 (Remarks to the Author):

In the revised manuscript the authors have performed extensive additional experiments which have increased the novelty and significance of the manuscript. The authors have satisfactorily addressed the majority of my concerns:

Specifically that authors have added these new data that have improved the impact of the current manuscript:

1) Inclusion of comprehensive flow cytometric and single cell-RNA sequencing profiling of the vascular, stromal and hematopoietic constitution of calvarium and femoral bones in aging, CML, PTH, and stroke as well as PGE2 and AMD3100 challenges. These studies demonstrate the unique features of the skull vasculature in protecting hematopoiesis against various stressors.

2) Molecular profiling of the HSPCs in the skull indicates that these cells are phenotypically more primitive. Similarly, specific subsets of the endothelial cells in skull express more Apelin-receptor, endomucin, Flt4 and ESM1 along with angiocrine factors, including Spp1, GPR182, LAMA4 and Robo1, among others which could explain their pro-angiogenic and pro-HSPC sustenance potential.

3) Addition of more revealing immunohistochemical analyses of the calvarium bone vascular remodeling.

4) Inclusion of post irradiation skull and leg shielding experiments, confirming the long-term reconstitution potential of both Skull and femoral resident HSPCs.

5) The authors now show the extreme sensitivity of the skull and femoral vasculature to VEGF-A/VEGFR2 dependent signaling by performing inhibition and activation studies. They also validate induction of the Vegf-A message upon various physiological and pathophysiological challenges.

Minor concern:

The authors might consider quoting the paper by Hooper AT et al, Engraftment and reconstitution of hematopoiesis is dependent on VEGFR2 mediated regeneration of sinusoidal endothelial cells. *Cell Stem Cell*, 4(3): 263–274, 2009, that supports their conclusion on the significance of VEGF-A/VEGFR2 as well as VEGF-C/VEGFR3 in the regulation of hematopoiesis.

Overall, the revised manuscript relays important findings that could be of great interest to broad readership worthy of publication.

Author Rebuttals to First Revision:

Again, we would like to thank all reviewers for their time, effort and constructive suggestions, which are greatly appreciated and have enabled us to improve the manuscript.

Referees' comments:**Referee #1** (Remarks to the Author):

The authors have addressed all concerns and are to be congratulated for outstanding work.

Thank you very much for all your constructive feedback, which is most appreciated.

Referee #2 (Remarks to the Author):

The revisions to the Koh et al. manuscript offer a good amount of additional data, addressing several comments though only partially addressing the broader concern that the report is descriptive. A strength of the revision is that additional flow cytometry data profiling marrow cellular content was added and appears robust and appropriate, clearly addressing comment 2. This now makes the finding of the report that there are cellular compositional differences in the marrow between calvarial and long bones solid. The requested improvements to osteoclast staining are also satisfactory. The scRNA-seq data provided in response to comment 8 addresses the request for investigation of the cellular sources for cytokine differences between long bone and calvarial marrow. However, the concern raised across several points on the initial submission that the report is descriptive was only partially addressed. In general, the response to these points have added additional descriptive data, which is generally helpful and improve the rigor of the manuscript, but do not fundamentally address the larger concern that the report is descriptive and observational. An exception to this criticism should be noted, that the VEGFA overexpression and VEGFR2 inhibition data and summarized in the rebuttal letter are helpful in building towards functional demonstration of the associated mechanisms, though it is seen as important that controls for the degree of VEGFA overexpression in long bones vs calvaria be provided, such controls were not immediately evident in Supp Data File 1. In its current state the manuscript makes a robust case that there are differences between long bones and vertebrae marrow, but does not make a very strong case that these differences are important and only a limited functional case for the associated mechanisms.

We appreciate the reviewer's detailed feedback and recognition of the additional data we provided. However, we respectfully disagree with the characterization that our manuscript is primarily descriptive. While our study indeed includes descriptive elements necessary to establish a comprehensive understanding of bone marrow heterogeneity, our findings also reveal fundamental insights into the distinct biological processes occurring in different bone environments.

The differences in cellular composition between calvarial and long bone marrow, as demonstrated through flow cytometry and scRNA-seq data, are not merely

observational. These findings uncover essential variations in the cellular landscape and cytokine profiles that influence bone homeostasis and remodeling. Importantly, these differences suggest distinct microenvironmental niches, which have implications for understanding bone marrow physiology and various pathologies. Likewise, our VEGF-A overexpression and VEGFR2 inhibition experiments go beyond mere description. These studies provide mechanistic insights into how this pathway differentially regulates vascularization and BM expansion.

Overall, our manuscript not only documents important differences in bone marrow environments but also begins to elucidate the underlying mechanisms driving these differences. By integrating descriptive data with functional experiments, we believe our study provides a robust foundation for future investigations and makes significant contributions to an important field.

Referee #3 (Remarks to the Author):

In the revised manuscript the authors have performed extensive additional experiments which have increased the novelty and significance of the manuscript. The authors have satisfactorily addressed the majority of my concerns:

Specifically that authors have added these new data that have improved the impact of the current manuscript:

- 1) Inclusion of comprehensive flow cytometric and single cell-RNA sequencing profiling of the vascular, stromal and hematopoietic constitution of calvarium and femoral bones in aging, CML, PTH, and stroke as well as PGE2 and AMD3100 challenges. These studies demonstrate the unique features of the skull vasculature in protecting hematopoiesis against various stressors.
- 2) Molecular profiling of the HSPCs in the skull indicates that these cells are phenotypically more primitive. Similarly, specific subsets of the endothelial cells in skull express more Apelin-receptor, endomucin, Flt4 and ESM1 along with angiocrine factors, including Spp1, GPR182, LAMA4 and Robo1, among others which could explain their pro-angiogenic and pro-HSPC sustenance potential.
- 3) Addition of more revealing immunohistochemical analyses of the calvarium bone vascular remodeling.
- 4) Inclusion of post irradiation skull and leg shielding experiments, confirming the long-term reconstitution potential of both Skull and femoral resident HSPCs.
- 5) The authors now show the extreme sensitivity of the skull and femoral vasculature to VEGF-A/VEGFR2 dependent signaling by performing inhibition and activation studies. They also validate induction of the Vegf-A message upon various physiological and pathophysiological challenges.

Minor concern:

The authors might consider quoting the paper by Hooper AT et al, Engraftment and reconstitution of hematopoiesis is dependent on VEGFR2 mediated regeneration of sinusoidal endothelial cells. *Cell Stem Cell*, 4(3): 263–274, 2009, that supports their conclusion on the significance of VEGF-A/VEGFR2 as well as VEGF-C/VEGFR3 in the regulation of hematopoiesis.

Overall, the revised manuscript relays important findings that could be of great interest to broad readership worthy of publication.

Thank you very much for the precise summary and all your feedback, which proved most helpful. We have included the additional reference.